# Dissecting the impact of transcription factor dose on cell reprogramming heterogeneity using scTF-seq

Wangjie Liu [1,2,6], Wouter Saelens [1,2,3,4,6], Pernille Rainer[1,2,6], Marjan Biočanin[1,2], Vincent Gardeux [1,2], Antoni Jakub Gralak [1,2], Guido van Mierlo [1,2], Angelika Gebhart [1,2], Julie Russeil [1,2], Tingdang Liu[5], Wanze Chen [1,5 ✉] & Bart Deplancke [1,2 ✉]

Reprogramming often yields heterogeneous cell fates, yet the underlying mechanisms remain poorly understood. To address this, we developed single-cell transcription factor sequencing (scTF-seq), a single-cell technique that induces barcoded, doxycycline-inducible TF overexpression and quantifies TF dose-dependent transcriptomic changes. Applied to mouse embryonic multipotent stromal cells, scTF-seq generated a gain-of-function atlas for 384 mouse TFs, identifying key regulators of lineage specification, cell cycle control and their interplay. Leveraging single-cell resolution, we uncovered how TF dose shapes reprogramming heterogeneity, revealing both dose-dependent and stochastic cell state transitions. We classified TFs into low-capacity and high-capacity groups, with the latter further subdivided by dose sensitivity. Combinatorial scTF-seq demonstrated that TF interactions can shift from synergistic to antagonistic depending on the relative dose. Altogether, scTF-seq enables the dissection of TF function, dose and cell fate control, providing a high-resolution framework to understand and predict reprogramming outcomes, advancing gene regulation research and the design of cell engineering strategies.

Understanding and controlling cell fates through gene regulatory programs, particularly through transcription factor (TF)-mediated cell reprogramming, are critical objectives in biomedical research. Past studies using the ectopic expression of single TFs or combinations have identified 'master regulators' that influence various cellular processes[1–3], including differentiation, transdifferentiation, dedifferentiation and reprogramming[4]. Here, we collectively refer to these processes as cell 'reprogramming'. For instance, the 'Yamanaka factors'

(OCT3/4, SOX2, KLF4 and c-MYC) can reprogram adult fibroblasts into induced pluripotent stem cells[5,6].

However, reprogramming is typically characterized by pronounced heterogeneity and inefficiency, posing a major challenge[4,7–9]. This reprogramming heterogeneity is not solely due to cell-to-cell variability of the starting population[9,10], as advancements in single-cell technology have revealed that cells can follow multiple branches along a reprogramming path[11]. In addition, inhibiting proliferation or

[1]Laboratory of Systems Biology and Genetics, Institute of Bioengineering, School of Life Sciences, École Polytechnique Fédérale de Lausanne (EPFL), Lausanne, Switzerland. [2]Swiss Institute of Bioinformatics, Lausanne, Switzerland. [3]Department of Biomedical Molecular Biology, Ghent University, Ghent, Belgium. [4]Laboratory of Myeloid Cell Biology in Tissue Homeostasis and Regeneration, VIB Center for Inflammation Research, Ghent, Belgium. [5]State Key Laboratory of Quantitative Synthetic Biology, Shenzhen Institute of Synthetic Biology, Shenzhen Institutes of Advanced Technology, Chinese Academy of Sciences, Shenzhen, China. [6]These authors contributed equally: Wangjie Liu, Wouter Saelens, Pernille Rainer. ✉e-mail: wz.chen@siat.ac.cn; bart.deplancke@epfl.ch

synchronizing the cell cycle substantially increased the reprogramming efficiency, emphasizing the critical role of the cell cycle in modulating a cell's reprogramming capacity[12]. Nevertheless, the molecular mechanisms underlying cell fate branching and TF–cell cycle interaction during reprogramming remain poorly understood. Another aspect that has historically received relatively little attention is the role of TF dose,

although TFs are known to vary in copy number over several orders of magnitude[13]. The dose of a TF does affect not only gene expression levels but also the set of targeted genes[13–15]. Consequently, TF dose may equally be key in steering cell reprogramming and thus account for the observed heterogeneity. The multifaceted nature of reprogramming is one of the primary reasons why it remains challenging to collectively

**Fig. 1 | scTF-seq design and the corresponding TF overexpression atlas.**
**a**, Schematic of the scTF-seq workflow. TF-ID, a unique barcode designed for mCherry (as control) or each individual TF; forward and reverse, primers to enrich TF-IDs. The arrayed screening schematic is created with BioRender.com. **b**, Fluorescence images of mCherry (red) and nuclei (DAPI, blue) in C3H10T1/2 cells treated without (no dox) or with doxycycline (dox). Representative images of more than three independent experiments. Scale bar = 125 μm. **c**, Schematic of the sequencing outputs of scTF-seq—count matrices of gene expression in 10x libraries (top) and ectopic TF-ID expression in TF-enrichment libraries (bottom) for each sequenced cell. **d**, Percentage of cell barcodes associated with TF-IDs in 10x or TF-enrichment libraries. Colors represent nine independent scTF-seq experiments (also referred to as 'batches', see color legend in **e**). Error bars represent the mean ± s.d. **e**, UMAP of scTF-seq data involving 45,987 cells

and 384 TFs after quality control and preprocessing (referred to as 'TF atlas'). Colors represent batches. **f**, Natural log-transformed TF expression levels (TF dose) in cells overexpressing individual TFs. Colors represent cell density (number of neighbors) after randomly sampling up to 500 cells for each TF. **g**, Left: RNAscope images for DAPI, WPRE (proxy for TF dose), ESR2–ORF in ESR2 (top) and control (bottom) cells. All fluorescence channels were merged for cell segmentation, indicated by the red (cell boundary) and purple (expanded cell boundary) outlines. Representative images of two independent experiments. Scale bar = 100 μm. Right: single-cell RNAscope quantification showing the log-normalized mean intensity of WPRE versus ESR2–ORF in control and ESR2 cells. Fitted model = LOESS (Extended Data Figs. 1 and 2). RT, reverse transcription; LOESS, locally estimated scatterplot smoothing; UMAP, uniform manifold approximation and projection; enrich., enrichment.

study heterogeneity-contributing factors and their influence on cell reprogramming, especially when using bulk assays that are constrained by population-averaging readouts.

To answer these questions, a systematic quantitative TF screen at the single-cell level is essential to link TF function with reprogramming efficiency. TF overexpression would thereby be preferred as it can induce cell reprogramming more efficiently than CRISPR activation due to post-translational regulation[16,17]. In the past 5 years, several studies have implemented TF overexpression screens by coupling pooled TF overexpression with high-throughput readouts of single-cell RNA sequencing (scRNA-seq) or single-cell multiomics[16,18–20]. However, none has systematically investigated the roles of TF dose, cell cycle and their interplay in steering cell reprogramming. To address this gap, we developed single-cell TF sequencing (scTF-seq), aligning doxycycline (dox)-inducible barcoded overexpression of individual TFs with transcriptomic changes captured by scRNA-seq. This allowed us to map reprogramming properties of each TF and its dose at single-cell resolution. We then conducted scTF-seq on mouse embryonic multipotent stromal cells (MSCs) for 419 mouse TFs in parallel. After rigid quality controls, the scTF-seq assays yielded a high-quality dataset that tabulates the TF overexpression level and respective TF-induced transcriptomic change for each of 45,978 cells linked to 384 TFs and 7 TF combinations. Our approaches identified previously undescribed cell reprogramming capacities of both known and uncharacterized TFs. In addition, we systematically studied heterogeneous molecular and cellular responses resulting from TF dose, stochasticity and/or cell cycle dynamics. Finally, targeted combinatorial TF analysis revealed that the same combination of TFs can interact synergistically and antagonistically depending on the TF dose. Our TF overexpression clone library, single-cell TF gain-of-function atlas and analytic frameworks serve as valuable resources for achieving a mechanistic understanding of TF roles in governing cell states.

## Results

### Constructing the scTF-seq library and single-cell atlas

To establish scTF-seq, we built a dox-inducible lentiviral open reading frame (ORF) library of 419 TFs, each tagged with a unique barcode

(termed TF-ID hereafter) close to the 3′ UTR, enabling precise TF identification and quantification through 3′ scRNA-seq (Fig. 1a,b and Supplementary Table 1; Methods). Notably, viral particles were produced by individually packaging each vector to avoid barcode recombination and ensure more efficient and controllable TF overexpression than pooled virus packaging as used in most published screens[3,16,18–20].

To assess the functionality of the scTF-seq library, we introduced it into mouse MSCs (C3H10T1/2)[21] through arrayed lentiviral packaging and transduction, enabling high transduction efficiencies and dox-induced overexpression of individual TFs (Fig. 1a and Supplementary Notes 1–4). We chose C3H10T1/2 cells for their multipotency to differentiate into adipocytes, chondrocytes, osteoblasts or myocytes, thus providing a diverse range of cell fates to investigate TF-driven reprogramming[22–24]. To correct for spontaneous differentiation of C3H10T1/2 cells when reaching confluence[21,25] and benchmark TF-induced changes, we included confluent and non-confluent mCherry-overexpressing cells as controls, and adipogenic cocktail-treated and *Myog*-overexpressing cells as references (Adipo ref and Myo ref; Methods). The transcriptomes of cells from nine batches were profiled using droplet-based scRNA-seq, while TF-IDs were enriched and robustly detected in parallel (Fig. 1a,c,d and Supplementary Note 5; Methods). After TF-ID assignment to cells and stringent quality control to remove low-quality cells and doublets (Extended Data Fig. 1a, Supplementary Table 2 and Supplementary Note 5; Methods), we obtained 45,978 cells covering 384 individual TFs and 7 TF combinations (detailed information is presented in the following sections). The number of cells (on average 116 cells per TF or TF combination) was uniformly distributed among TFs and batches, supporting the advantage of array-based sample preparation (Extended Data Fig. 1b). Leveraging the TF-enrichment library as a highly accurate and sensitive readout of the TF-ID, we quantified the TF overexpression level in a cell by the log-transformed unique molecular identifier (UMI) count of its assigned TF-ID (referred to from now on as TF dose). Batch effects were systematically evaluated and effectively corrected, allowing robust data integration (Fig. 1e and Supplementary Note 6).

As designed, the array-based lentiviral transfection and transduction strategies allow the implementation of a high multiplicity

**Fig. 2 | TFs directing lineage differentiation and immunomodulation.**
**a,b**, UMAP plot of the integrated TF atlas with control, functional and proliferating cells (referred to as the 'functional TF atlas'). Colors represent assigned TFs (**a**) and clusters (**b**). 'Ctr.conf' and 'Ctr.non.conf' in **a** represent confluent and non-confluent control (mCherry-overexpressing) cells, respectively. Colored circles in **b** highlight clusters having gene expression profiles related to myogenic, osteogenic, adipogenic lineages or immunomodulation (Inflammatory). **c**, Heatmap showing a pairwise Pearson correlation of functional TF cells annotated by TF (in column) and batch (in row). Cells are ordered by hierarchical clustering. The red dashed box represents the transcriptomic similarity of cells reprogrammed by FOS and ATF TF families. **d**, Dot plot showing a functional cell expression profile enrichment of each TF in the four main differentiation lineages of multipotent stromal stem cells. Only TFs having at least 25 functional

cells and enriched in at least one of the four lineages with adjusted $P < 0.05$ are shown. **e**, Fluorescence images of lipids droplets (stained with Bodipy, yellow) and nuclei (stained with DAPI, blue) in CEBPA, MYCN, RHOX12, PPARG and mCherry (control) cells after 5 days of dox-induced overexpression. Representative images of two independent experiments, with one to two independent wells for each. Scale bar = 100 μm. **f**, Standard boxplot (Methods) showing the quantified lipid scores (Bodipy area/DAPI area on the images shown in **e**) of individual TFs and the control. Data were collected from two independent experiments, with one to two independent wells for each. *$P < 0.05$, **$P < 0.01$, ***$P < 0.001$, pairwise two-sided $t$ test followed by false discovery rate (FDR) correction. See Supplementary Table 5 and Methods for statistics and exact $P$ values (Extended Data Fig. 3). Myo, myogenic; osteo, osteogenic; adipo, adipogenic.

of infection (MOI; Supplementary Notes 2 and 3), leading to broad viral copy number variations. This, together with differences in transcriptional activity driven by random transgene integration and promoter fluctuation, likely contributes to the substantial dose variation

observed across cells for most TFs (Fig. 1f). We validated that TF-ID counts correlate well with actual TF ORF expression using multiplex RNA in situ hybridization (RNAscope; Fig. 1g and Extended Data Fig. 1c). This supports the use of TF-ID counts as a reliable proxy for exogenous

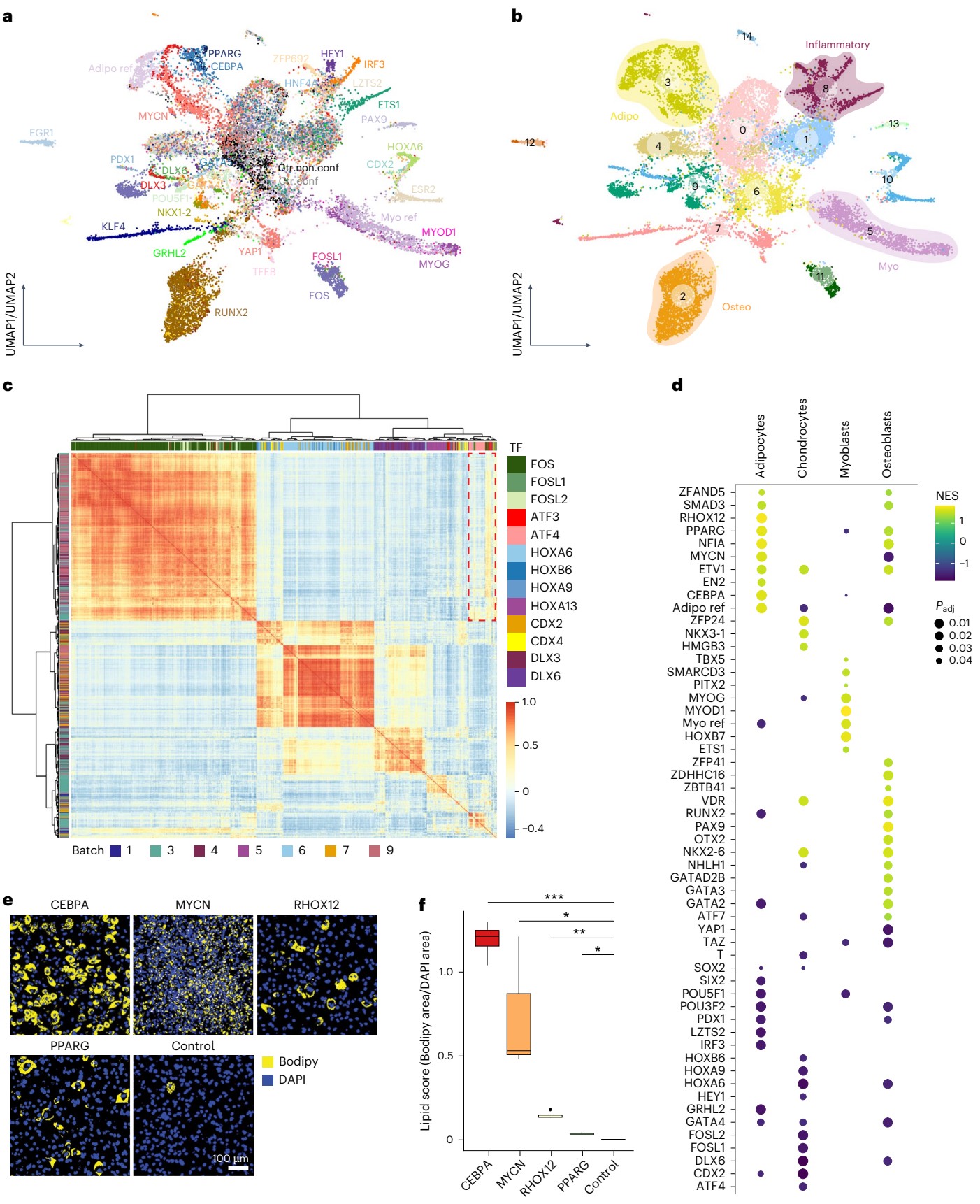

TF expression both at the RNA and protein level, which is a sensible approach given the generally reasonable correlation between mRNA and protein abundance across various contexts[26]. Finally, we determined that a wide dose range is critical for enhancing sensitivity in detecting differentially expressed genes (Extended Data Fig. 2a), uncovering both linear and nonlinear (and non-monotonic) dose-related effects missed in prior studies (Extended Data Fig. 2b–d).

### Identifying TFs directing lineage differentiation

As the activation of lineage developmental genes generally occurs in the G0/G1 phase[27], we focused on G0/G1 cells (Extended Data Fig. 2e and Supplementary Note 7) to study the roles of TFs in directing lineage differentiation. By quantifying TF-driven transcriptomic variation, we identified a subset of TF-overexpressing cells (simplified as 'TF cells' hereafter) that were transcriptomically similar to controls and labeled them as 'non-functional' (Supplementary Note 8). This was commonly observed among TFs but typically only in a subset of TF cells, implying that TF overexpression tends to induce various degrees of transcriptomic reprogramming. Upon closer inspection, we found that higher doses correlate with more pronounced transcriptomic changes, indicating TF dose as a primary determinant of this reprogramming heterogeneity (Supplementary Note 8). Subsequently, we performed clustering on the TF atlas excluding non-functional TF cells (Fig. 2a,b and Extended Data Fig. 3a,b; Methods). Clusters 2, 3 and 5 showed strikingly higher levels of lineage markers *Bglap2*, *Fabp4* and *Mylpf* (Extended Data Fig. 3c), representing osteogenic, adipogenic and myogenic programs, respectively. Adipo and Myo ref cells colocalized with clusters 3 and 5, respectively (Fig. 2a,b), validating the adipogenic and myogenic identities of these two clusters. Cluster 8 showed high expression of interferon-stimulated genes like *Isg15* and was enriched for inflammatory pathways (Extended Data Fig. 3c,d). Cells reprogrammed by HEY1 (ref. 28), LZTS2 (ref. 29), HNF4A[30] and ZFP692 were predominantly distributed in cluster 8. Despite the lack of clear functional information associated with inflammation for these TFs, the colocalization of their cells in cluster 8 with IRF3 cells (a well-established immunomodulator[31]) suggests their role in regulating inflammatory response genes.

We then computed TF–cell similarities to infer functional modules that govern the same gene expression programs (Extended Data Fig. 3e; Methods). As exemplified in Fig. 2c, pronounced intrafamily and interfamily correlations were detected among CDX, HOX, and DLX TFs, consistent with their shared role in anterior-posterior patterning and their common evolutionary origin[32]. However, correlations were less evident between HOXA13 and most TFs in these families (Fig. 2c), corroborating a distinct role for HOXA13 (refs. 33,34). Analogous functional characteristics were also observed for TFs with known physical interactions, such as the activator protein 1 (AP-1) formed by cross-family FOS and ATF family members[35]. These results emphasize the value of our scTF-seq atlas for exploring TF interactions and functional analogies.

Gene set enrichment analysis (Methods) recovered known MSC lineage-specific TFs, such as RUNX2, PAX9 and GATA2 for osteogenesis[36–39]; HOXB7, MYOG and MYOD1 for myogenesis[40–42]; NKX3-1 for chondrogenesis[43]; and SMAD3, PPARG and CEBPA for adipogenesis[44–46] (Fig. 2d). We also identified TF candidates not yet described as implicated in MSC lineages, including OTX2 in osteogenesis, HMGB3 in chondrogenesis and MYCN and RHOX12 in adipogenesis, as experimentally validated for the latter two TFs (Fig. 2d–f and Extended Data Fig. 3f). However, unlike CEBPA, PPARG and RHOX12 cells, MYCN cells lacked *Plin4* expression (Supplementary Table 3), a late adipocyte differentiation marker essential for lipid droplet association[47]. This is consistent with the smaller, scattered lipid droplets observed in MYCN cells (Fig. 2e,f). Thus, while all these TFs promoted adipogenesis, scTF-seq data suggest that MYCN may act using a distinct mechanism, which is explored further below.

### Quantifying TF reprogramming capacity and dose sensitivity

We then quantified the relative transcriptome variation between each cell and the centroid of controls (Fig. 3a; Methods). As expected, the transcriptomic alterations were overall greater in TF cells compared to control ones, as well as in functional TF cells relative to their non-functional counterparts (Extended Data Fig. 4a,b). To compare the exogenously expressed TF dose to the endogenous one in normal physiological contexts, we contrasted the minimal functional dose at which an overexpressed TF leads to a substantial transcriptomic difference to the dose observed in vivo (Methods). We found that, for about half of TFs, the exogenous functional dose aligns with its physiological range, including TFs such as *Runx2* in plasmacytoid dendritic cells, *Meis2* in neuron subsets and *Cebpa* in adipocytes (Fig. 3b–e and Extended Data Fig. 4c). Notable exceptions include *Pparg*, lipid ligand-activated, *Nfkb1*, inhibited in steady-state by IκB and various homeobox TFs that tend to function combinatorially (Fig. 3d and Extended Data Fig. 4c). By visualizing transcriptomic change over TF dose, we found that TFs differ in how their effect is modulated by dose (Fig. 3b–f). Some TFs induce substantial transcriptomic changes even at very low doses, while others require higher doses to achieve their effect plateau.

To better capture the TF dose–response relationships, we modeled the transcriptomic change in function of TF dose using a logistic model (Supplementary Note 9). Leveraging the model parameters, we defined TF reprogramming capacity and dose sensitivity, and broadly classified TFs into the following three major groups (Fig. 3f,g, Supplementary Table 4 and Supplementary Note 9): (1) 32 high-capacity and high-dose-sensitive TFs, including HOX and CDX TFs; (2) 44 high-capacity and low-dose-sensitive TFs, such as POU5F1, that required a high dose to reach high capacity and (3) 158 low-capacity TFs like VDR that induced no to only very mild transcriptomic effects across a wide dose range.

To explore the functional relevance of TF reprogramming capacity, we analyzed mutational constraint data, including the probability of loss-of-function intolerance and loss-of-function observed/expected upper bound fraction, from gnomAD[48,49] for human orthologs (Supplementary Note 10). We found that high-capacity TFs are substantially enriched among genes intolerant to loss-of-function mutations (Supplementary Table 5), suggesting a more substantial impact on cellular and ultimately organismal phenotypes compared to low-capacity ones[48,50]. Enrichment analysis on TF classes showed that zinc-finger TFs were under-represented and homeodomain TFs over-represented among high-capacity TFs (Supplementary Table 5; Methods). Moreover,

**Fig. 3 | Characterizing TF dose sensitivity and reprogramming capacity. a**, UMAP plot of the TF atlas after regressing out the heterogeneity specific to control cells, colored by overall transcriptomic changes (Methods). **b–e**, Comparison of physiological and exogenous dose for RUNX2 (**b**), MEIS2 (**c**), CEBPA (**d**) and PPARG (**e**). Top scatterplots indicate the change in overall transcriptomic response (distance in PCA space to control cells) over various doses. The dashed line represents the minimal functional dose at which the overall transcriptomic change is above 0.23. Bottom boxplots show the range of doses in the given cell type (boxes representing 25th and 75th percentiles, with 1.5× IQR as whiskers and the mean as the white dot). Endogenous TF expression for induced adipogenesis or myogenesis (teal), the endogenous TF expression in mCherry-overexpressing cells (blue) and this expression added to the exogenous expression (purple). **f**, Dot plot showing the scaled, overall transcriptomic change of TF-overexpressing cells over TF dose. Each dot represents a cell. Each row represents a TF. Color bars on the left represent TF groups categorized according to dose sensitivity and reprogramming capacity. **g**, Scatterplot showing the overall transcriptomic change of one representative TF of each TF category across TF dose. The lines represent the fitted logistic regression (Extended Data Fig. 4).

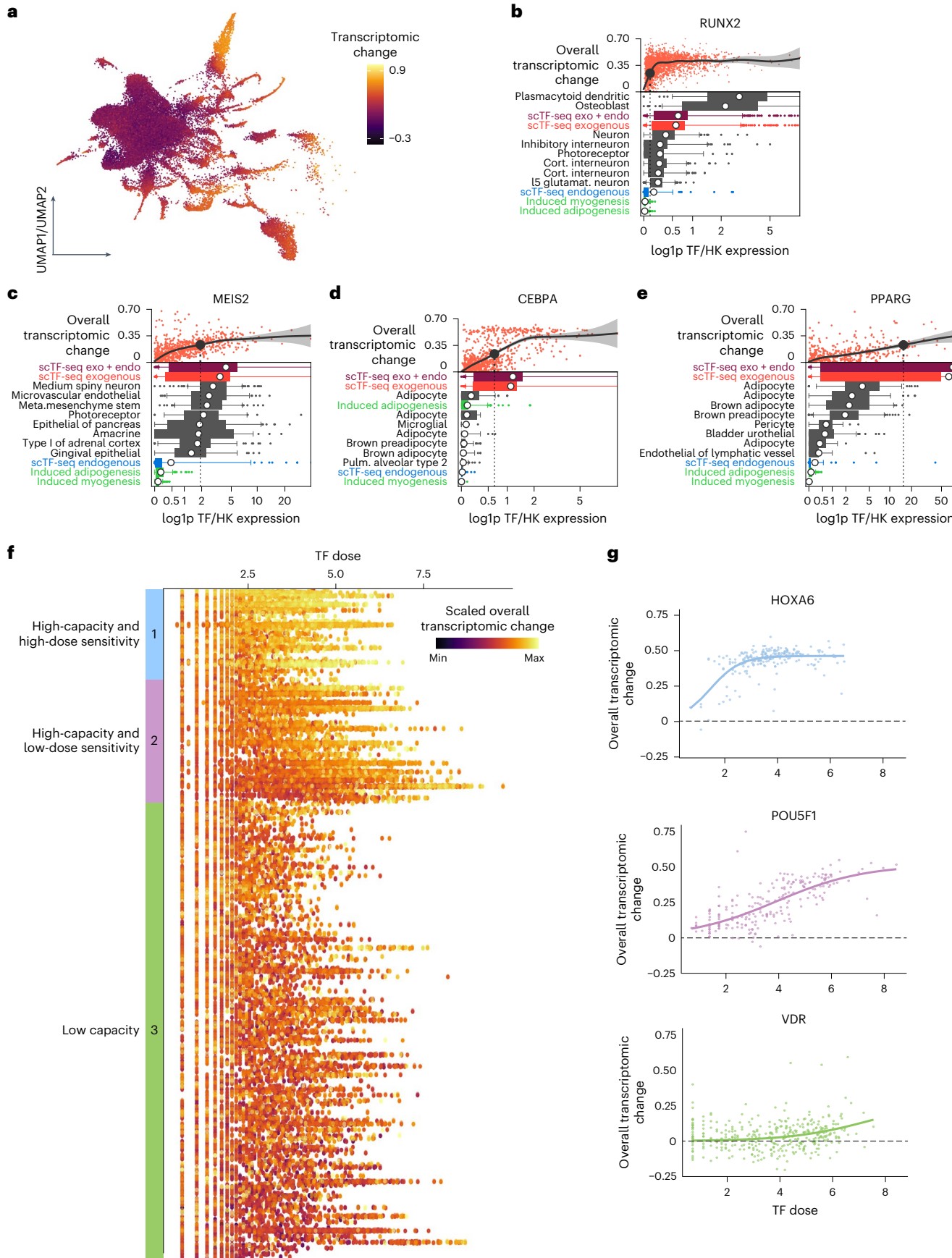

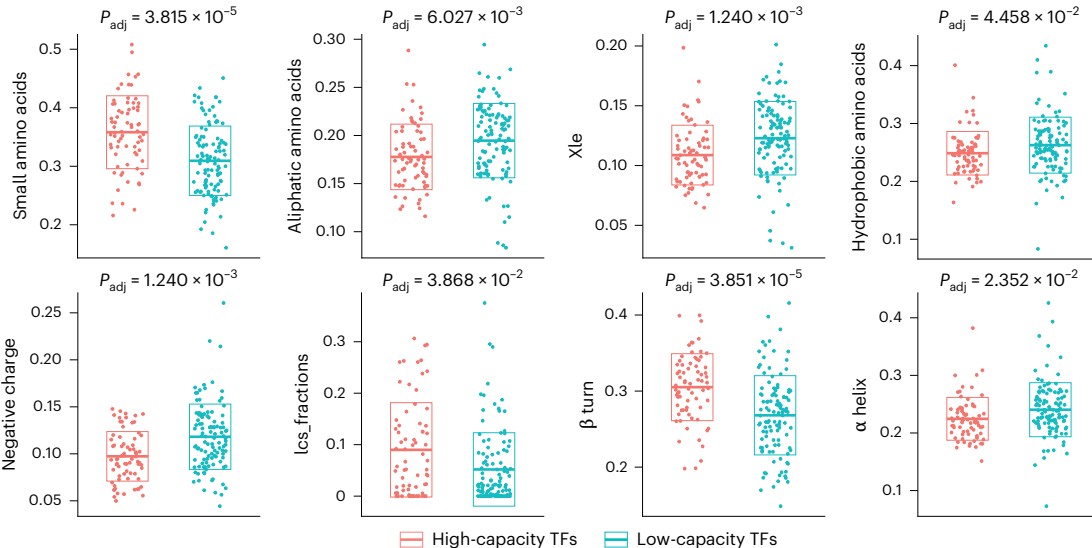

**Fig. 4 | Features of high-capacity TFs.** Scatterplots showing the distribution of various TF features (small amino acids, aliphatic amino acids, Xle, hydrophobic amino acids, negative charge, lcs, β turn, α helix) across high-capacity and low-capacity TFs. Crossbars and boxes represent the mean ± s.d. See Supplementary Table 5 and Methods for statistics and exact *P* values (Extended Data Fig. 4).

Small amino acids, fraction of small amino acids; aliphatic amino acids, fraction of aliphatic amino acids; Xle, fraction of leucine and isoleucine; hydrophobic amino acids, fraction of hydrophobic amino acids; negative charge, fraction of negatively charged amino acids; lcs, low sequence complexity scores; β turn, fraction of β turns; α helix, fraction of α helices.

protein feature analysis revealed that high-capacity TFs are enriched for small amino acids like proline and serine, low sequence complexity and β turns that represent energetically favored nucleation points[51], while being depleted in aliphatic amino acids (including leucine and isoleucine), hydrophobic amino acids, negative charge and α helices (Fig. 4, Extended Data Fig. 4d and Supplementary Table 6; Methods). Similar compositional biases have been revealed as evolutionarily conserved patterns associated with phase-separating proteins, including specific TFs and coregulators whose condensate formation ability is thought to have a key role in gene regulation[52–54].

While the wide dose range is a key feature of scTF-seq, some TFs may still not reach high enough doses for accurate capacity assessment. A power analysis revealed that the predicted probability of correctly classifying TFs (at the maximum dose >3.5) as having low capacity was 77% (Extended Data Fig. 4e–g; Methods). This indicates that the dose levels reached in this study are sufficient to accurately resolve the regulatory capacity for most TFs (198 of 234) in MSCs.

**Reprogramming heterogeneity is driven by dose and stochasticity**

TF dose strongly contributes to reprogramming heterogeneity; however, overall transcriptomic changes lack directionality and gene-specific resolution (Fig. 3a). Therefore, we also investigated whether individual genes or gene sets respond consistently or variably to TF dose, thereby facilitating the emergence of different forms of reprogramming heterogeneity. We identified TFs inducing heterogeneous responses by systematic clustering (Extended Data Fig. 5a–c; Methods). Focusing first on lineage-driving TFs, heterogeneous cell states within a single lineage could be explained by monotonic effects of TF dose on early and late differentiation genes. For example, the adipogenic gene expression signature (termed adiposcore hereafter) of CEBPA cells strongly correlated with *Cebpa* dose (Fig. 5a). Early adipogenesis regulator *Cebpd* was down-regulated, whereas the master regulator of adipocyte differentiation *Pparg* and mature adipocyte markers like *Fabp5* and *C3* were upregulated with increasing *Cebpa* doses (Fig. 5b).

Beyond monotonic effects within one lineage, some TFs induced non-monotonic dose–response patterns across gene sets, driving distinct cell fate specifications and thus contributing to a more complex

form of reprogramming heterogeneity. Using KLF4 as an example, three subclusters of KLF4 cells exhibited substantial differences in *Klf4* doses and gene expression patterns (Fig. 5c–f). Low-dose KLF4 cells (cluster 1) uniquely expressed genes related to gene ontology (GO) terms such as ossification, skeletal system morphogenesis and cardiac chamber morphogenesis (Fig. 5f,g). Moderate *Klf4* doses upregulated genes associated with regulation of cellular component size, protein-containing complex assembly and intracellular transport, while high *Klf4* doses induced genes involved in regulating developmental growth, epithelial cell development and face development (Fig. 5f,g). These findings suggest that *Klf4* dose variations direct cells toward different functional states, regulating differentiation, cellular organization and development, respectively. Similar patterns were observed for many other TFs, including RUNX2, ETV1, EGR1, GRHL2 and ESR2, and were reproducible across batches (Extended Data Figs. 5d–f and 6a–h). Using RNAscope, we probed the TF dose (using WPRE, a viral element in the TF-ID-containing mRNA, as a proxy; Methods) and marker genes that are specific to particular KLF4 or ESR2 subpopulations, and cross-validated their dose-dependent expression patterns (Methods). In line with the scTF-seq results, RNAscope quantification accurately captured the mutually exclusive expression of *Glul* and *Postn* in low versus intermediate/high KLF4 cells, as well as the non-monotonic dose responses of *Gng12* and *Aspn* in ESR2 cells (Fig. 5h–j and Extended Data Fig. 6i–k).

While TF dose is a key factor influencing cell fate, we also identified TFs including MEIS2 and MYOG that reproducibly stratified cells into distinct states despite similar TF doses (Fig. 5k–m and Extended Data Figs. 5d and 7a–d). For *Meis2*, intermediate doses generated multiple cell states (Fig. 5k,l, clusters 1–4) with minimal differences in dose distribution and each characterized by the expression of unique gene modules (Fig. 5m and Extended Data Fig. 7e,f). In fact, MEIS2 cells that were conservatively enriched for modules 2 and 3 displayed opposing dose relationships across two batches, thereby obscuring any consistent dose-dependent trend in the aggregate data (Extended Data Fig. 7g–l) and suggesting the emergence of multiple alternative cell states at moderate *Meis2* doses. At higher *Meis2* doses, cells appeared to converge on a more homogeneous cell state (Fig. 5m and Extended Data Fig. 7f). Altogether, these findings indicate that, while TF dose is a

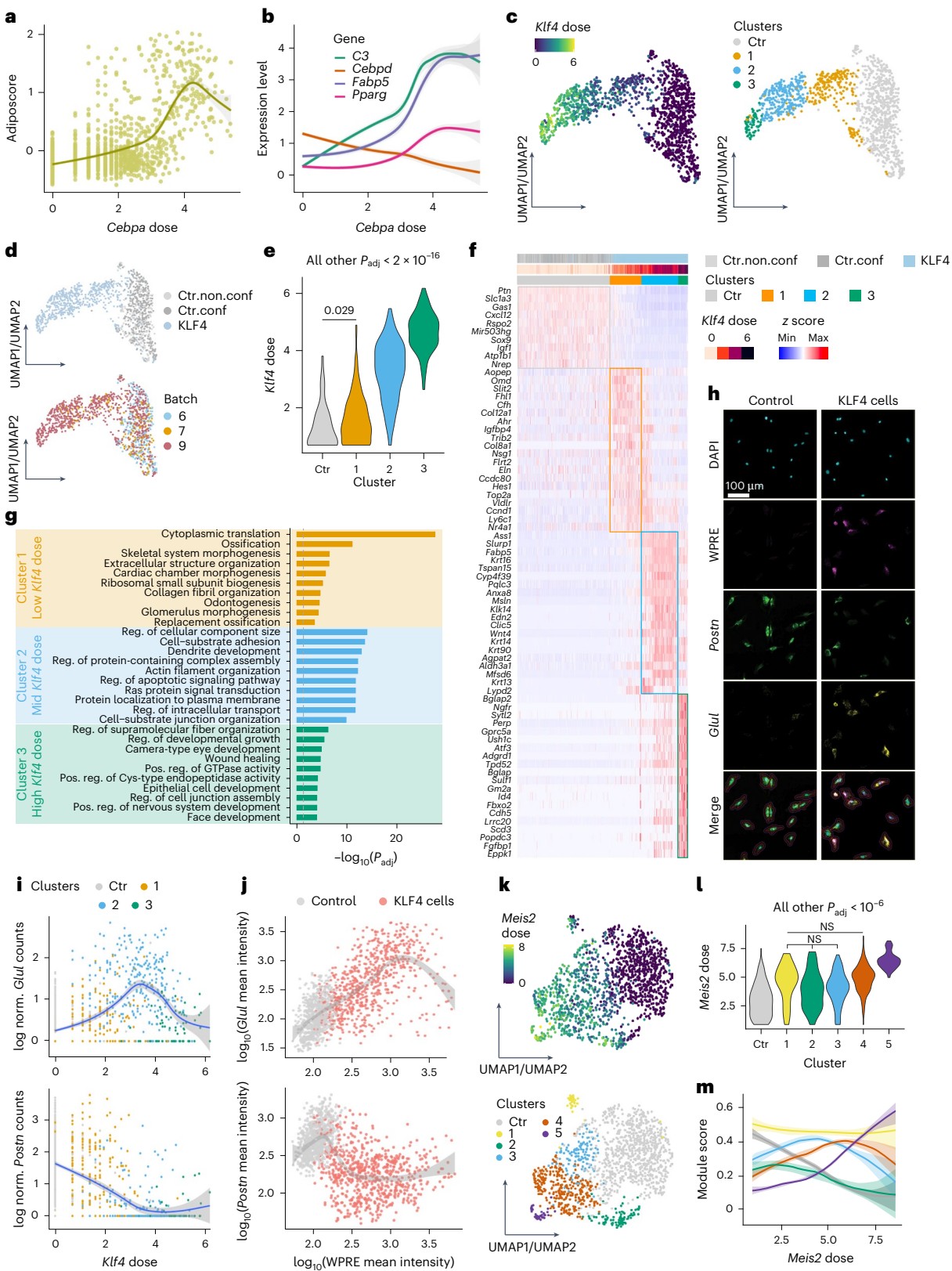

critical determinant of cell fate, additional, possibly stochastic factors likely have important roles in regulating cell fate decisions.

## Dissecting the impact of cell cycle on reprogramming

One factor that also contributes to transcriptomic heterogeneity, alongside TF dose, is the cell cycle, given its fundamental role in stem cell self-renewal and lineage determination[27,55]. To address our limited understanding of how the cell cycle interacts with TFs and their dose, and contributes to reprogramming heterogeneity, we leveraged our scTF-seq data to systematically study TF overexpression and cell cycle dynamics interactions. Cell cycle phase was inferred and adjusted for each cell, and the proportion of cells in each adjusted phase was compared across all TFs (Fig. 6a, Extended Data Fig. 2e and Supplementary Notes 7 and 11). As expected, known cell cycle-driving TFs such as E2F2

**Fig. 5 | Reprogramming heterogeneity induced by TFs. a,b**, Adiposcore (**a**; Methods) and expression level of adipogenesis-related genes (**b**) in CEBPA cells at different doses and batch-paired control cells (dose = 0). **c,d**, KLF4 and batch-paired control cells colored by *Klf4* dose (**c**, left), cluster (**c**, right), category (**d**, top) or batch (**d**, bottom). Fitted model = LOESS. **e**, Dose distribution of KLF4 cells in each cluster shown in **c** (right). **f**, Heatmap displaying log-normalized expression (*z* score scaled by gene) of the top differentially expressed genes of KLF4 clusters (shown in **c** (right) and **d** (top)). Colored outlines indicate marker genes for respective clusters from **e**. **g**, Top ten unique biological process terms identified by GO enrichment analysis on the substantially differentially expressed genes of each KLF4 cluster (shown in **c** (right)). **h**, RNAscope images showing DAPI, WPRE (proxy for TF dose), *Postn* and *Glul* expression in control and KLF4 cells. Representative images of two independent experiments. Red and purple outlines indicate the cell boundary and expanded cell boundary, respectively. Scale bar = 100 µm. **i**, Scatterplot showing the expression of

*Glul* (top) or *Postn* (bottom) in KLF4 cells (colored by the clusters shown in **c** (right)) and batch-paired control cells in function of *Klf4* dose. **j**, Single-cell quantification of RNAscope (as shown in **h**) showing the log-normalized mean fluorescence intensity of WPRE (proxy for TF dose) versus *Glul* or *Postn* in KLF4 and control cells. Fitted model = GAM. **k**, UMAP plots of MEIS2 and batch-paired control cells colored by *Meis2* dose (top) or cluster (bottom). **l**, Violin plot showing the dose distribution of MEIS2 cells in each cluster shown in **k**. **m**, Dose–response curves for the scores of five distinct gene expression modules regulated by MEIS2. Each module represents the substantially differentially expressed genes from the individual MEIS2 clusters in **k**. The same color scheme used for the clusters in **k** is applied to the corresponding modules here. Fitted model = GAM. See Supplementary Table 5 and Methods for statistics and exact *P* values (Extended Data Figs. 5–7). Ctr, clusters containing fewer than 60% TF cells; GAM, generalized additive model; norm., normalized; pos. reg., positive regulation.

(ref. [56]), T[57] and MYCN[58] substantially increased the proportion of S and G2/M cells (Fig. 6b). Beyond discrete phase classification, which overlooks the circular and continuous nature of the cell cycle, we examined the density distributions of cell cycle scores. One-dimensional distributions revealed that E2F2 overexpression primarily shifted cells toward high S scores, while T and MYCN increased both S and G2/M scores (Fig. 6c). Two-dimensional density estimation further clarified that E2F2 may not only drive entry into the S phase but also block cells from progressing to G2/M (Fig. 6c and Extended Data Fig. 8a–d). This aligned with previous findings showing that stabilized E2F2 activity throughout the cell cycle accelerates G1/S transition in the short term but initiates replication stress, DNA damage and apoptosis, thereby impairing long-term cell fitness[59].

Interestingly, the proportion of S and G2/M cells generally increased with rising *T* and *E2f2* doses (Fig. 6d). However, TFs such as MYCN, RUNX2 and PAX9 exhibited a non-monotonic relation between dose and cell cycle, with the largest fraction of S and G2/M cells observed at intermediate doses (Fig. 6d and Extended Data Fig. 8e). This prompted us to explore how TFs dose-dependently coordinate cell cycle dynamics and lineage differentiation, revealing, for example, for adipogenesis that cell proliferation and the adiposcore were mutually exclusive in CEBPA or PPARG cells (Fig. 6e). This aligns with the established notion that lineage differentiation, including adipogenesis, requires cell cycle exit[27,55,60]. Indeed, *p21*, encoding a cyclin-dependent kinase inhibitor critical for harmonizing cell cycle exit and adipocyte differentiation[60], was upregulated at high *Cebpa* or *Pparg* doses (Fig. 6e). In contrast, cell cycle exit and cell differentiation were decoupled in high *Mycn* cells, as evidenced by the concurrent high adiposcore and *p21* expression in S and G2/M, and the observed accumulation of lipid droplets alongside increasing nuclei counts (Figs. 2e and 6e and Extended Data Fig. 3f). However, this aberrant differentiation under high *Mycn* doses was accompanied by evident cell death (Fig. 6f). These findings collectively underscore the intricate interplay among TFs, TF dose, cell cycle dynamics and lineage differentiation.

### Dose influences TF combination synergy or antagony
TFs do not operate in isolation and their effects depend on the relative dose[61,62]. Yet, how one TF's dose influences the effects of another TF is

poorly understood due to the complexity underlying combinatorial analysis. To explore this, we selected TFs with strong lineage-driving potential, including CEBPA, PPARG and MYCN for adipogenesis, MYOG for myogenesis and RUNX2 for osteogenesis, and performed combinatorial scTF-seq experiments (Fig. 7a; Methods).

Using single-cell readouts, we evaluated whether TF pairs induced distinct cell states compared to those induced by either TF alone (Methods). Typically, one TF dominated the transcriptomic outcome, forming a directed network of TF dominance (Fig. 7b,c). Yet, pairs such as CEBPA + MYCN, MYCN+MYOG and MYCN + RUNX2 produced unique states not explainable as simple combinations of individual TF effects, marked by distinct gene expression profiles (Fig. 7b and Extended Data Fig. 9a). For instance, CEBPA + MYCN uniquely upregulated adipogenesis-related genes (*Fabp4* and *Gpd1l*), suggesting a synergistic interaction (Extended Data Fig. 9a). Interestingly, adipogenic TFs paired with either adipogenic or lineage-diverting partners had synergistic or antagonistic effects, respectively, on adipogenic capacity (Extended Data Fig. 9b). These findings were substantiated by the respectively higher or lower lipid score for MYCN + CEBPA or MYOG + CEBPA compared to CEBPA cells (Extended Data Fig. 9c,d).

We then investigated how TF dose shapes combinatorial effects. For overall cell identity, we found that any TF with much greater doses than another was able to overcome the dominant effect, except for PPARG, possibly due to its low dose sensitivity (Fig. 7d, Extended Data Fig. 9e and Supplementary Table 4). Unique combinatorial states often required high doses (Fig. 7d). Additionally, TF dose sensitivity could shift in competitive contexts. For example, MYOG was highly dose sensitive alone, whereas it was mostly dominated by other less dose-sensitive TFs at low doses (Fig. 7d, Extended Data Fig. 9e and Supplementary Table 4). MYCN, despite lower dose sensitivity than CEBPA, dominated over CEBPA when they were at similar doses (Fig. 7d, Extended Data Fig. 9e and Supplementary Table 4). TF combinations also exhibited dose-dependent effects on adipogenic capacity, with some interactions being non-monotonic (Fig. 7e). For instance, CEBPA + MYCN synergized globally, yet MYCN at intermediate levels antagonized adipogenesis in high *Cebpa* cells (Fig. 7e and Extended Data Fig. 9b). Conversely, the highest adipogenic capacity of the CEBPA + PPARG combination was observed at a low *Pparg* dose, a surprising finding given PPARG's role as

**Fig. 6 | Interactions between TFs, the cell cycle and differentiation (adipogenesis). a**, UMAP plot of the TF atlas colored by adjusted cell cycle phase (Supplementary Note 7). **b**, Bar plot showing the fraction of cells in the adjusted phase for each TF. The total number of cells is indicated in brackets. A Fisher's exact test was performed between confluent control cells (Ctr.conf) and each TF. In addition to Ctr.conf, only TFs and the non-confluent control cells (Ctr.non.conf) that tested significantly (FDR-adjusted *P* < 0.05) are visualized here. The top three TFs and controls are highlighted in red. **c**, Density plots showing the distributions of S and G2/M scores of TF cells (T, E2F2 or MYCN in red) compared to confluent control cells (Ctr.conf in teal). **d**, Bar plots

showing the fraction of cells in each adjusted cell cycle phase across binned doses of *T*, *E2f2* or *Mycn*. **e**, Heatmaps showing the transcriptomic adiposcore and the mean expression level of *p21* in CEBPA, PPARG and MYCN cells, which are binned according to their adjusted cell cycle phase and TF dose. Bins with less than three cells were excluded (white square). **f**, Fluorescence images showing the viability of control, CEBPA and MYCN cells, indicated by PI staining in red (Supplementary Note 12). Nuclei were stained with Hoechst in blue. Representative images of two independent experiments. Scale bar = 200 µm. See Supplementary Table 5 and Methods for statistics and exact *P* values (Extended Data Fig. 8). PI, propidium iodide.

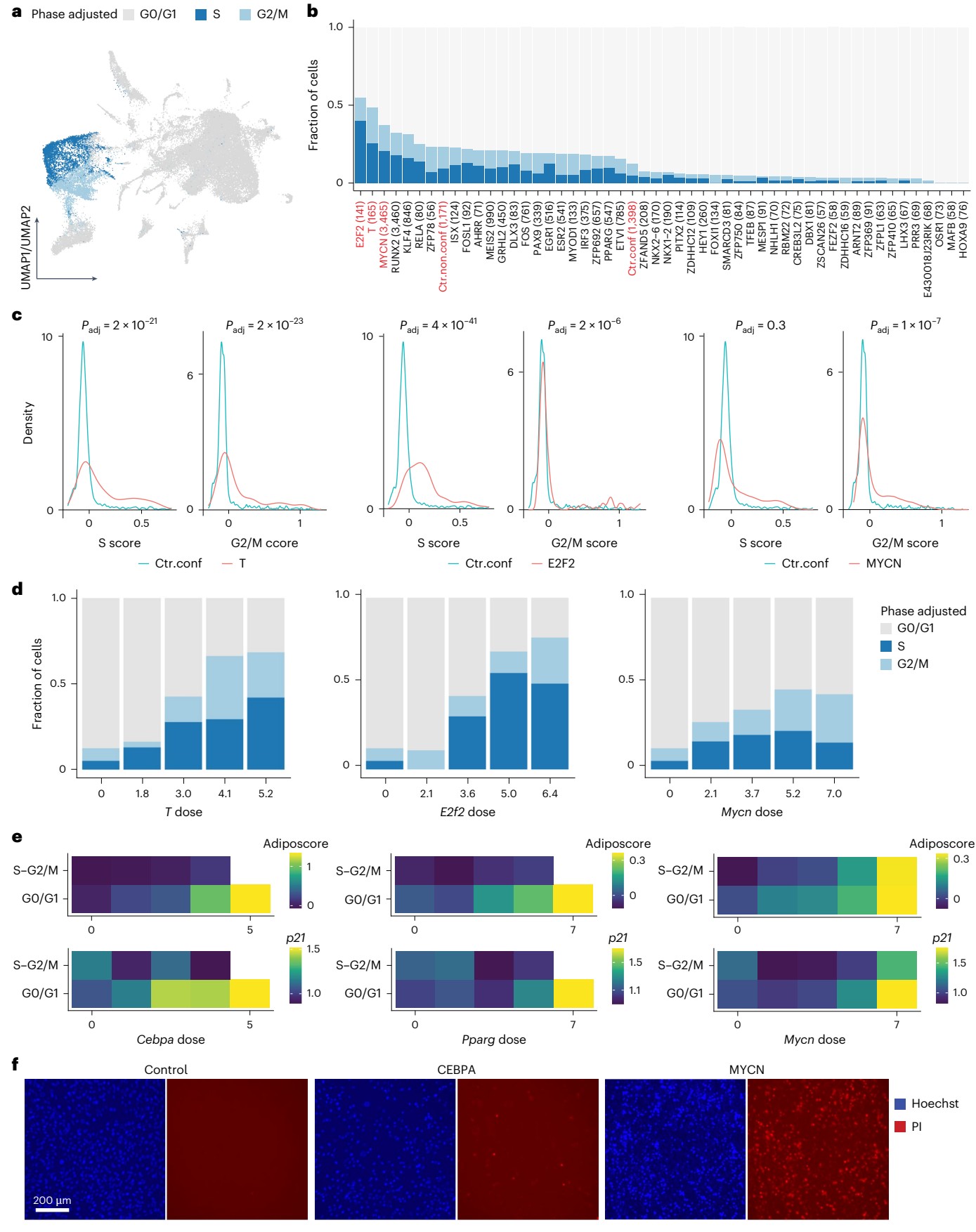

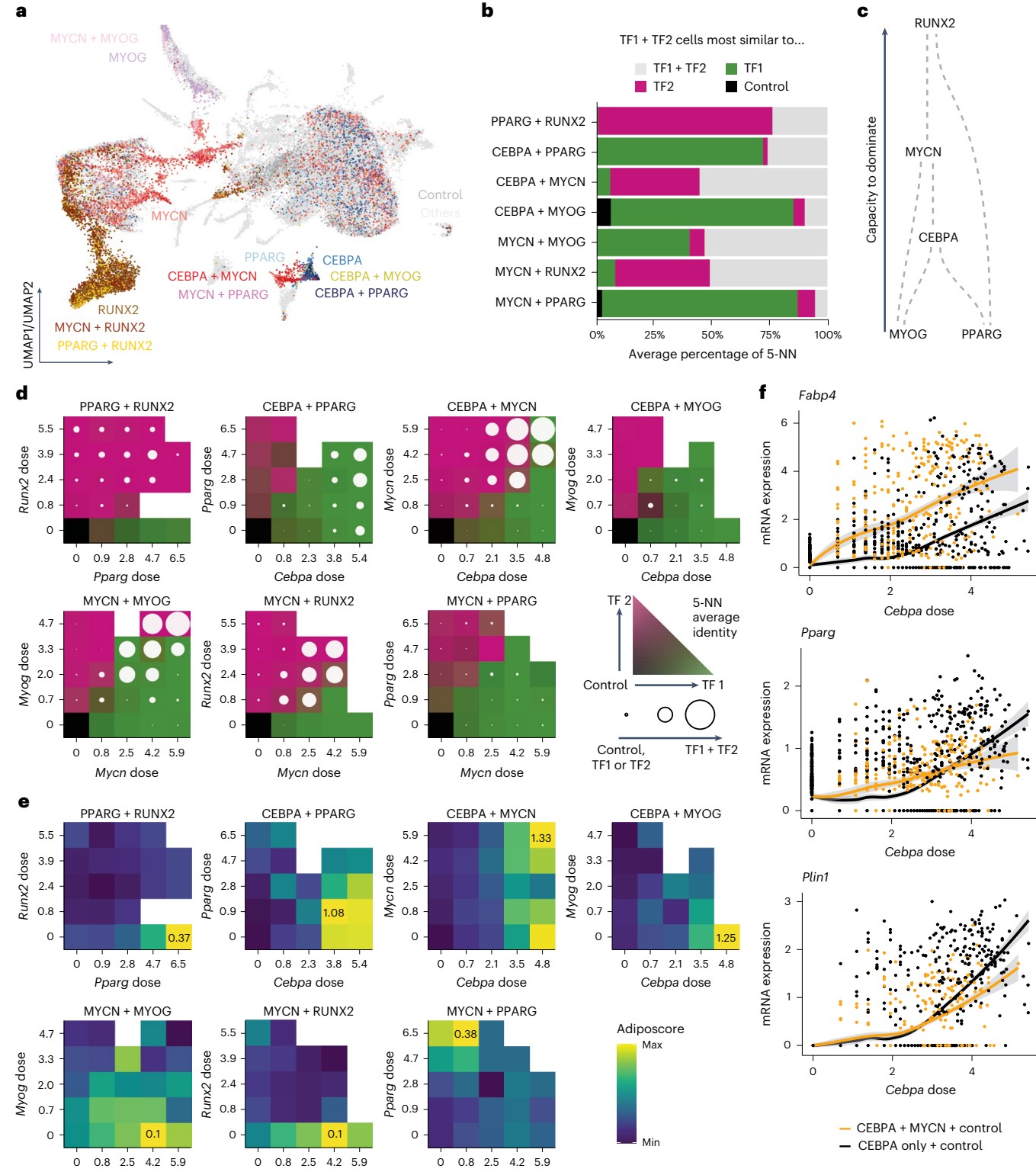

**Fig. 7 | Dose-dependent effects of interactions between TFs. a**, UMAP plot showing the positions of single and combinations of TFs with respect to all other tested TFs and control cells. **b**, Percentage of five nearest neighbors to which each combinatorial TF (TF1 + TF2) cell is closest. **c**, Schematic representation of the dominance of each TF inferred from **b**. For example, RUNX2 dominates other TFs because combining RUNX2 with another TF results in transcriptomes that closely resemble those of RUNX2-only cells. **d**, Percentage of five nearest neighbors to which each combination cell is closest within a pair of dose bins. Bins were determined by uniformly splitting the interval between 0 and the maximum dosage, with an additional bin at a dose of 0. The color scale represents the percentage of cells closest to TF1, TF2 or the control cells, respectively, represented using a bilinear interpolation between green, pink and gray. The circle represents the percentage of cells closest to the TF1 + TF2 cells, with a full circle meaning that all cells were closest to TF1 + TF2. **e**, Transcriptomic adiposcore between different dose bins. The color scale ranges from the maximal to the minimal adiposcore for each combination of TFs. The value of the maximal adiposcore is indicated in the corresponding bin for each combination. **f**, Dose–response curves for control cells with only CEBPA cells (black) or with MYCN + CEBPA cells (orange). Fitted model = LOESS (Extended Data Fig. 9). NN, nearest neighbors.

a master regulator of adipogenesis (Fig. 7e)[45]. Finally, we observed that dose-dependent synergism can be gene-specific. CEBPA + MYCN synergistically activated adipocyte markers (*Fabp4*, *Adipoq*), while other adipogenesis-related genes (*Pparg*, *Plin1*) switched between synergy and antagonism depending on *Cebpa* dose (Fig. 7f and Extended Data Fig. 9f). These nonlinear and gene-specific interactions reflect the complexity of regulatory architectures, necessitating TF dose-resolved approaches for deeper insight.

## Discussion

Numerous studies have highlighted the transformative impact of TF dose on molecular and cellular states[9,13,15,63–66]. However, the substantial cellular heterogeneity observed upon TF overexpression in ex vivo experiments contrasts with the precise control of cell fate alterations in vivo. This discrepancy highlights a gap in our understanding of how cellular programs intricately respond to variations in TF dose. To address this, we developed scTF-seq, a scalable approach that enables the following: (1) identifying lineage regulators and functional modules, rendering the resulting TF atlas a comprehensive reference for discovering TFs that induce specific phenotypes of interest (Fig. 2) and (2) leveraging a broad range of TF doses across thousands of cells (Fig. 1) to systematically, quantitatively and reproducibly map the influence of TF dose on cell reprogramming at the single-cell level. This unique capability distinguishes scTF-seq from other large-scale single-cell[16,18–20,67] or bulk[3,68,69] TF screening strategies (Figs. 3–7).

By exploring this intricate relationship between TF dose and function, we were able to stratify TFs into the following three distinct categories: low versus high-capacity TFs with the latter further subdivided into 'low' or 'high' dose-sensitive groups (Fig. 3). Although the biological meaning of this TF classification is not yet fully clear, high-capacity TFs show greater loss-of-function intolerance and are enriched for phase-separation-related features (Fig. 4), pointing to a potential connection between TF capacity, regulatory impact and condensate formation[48–54]. A TF's dose sensitivity may also be highly relevant to how TFs exert their function in response to stimuli or developmental signals. For example, most HOX and CDX TF family members feature a high-capacity and high dose sensitivity, aligned with their known influence in development through a concentration gradient[70,71]. In contrast, POU5F1 is a high-capacity, but low-dose-sensitive TF, consistent with observations that the highest reprogramming efficiencies were reached at the highest *Pou5f1* overexpression levels[65,72,73]. Many TFs appeared to have low capacity, exemplified by vitamin D3 receptor VDR, which is likely ineffective without a sufficient supply of its ligand. We thus cannot rule out that certain TFs might have different classifications depending on factors such as the probed system, stimuli or even used approach. Furthermore, the definition of low-capacity TFs may also be influenced by the maximum dose achieved (Fig. 3).

Within high-capacity TFs, our findings illuminate the crucial role of TF dose in modulating cell states and driving reprogramming heterogeneity (Fig. 5). However, because our data are from a single snapshot, it remains difficult to infer the exact trajectory, that is, whether the observed TF-driven nonmonotonic expression patterns reflect true cell fate branching or, alternatively, progressive state transitions[74]. Future time-resolved studies will be essential to disentangle this complex relationship. Moreover, not all observed cell state transitions were strictly dose-dependent (Fig. 5). This may reflect the stochastic nature of gene transcription, arising from the dynamic interplay among transcriptional processes (such as TF–DNA binding kinetics), epigenetic modifications and post-transcriptional events in individual cells[75–79].

Alternatively, dose-independent cell state transitions may be influenced by more deterministic factors such as the cell cycle phase during initial TF overexpression[12], although our observations indicate that the influence of the cell cycle can extend beyond the starting cell population (Fig. 6). Several TFs, including master regulators RUNX2 and PAX9, exhibit a complex, non-monotonic interplay between the cell cycle and TF dose. This implies that such TFs can function as rheostats, regulating dose-dependent entry into the cell cycle to control terminal differentiation, consistent with previous observations for the TF MITF[80]. We also revealed that MYCN challenges the conventional requirement for cell cycle exit in terminal differentiation, displaying a unique dynamic where cells actively cycled while concurrently expressing adipogenic genes (Fig. 6). Unraveling how MYCN regulates this intriguing state will necessitate more investigations, but it reflects MYCN's pleiotropic role in controlling multiple cellular processes underlying organogenesis[58].

Furthermore, our study underscores the non-monotonic, gene-specific dose dependency of TF interactions (Fig. 7), possibly indicating diverse roles of implicated TFs in mediating various aspects of gene regulation, such as controlling chromatin accessibility, regulatory element interactions and gene activation[15,81]. The observed complexity in TF interactions points to the critical challenge of determining optimal dose regimes for sets of TFs required to generate specific cell states.

In summary, our study not only sheds light on the pivotal role of TF dose in cellular reprogramming but also opens avenues for further exploration. scTF-seq's agnostic nature to the cell system or species, coupled with its potential to uncover regulatory TF properties, positions it as a valuable tool for future research. However, certain limitations of the current study should also be acknowledged such as the lack of temporal resolution, emphasizing the need for investigating reprogramming over time. In addition, future iterations of the analysis should consider incorporating additional modalities, such as chromatin accessibility, to unravel molecular mechanisms underlying TF dose effects. This integrative approach would hold promise for deepening our understanding of TF-mediated changes in the chromatin landscape and their implications for cellular reprogramming.

## Online content

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

## Methods

The experiments conducted in this study did not require approval from a specific ethics board.

### Key resources table

Primer sequences, key resources (like cell lines, bacteria strains, reagents, compounds and commercial assays) and their sources and identifiers, and software versions can be found in Supplementary Table 7.

### Experimental model and subject details

HEK293T and C3H10T1/2 cells were used in this study. Detailed protocols for cell culture and differentiation, lentivirus production and transduction were described in Supplementary Notes 1–3.

### Experimental details

**Barcoding and cloning of TF ORF libraries.** Barcoded dox-inducible lentiviral expression vectors carrying TF ORFs (pEXPRESS) were generated individually using the Gateway cloning system in two steps. In the first step, barcoded destination vectors were generated by introducing random nucleotides to the upstream region of the 3′ LTR of pSIN-TRE-GW-3xHA-puroR vector. Two fragments were amplified from the pSIN-TRE-GW-3xHA-puroR vector using Kapa HiFi ready mix with 0.3 μM Enrich_F3 and 0.3 μM pTREP-BC-RamR, 0.3 μM pTREP-vec-R and 0.3 μM pTREP-BC-RamF, respectively. The PCR program was as follows: (1) 98 °C for 3 min, (2) 98 °C for 30 s, (3) 63 °C for 30 s, (4) 72 °C for 5 min, repeat steps 2–4 for 15 cycles and (5) 72 °C for 5 min. After purifying both PCR products using a 1% agarose gel and a gel purification kit, the two fragments were assembled using a Gibson assembly mix according to the manufacturer's instructions. Assembled plasmids (termed pTREP-ID vector hereafter) were then purified using a DNA Clean and Concentrator purification kit and transformed into one-shot ccdB survival 2 T1R resistant competent cells. Successful colonies were then inoculated to growth medium containing Ampicillin and Chloramphenicol for miniprep and validation. In the second step, TF ORFs were transferred from generated entry clones[82] to pTREP-ID vectors using LR Clonase II enzyme mix, producing pEXPRESS plasmids. Stbl3 one-shot competent cells were then transformed with pEXPRESS and grown on ampicillin (100 μg ml⁻¹) plates overnight. Colonies were picked and transferred to Luria-Bertani with ampicillin for miniprep or midiprep. The barcodes (termed TF-IDs hereafter) and TF ORF on the pEXPRESS were examined by Sanger sequencing with the usage of microsynth standard primers: EGFP-C-Rev and TET-CMV-for.

**Single TF overexpression screening, 10x scRNA-seq sequencing and TF-ID enrichment.** Only TF-IDs with a hamming distance greater than 2 nucleotides were retained within each experiment for demultiplexing. In addition, C3H10T1/2 cells were transduced with the lentivirus particles carrying each barcoded TF ORF expression vector individually. Puromycin selection was performed to enrich successfully transduced cells. TF expression was induced by dox (2 μg ml⁻¹) treatment during 5 days in cells placed in a basic culture medium refreshed every 48 h. Then, cells were collected (Supplementary Note 1), pooled and loaded in the 10x Genomics Chromium Controller targeting 8,000–10,000 cells per experiment. Because C3H10T1/2 cells might undergo spontaneous differentiation once reaching 100% confluency, mCherry was overexpressed under the same conditions in both non-confluent and confluent C3H10T1/2 cells as a control. Unless specified, all control cells were considered in subsequent analyses by default. To ensure reproducibility, negative controls (mCherry-overexpressing) and positive controls (cells induced for differentiation using an adipogenic cocktail or cells overexpressing TFs showing known reprogramming effects) were included in every experiment. At least six TFs were shared in each experiment with other experiments as biological replicates (Supplementary Table 1). All scRNA-seq experiments were performed using Chromium Single Cell Expression 3′ Reagent Kits after the

manufacturer's instructions. To specifically enrich the TF-ID, an additional PCR amplification targeting the 10x barcode, UMI and TF-ID were conducted using the full-length cDNA product of the 10x scRNA-seq library. The cDNA library (6 ng), BC_vec_target_10X_F1 vector-specific forward primer (0.3 μM), Truseq_universal_adaptor (0.3 μM) and Kapa HiFi ready mix (1×) were used after the program—(1) 98 °C for 30 s, (2) ten cycles of 98 °C for 10 s, 63 °C for 20 s and 72 °C for 30 s and (3) 72 °C for 5 min. The resulting amplicons were then purified using Ampure beads and further amplified to generate TF-ID-enriched libraries compatible with 10x cDNA libraries with Truseq_D7_adapter (0.3 μM), Truseq_universal_adapter (0.3 μM) and Kapa HiFi ready mix (1×) after the program—(1) 98 °C for 30 s, (2) four cycles of 98 °C for 10 s, 63 °C for 20 s and 72 °C for 30 s and (3) 72 °C for 5 min. The TF-ID-enriched libraries were then purified twice using 0.6× Ampure beads and pooled with the regular 10x sequencing libraries, which were sequenced together on the Illumina NextSeq 500/Hiseq 4000/NovaSeq 6000 platform using the dual-index configuration after manufacturer's instructions to obtain a mean depth of 50,000 reads per cell.

**Constructing adipogenic and myogenic reference cells.** For in vitro adipogenic differentiation, mCherry-overexpressing cells were first cultured in the basic culture medium supplemented with 100 ng ml⁻¹ BMP4 for 3 days. Then the induction medium was added for 2 days, which was composed of the basic culture medium and MDI cocktail containing 1 μM dexamethasone, 0.5 mM 3-isobutyl-1-methylxanthine and 167 nM insulin. The cells were maintained in the basic culture medium supplemented with 167 nM insulin until collection. Myogenic reference cells were generated by transducing *Myog* (encoding a key myogenesis regulator[40]) and inducing its overexpression in C3H10T1/2 cells for up to 5 days (Supplementary Note 3).

**TF pair screening.** To generate data with combinations of TFs, C3H10T1/2 cells were transduced with the first TF and selected with puromycin (Supplementary Note 3). Thereafter, the selected cells were transfected with a second TF (virus MOI around 3). The overexpression of both TFs was induced by dox following the conditions described in the above single TF overexpression screening section.

**Multiplex RNA in situ hybridization.** TF or mCherry-overexpressing cells were prepared through transduction, puromycin selection and 2 μg ml⁻¹ dox induction as described in Supplementary Note 3 and the above single TF overexpression screening section. TF-overexpressing cells, with wild-type C3H10T1/2 and mCherry-noDox controls (transduced with mCherry but lacking dox treatment), were seeded onto individual wells of 96-well plates at ~10% density. Multiplex RNA in situ hybridization was performed on 96-well plates using RNAscope technology (Advanced Cell Diagnostics)[83] per the manufacturer's instructions. Briefly, cells were fixed with 10% neutral buffered formalin for 30 min at reverse transcription, washed with PBS, dehydrated with 50%, 75% and 100% ethanol for 1 min each, and stored at −20 °C. RNAscope was performed within the next 2 days. In situ probes against mouse *Glul*, *Postn*, *Gng12*, *Aspn* and lentiviral element WPRE were used in combination with the RNAscope Multiplex Fluorescent Reagent Kit v2 for target detection.

**Validation of the adipogenic capacity of single TFs or TF pairs.** C3H10T1/2 cells were transduced with the barcoded TF ORF expression vector with mCherry (control), individual adipogenic TFs or TF pairs, followed by Puromycin selection and 5 days of dox-induced TF overexpression (Supplementary Note 3 and the above single TF overexpression screening and TF pair screening sections). Cells were then fixed with 4% PFA for 15 min at room temperature, permeabilized with PBS and Triton and stained with fluorescence dyes—Bodipy 10 μg ml⁻¹ for lipids and DAPI for nuclei. Cells were incubated with dyes in PBS for 30 min in the dark, washed twice with PBS and imaged. Image stacks (10

per well, 96-well plate) were collected for each replicate using the blue and green channels with a ×20/0.8 objective. Adipocyte differentiation was quantified using an image preprocessing and analysis algorithm per the developer's instruction[84]. The lipid score was defined as the ratio of Bodipy signals to DAPI signals.

## Quantification and statistical analysis

**10x scRNA-seq data preprocessing and quality control.** Basecalls were performed using bcl2fastq. Sequencing reads were aligned and quantified using Cell Ranger against the GRCm38 (mm10, Ensembl release 96) mouse reference genome with default settings to generate count matrices of genes × cell barcodes. To match TF-IDs to cells, reads were also mapped to the pEXPRESS vector sequence, where each TF-ID nucleotide was replaced by 'N'. TF-IDs from aligned reads at the location of 'Ns' were extracted with 1 nt mismatch allowed and matched to the corresponding cell barcodes and UMIs using an in-house framework, TFseqTools (https://github.com/DeplanckeLab/TFseqTools), yielding TF-IDs × cell barcodes read/UMI matrices.

All the data were loaded and processed on R. Doublet removal and TF-ID assignments were performed per experiment as described in Supplementary Note 5. Remaining cells from each experiment were analyzed using Seurat[85]. TF dose was computed as ln(1 + UMIs of the assigned TF-ID). Low-quality cells were filtered out using the isOutlier function of package scran[86], using an nmads cutoff of 4–6 of the lower end tail depending on the gene expression matrix of individual experiments. Cells with >10% or 15% of mitochondrial gene expression, >40% or 60% of ribosomal RNAs and <75% of protein-coding genes were also filtered out. TFs having <8 cells were excluded. Batch correction and data integration were performed as described in Supplementary Note 6. The clustree function from the clustree package was applied to find an optimal resolution for clustering[87]. The exact resolution of clustering was specified in downstream analyses. Cells and clusters were visualized using uniform manifold approximation and projection.

**Dose comparison with public data.** To functionally compare the dose reached by scTF-seq with that of alternative datasets, we obtained MORF data from GSE216595 (ref. 16). The data were preprocessed using the standard scanpy pipeline, and *Fos*-overexpressing and mCherry-overexpressing cells were subsetted as provided by the original authors. Differential expression for both scTF-seq and MORF data was calculated using scanpy's rank_gene_groups function with default parameters (method = *t* test, correction = Benjamini–Hochberg). Common differentially expressed genes were selected by selecting those orthologs that were differential (scanpy's score of >5). To compare effect sizes at various doses, scTF-seq *Fos*-overexpressing cells were subsetted by removing cells with a dose higher than a certain cutoff.

**RNAscope quantification.** Images of 25 fields, each with five Z stacks, were collected for four fluorescence channels (blue, green, red and infrared) per well (96-well plate) using a ×20/0.8 objective. After flat-field correction, the best focus among the five Z stacks was selected for each field and channel. All fields with the best focus were further fused and represented as pyramidal images. Cell segmentation was conducted using the cytoplasm model (cyto3) of Cellpose3 (ref. 88). The segmentation channel was generated by summing the green, red and infrared channels, while DAPI was used as the nuclear channel. A median cell diameter of 50 μm was specified, and a 10 μm nuclear expansion was applied to capture signals near cell boundaries. Mean fluorescence intensity was measured for each segmented cell and corrected for background by subtracting the median intensity of noncellular regions. Segmentations erroneously assigned to debris or dirt were excluded based on their detected features, such as the small cell size, abnormally low DAPI in the segmented cell region, or high intensity of DAPI in the expanded cytoplasmic region. Cell clumps were excluded based on a low ratio of expanded area to the total cell

area. Additionally, 1% of outliers at the extreme lower or upper tails of the mean intensity distribution for each individual channel were filtered out. Spillover between spectrally adjacent channels was modeled using linear regression on control (mCherry-noDox or wild-type C3H10T1/2) cells and corrected for TF and wild-type control cells when the estimated slope exceeded 0.01.

**Differential expression and enrichment analyses.** Differential expression analysis was performed on all detected genes using generalized linear models with batch as a covariate, as implemented in edgeR[89]. A false discovery rate (FDR) cutoff 5% was used to select substantially differentially expressed genes. 'is.TFoe' in Supplementary Table 3 indicates whether the differentially expressed genes in TF-overexpressing or reference cells are the endogenous counterparts of overexpressed TFs. Marker genes of cell types of interest and hallmark gene sets were downloaded from MSigDB[90] and PanglaoDB[91]. A clustering resolution of 0.2 was used for enrichment analysis of clusters on hallmark gene sets. A customized gene set containing more mature adipocyte markers[92] (Supplementary Note 13) was used to compute the adipocyte module score (referred to as the adiposcore) by using the AddModuleScore function from Seurat. Gene set enrichment analysis was performed using the package fgsea[93]. Due to the relatively small number of gene sets (adipocytes, chondrocytes, myoblasts and osteoblasts) being analyzed, an FDR cutoff 5% was used. GO enrichment analysis was performed using the enrichGO function from clusterProfiler[94]. GO terms with more than 50% genes overlapping were excluded.

**Cellular similarity analysis.** To assess cellular similarities and identify TFs that have similar biological functions (referred to as functional modules), pairwise Pearson correlation coefficients were computed using the rcorr function of the Hmisc package[95], in a PCA space that was constructed from the first 50 PCs, and was inclusive of both control and functional TF cells in G1 (default phase).

**Calculation of overall transcriptomic change.** To quantify the overall transcriptomic change of the TF cells relative to the control cells, the heterogeneity among control cells was regressed out per batch by projecting cells to a PCA space derived from control cells, before integrating all G1 (adjusted phase) cells of TFs and controls from all batches. Subsequently, a negative Pearson correlation between each cell and the centroid of control cells was computed in a unified high-dimensional space, derived from the top 200 PCs of the integrated data. The resultant values were then adjusted by subtracting the mean of the negative correlation between control cells and their centroid.

**Comparison of endogenous and exogenous TF expression.** The physiological (in vivo) expression range of TFs was extracted for all cell types in the CELLxGENE census database, covering over 150 annotated single-cell datasets[96]. The physiological, exogenous (TF dose in the scTF-seq data) and endogenous (endogenous TF expression in the scTF-seq data) TF expression levels were normalized against 13 housekeeping genes (Supplementary Note 14) covering a variety of central cellular processes or systems such as cytoskeleton, translation and ubiquitination. A minimal functional TF dose was defined for each TF as the exogenous dose required to reach a strong transcriptional effect (>0.23 overall transcriptomic change). The minimal functional dose of a TF was then compared to the range (5–95% quantile) of physiological doses found in the cell type with the highest expression of the respective TF.

**TF class and feature enrichment.** TF classes were annotated according to AnimalTFDB[97]. Fisher's exact test was applied to compare the number of zinc-finger or homeodomain TFs across high- and low-capacity TFs. TF features, including amino acid content, low complexity score and β turn fraction as listed in Supplementary Table 6, were

calculated by using the phase separation analysis and prediction classifier[52]. A two-sided Wilcoxon Rank Sum test followed by FDR correction was applied to compare the distribution of TF features between high-capacity TFs and low-capacity TFs. An adjusted $P$ value of <0.05 was considered statistically significant.

**Low-capacity power analysis.** To determine the power to correctly identify a TF as high capacity, we simulated lower maximal doses for all 76 high-capacity TFs. In particular, for each TF, we removed cells above a certain dose threshold, reran the aforementioned logistic modeling and determined whether the TF was still (correctly) classified as being high capacity. By performing this analysis at different thresholds ranging from the TF's max dose to a dose of 2, we quantified the percentage of TFs falsely classified as low capacity at this threshold. This percentage was then used to calculate for each low-capacity TF its probability of being falsely classified as low capacity given its observed max dose (Supplementary Table 4).

**Cell state transition analysis.** To identify TF cells that underwent specialized cell state transitions, clustering analysis was performed on control and functional TF cells in G1 (adjusted phase) using a resolution of 1.2 with the FindCluster function from Seurat. Clusters that were predominantly composed of control cells were classified as control clusters. The remaining clusters were annotated as functional clusters. A TF with a certain proportion of its cells (5–95% cells for that TF) in at least two functionally distinct clusters was deemed to be a candidate steering cell state transition. TFs represented by fewer than 30 cells in total were excluded from analysis. To track cellular state divergence, for each remaining TF candidate, cells in G1 (adjusted phase) were pooled with their batch-paired control cells and reclustered. Control clusters were defined as those in which fewer than 60% cells originated from the focal TF. Monocle3 (refs. 98,99) was used to infer trajectory and pseudotime using the control clusters as roots.

**Analyses for TF pair screening.** Cells overexpressing a pair of TFs were detected as explained in Supplementary Note 15. The following analyses were performed separately for each TF pair. We subsetted cells that were assigned TF1 + TF2, TF1, TF2 or mCherry. Then we assigned each cell to one of the four following groups: TF1 + TF2 (>4 UMIs for both TF1 and TF2), TF1 (>4 UMIs for only TF1), TF2 (>4 UMIs for only TF2) or control (all other cells). To determine whether the TF pair cells grouped together into a state distinct from either the TF1 or the TF2 groups, we identified for each cell its five nearest neighbors in PCA space (first 20 dimensions). We then quantified for each cell within the TF1 + TF2 group the proportion of cells to which it was closest in the other groups, and averaged this over all cells. Cells were binned for both TFs into four uniform bins spanning the range from 0 to the maximum log1p UMI counts, with an additional bin for 0 UMI counts.

To detect genes that were uniquely expressed in TF pair cells, we performed differential expression using Seurat's FindMarkers. Specifically, for each cell within the TF1 + TF2 group that had at least 50% TF1 + TF2 cells as nearest neighbor, we determined its closest matches to either the TF1 or the TF2 groups by performing the five-nearest neighbor analysis in PCA space (first 20 dimensions), and performing differential expression between the union of these cells with the TF1 + TF2 cells. Genes unique to the TF1 + TF2 group were defined as those with FDR-corrected $P$ value of <0.05 and absolute fold change of >1.5.

**Statistics and reproducibility.** $P$ values of $<2.2 \times 10^{-16}$ or $<2 \times 10^{-16}$ are the default cutoff in R. Statistics, sample sizes, multiple testing corrections and exact $P$ values are listed in Supplementary Table 5 when applicable. For the $t$ test, data distribution was assumed to be normal, but this was not formally tested. Unless specified, $P$ values

are visualized as NS = $P$ > 0.05, *$P$ < 0.05, **$P$ < 0.01, ***$P$ < 0.001 and ****$P$ < 0.0001. By default, the band represents the 95% confidence interval on the smoothed mean of the specified model. If not specified, boxes in standard boxplots indicate the first and third quartiles, the line indicates the median, and the whiskers indicate the first and third quartiles expanded by 1.5× the interquartile range.

The data collection was not randomized. Data collection and analysis were not performed in a blinded manner with respect to the experimental conditions. No statistical method was used to predetermine the sample size. No data were excluded from the analyses, as filtering steps were specified in the respective Methods.

### Reporting summary
Further information on research design is available in the Nature Portfolio Reporting Summary linked to this article.

### Data availability
All raw and processed scTF-seq data are available at ArrayExpress under accession E-MTAB-13010. Uncropped microscopy images reported in this paper are provided as Source data at figshare (https://doi.org/10.6084/m9.figshare.29290625)[100]. Source data are provided with this paper.

### Code availability
All source code is available at GitHub (https://github.com/DeplanckeLab/TF-seq) and Zenodo (https://doi.org/10.5281/zenodo.16892802)[101].

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

## Acknowledgements

We thank C. Aztekin (EPFL), Q. Zhou (EPFL), O. Pushkarev (EPFL) and O. Guillaume-Gentil (EPFL) for reviewing the paper and providing valuable feedback. We thank J. Han (Xiamen University) for providing the pBOB-GFP plasmid and D. Trono (EPFL) for providing the pSIN-TRE-GW-3xHA-puroR plasmid. We thank H. Hashimi (EPFL) and M. Rumpler (EPFL) for their support with image analysis. We also thank the Gene Expression Core Facility (EPFL; especially B. Mangeat), Histology Core Facility (EPFL; especially J. Sordet-Dessimoz), Bioimaging and Optics Platform (EPFL; especially O. Burri and N. Chiaruttini) and Flow Cytometry Core Facility (EPFL) for technical support. This work was supported by a National Key R&D Program of China, Shenzhen Medical Research Funding Program, National Natural Science Foundation of China and a Shenzhen Institute of Synthetic Biology Scientific Research Program (2021YFA0911100, B2302017, 32422049 and JCHZ20210003 to W.C.), Swiss National Science Foundation (grants 310030_197082, CRSII5_186271 and TMAG-3_209335 and institutional funding (EPFL) to B.D.), a Marie Skłodowska-Curie fellowship and FWO senior postdoctoral fellowship (101028476 and 12A0025N to W.S.) and a Marie Skłodowska-Curie fellowship and an EMBO long-term fellowship (101026623 and 2020-895 to G.v.M.).

## Author contributions

W.L., W.S., P.R., W.C. and B.D. designed the study. M.B., W.C. and W.L. performed experiments with the support of A.G., J.R. and T.L. W.L., W.S., P.R. and V.G. performed data analysis with the support of A.J.G. and G.v.M. W.L., W.S., P.R., W.C. and B.D. wrote the paper with input from M.B., V.G., A.J.G., A.G. and G.v.M.

## Competing interests

The authors declare no competing interests.

## Additional information

**Extended data** is available for this paper at https://doi.org/10.1038/s41588-025-02343-7.

**Correspondence and requests for materials** should be addressed to Wanze Chen or Bart Deplancke.

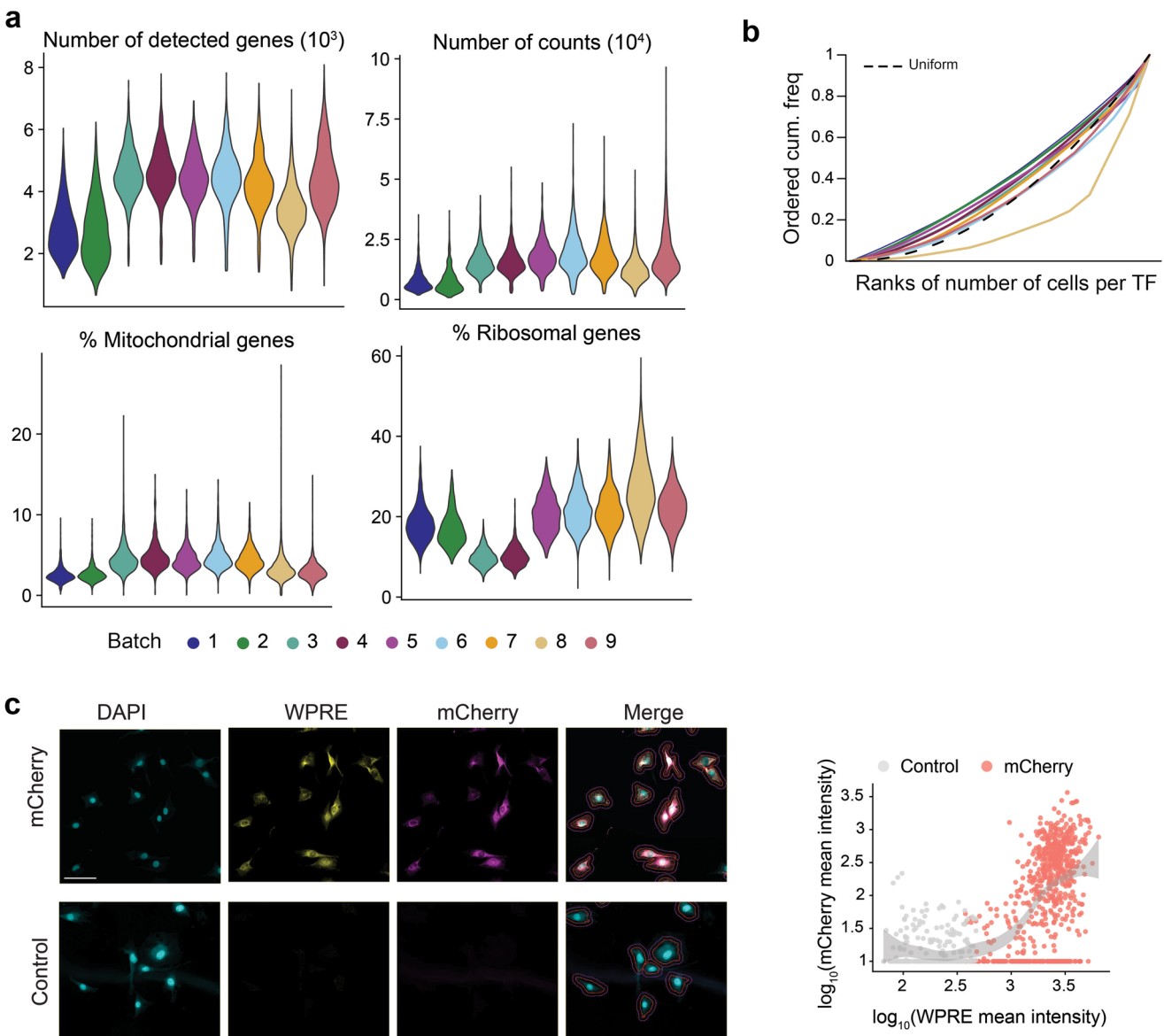

**Extended Data Fig. 1 | Data quality control and dose validation, related to Fig. 1. a**, Violin plots presenting the number of detected genes and counts, the percentage of mitochondrial and ribosomal gene expression of each cell across batches after quality control. **b**, Ordered cumulative frequency (cum. freq) of the number of cells assigned to each TF in each batch. The dashed line shows the cumulative ordered frequency of a uniform distribution as comparison. **c**, Left: representative fluorescence images showing the RNA transcripts of WPRE (proxy for TF dose) probed by RNAscope and mCherry (protein) in mCherry and control cells. Red and purple outlines indicate the cell boundary and expanded cell boundary, respectively. Representative images of two independent experiments. Scale bar = 100 μm. Right: single-cell quantification showing the correlation between the log-normalized mean fluorescence intensity of mCherry versus WPRE mRNA in mCherry and control cells. Fitted model: LOESS.

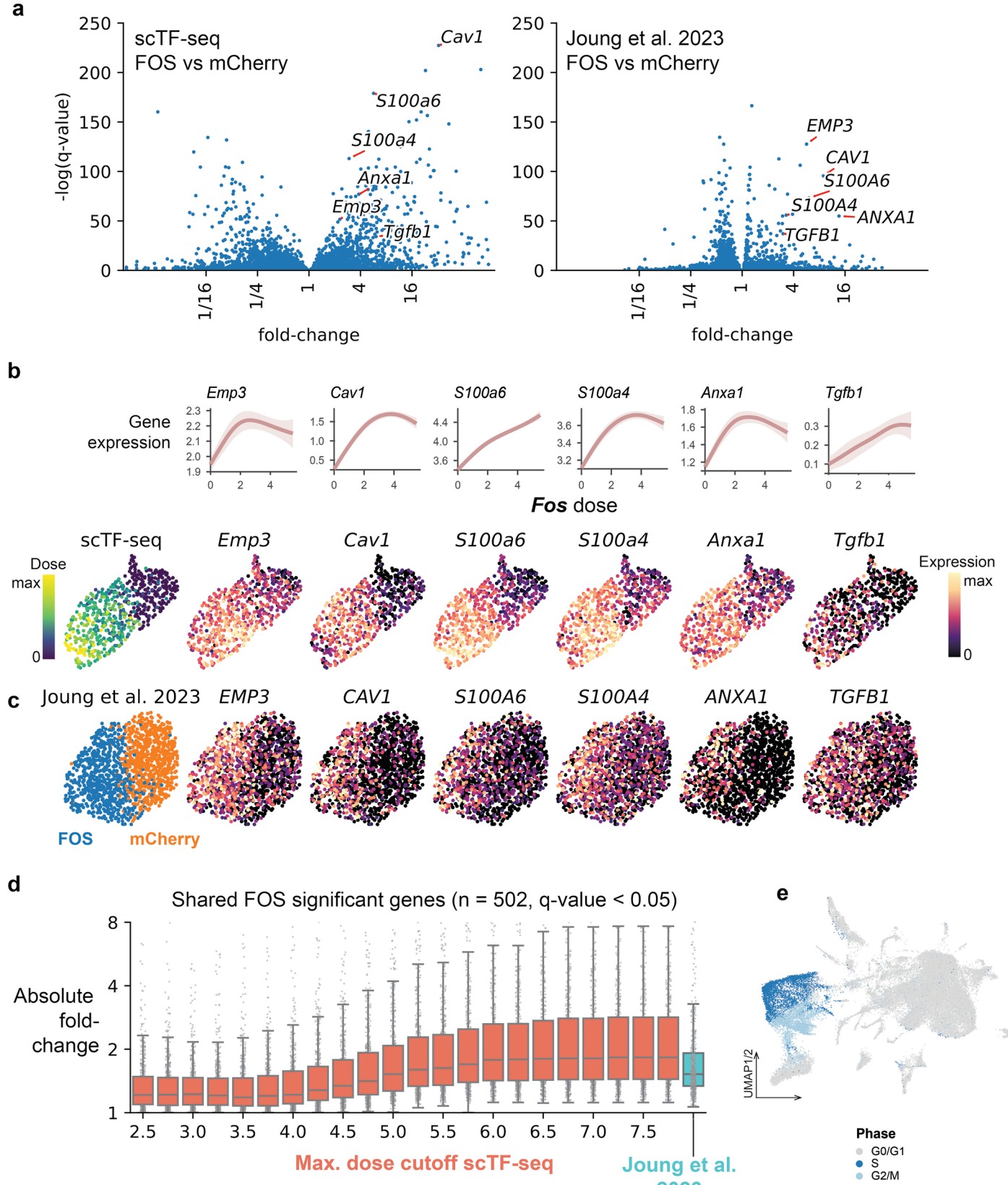

**Extended Data Fig. 2 | See next page for caption.**

**Extended Data Fig. 2 | A wide dose range generated by scTF-seq, related to Fig. 1. a**, Volcano plots showing differential gene expression between control (mCherry) and FOS cells in our study (batch 9) and ref. 16. The q-value is the adjusted *P* value generated by the *t* test followed by FDR correction. **b**, Dose-response curves (top) and exemplary UMAP plots (bottom) for the expression of the top six shared up- and down-regulated genes in response to FOS overexpression in the scTF-seq dataset. These genes were selected by first filtering on significant (FDR <5%) differential expression in both the scTF-seq and Joung et al.[16] datasets, and subsequently selecting the top six differentially expressed genes in ref. 16 dataset. **c**, UMAPs for the expressions of the orthologous genes from **b** on ref. 16 dataset. **d**, Standard boxplots (Methods) showing the distribution of fold changes when filtering the scTF-seq cells according to several maximal dose cutoffs (red) compared to the distribution of differential expression from ref. 16. **e**, UMAP plot of the TF atlas colored by cell cycle phase assigned with a default Seurat cutoff of 0 for cell cycle scores.

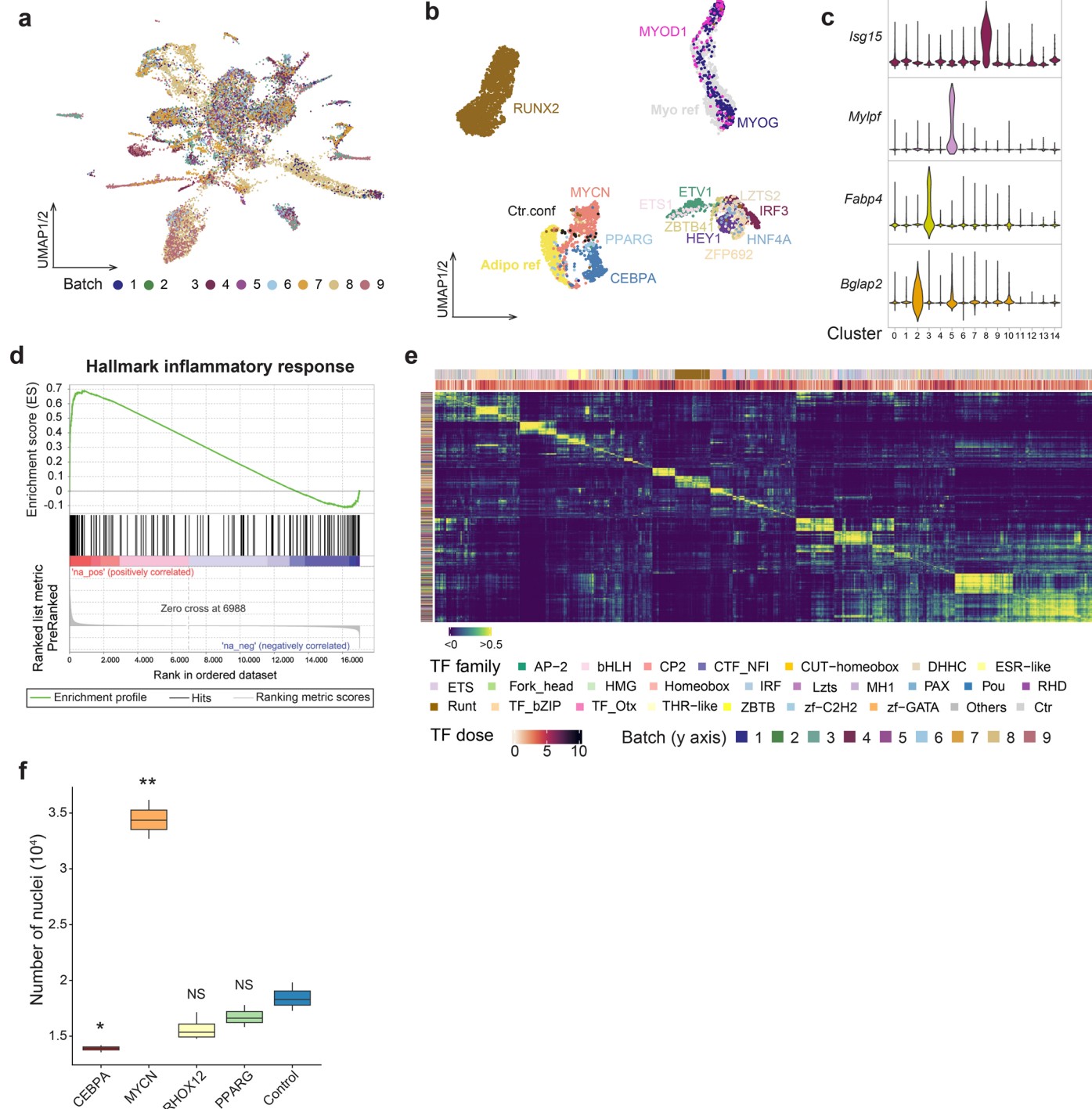

**Extended Data Fig. 3 | Identification of lineage-specific clusters, functional similarity of TF families, related to Fig. 2. a**, UMAP plot of the functional TF atlas (see Fig. 2a, b) colored by batch. **b**, UMAP plot showing a subset of the functional TF atlas, including cells in the four major MSC clusters (adipogenic, myogenic, osteogenic and inflammatory) in Fig. 2b. Cells are colored by TF. Only TFs that have more than 25 cells were plotted. Adipo/Myo ref, adipocyte or myo reference cells. Ctr.conf, confluent control cells. **c**, Expression of marker genes linked to immune response (*Isg15*), myogenic (*Mylpf*), adipogenic (*Fabp4*), or osteogenic (*Bglap2*) lineages in each cluster in Fig. 2b. **d**, Gene set enrichment analysis (GSEA) result of a hallmark inflammatory response performed between cluster 8 versus clusters 0, 1, and 6 combined (containing most of the control cells) in the functional TF atlas (see Fig. 2b). **e**, Heatmap showing a pairwise Pearson correlation of cells annotated by TF family, TF dose (in column) and batch (in row). Only TF families that have at least 30 cells were plotted. Cells are ordered by hierarchical clustering. **f**, Standard boxplot (Methods) showing nuclei counts quantified on images shown in Fig. 2e. Comparison was performed between the control and each TF. Data were collected from two independent experiments, with 1-2 independent wells for each. See Supplementary Table 5 and Methods for statistics and exact *P* values.

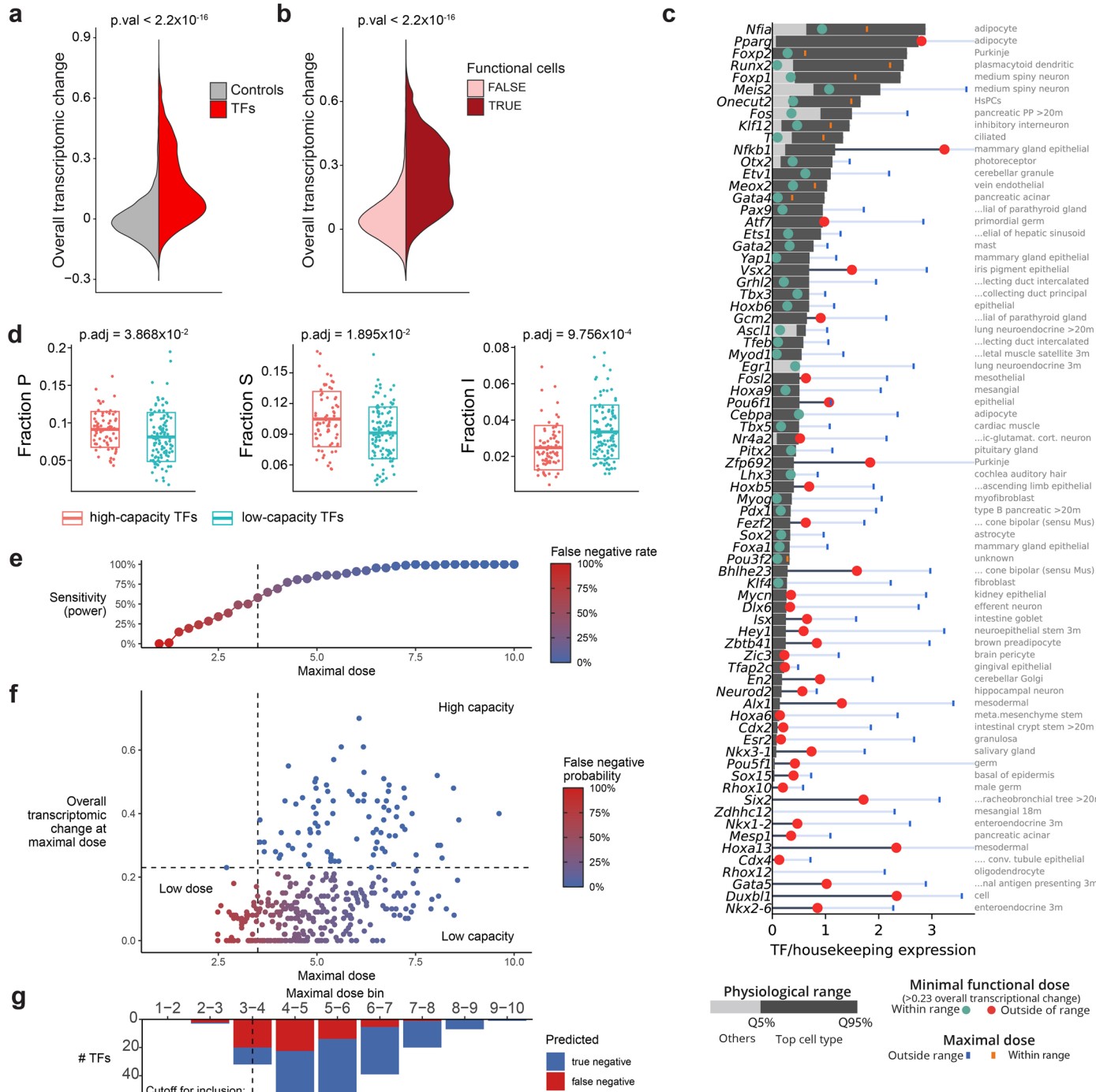

**Extended Data Fig. 4 | Categorizing TFs based on dose sensitivity and reprogramming capacity, related to Figs. 3 and 4. a,b**, Violin plots showing overall transcriptomic change grouped by (**a**) control (mCherry) and TFs cells, or (**b**) functional (TRUE) and non-functional (FALSE) TF cells. p.val, *P* value. **c**, Comparison of the minimal functional dose required to reach a significant transcriptomic effect (circles) to the high endogenous dose observed in *in vivo* single-cell datasets (gray bands, Methods). The minimal functional dose (see **d**) is colored respectively in teal and red if it is above or below the physiological dose. The small vertical lines indicate the maximal dose observed in our study and are colored blue, except when the maximal dose falls below the highest dose observed endogenously, in which case they are colored orange. **d**, Boxplots showing the fraction of proline P, fraction of serine S, fraction of isoleucine I across high-capacity and low-capacity TFs. p.adj, adjusted *P* value. Crossbars and boxes represent mean ± s.d. **e**, Sensitivity to identify a TF as being high-capacity in function of the maximal dose reached by a TF. **f**, Overall transcriptomic change compared to maximal dose, with the probability for a TF as being identified as a false-negative, low-capacity TF highlighted in color. **g**, Predicted number of false-negative, low-capacity TFs at various maximal dose bins. See Supplementary Table 5 and Methods for statistics and exact *P* values.

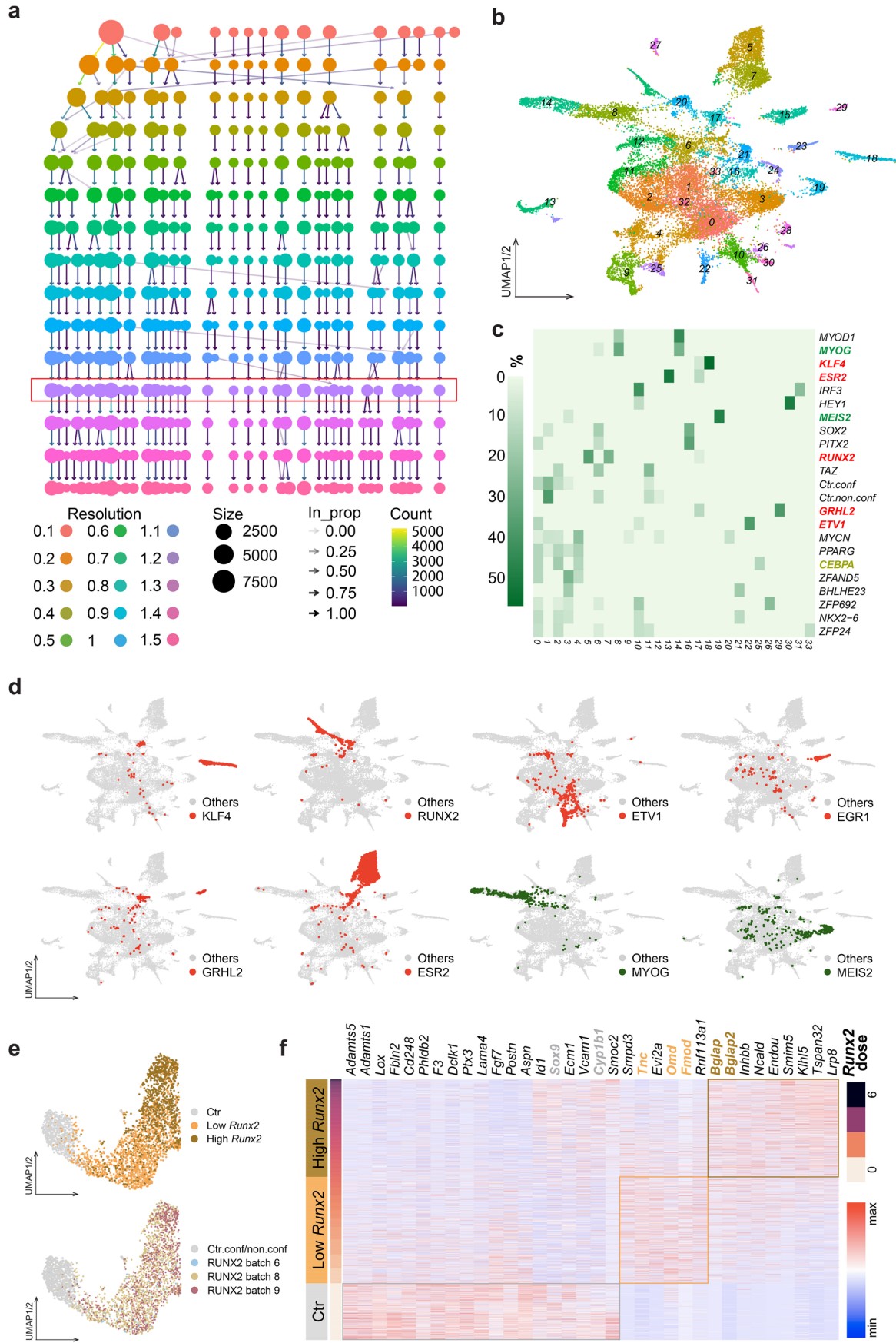

**Extended Data Fig. 5 | See next page for caption.**

**Extended Data Fig. 5 | Discovery of candidate TFs associated with cell state transition, related to Fig. 5. a**, Clustering tree of the Seurat-based clustering result of the phase-adjusted functional TF atlas (consisting of phase-adjusted G1 cells, Methods), visualizing the relationships between clustering at different resolutions. The resolution 1.2 was used for the functional atlas in **b** to generate an optimal number of clusters such that known lineage cells are separated. **b**, UMAP plots of the phase-adjusted functional TF atlas colored by the clustering result at resolution 1.2. **c**, Heatmap showing the proportion of TF cells in each cluster relative to the total number of TF cells. Ctr.conf, confluent control cells; Ctr.non.conf, non-confluent control cells. Only controls and TF candidates

identified by cell state transition analysis (see Methods) were plotted. **d**, UMAP plots of the phase-adjusted functional TF atlas (shown in **b**) with KLF4, RUNX2, ETV1, EGR1, GRHL2, ESR2, MYOG and MEIS2 cells highlighted (red for TFs inducing the dose-dependent cell state transition, green for TFs exhibiting the stochastic state transition). **e**, UMAP plots of RUNX2 cells and their batch-paired control colored by groups classified according to *Runx2* dose (top) or batch (bottom). **f**, Heatmap displaying log-normalized expression (*z* score scaled by gene) of differentially expressed genes of high and low *Runx2* and control (Ctr) cells. Cells are ordered by *Runx2* dose as indicated by the color bar on the left.

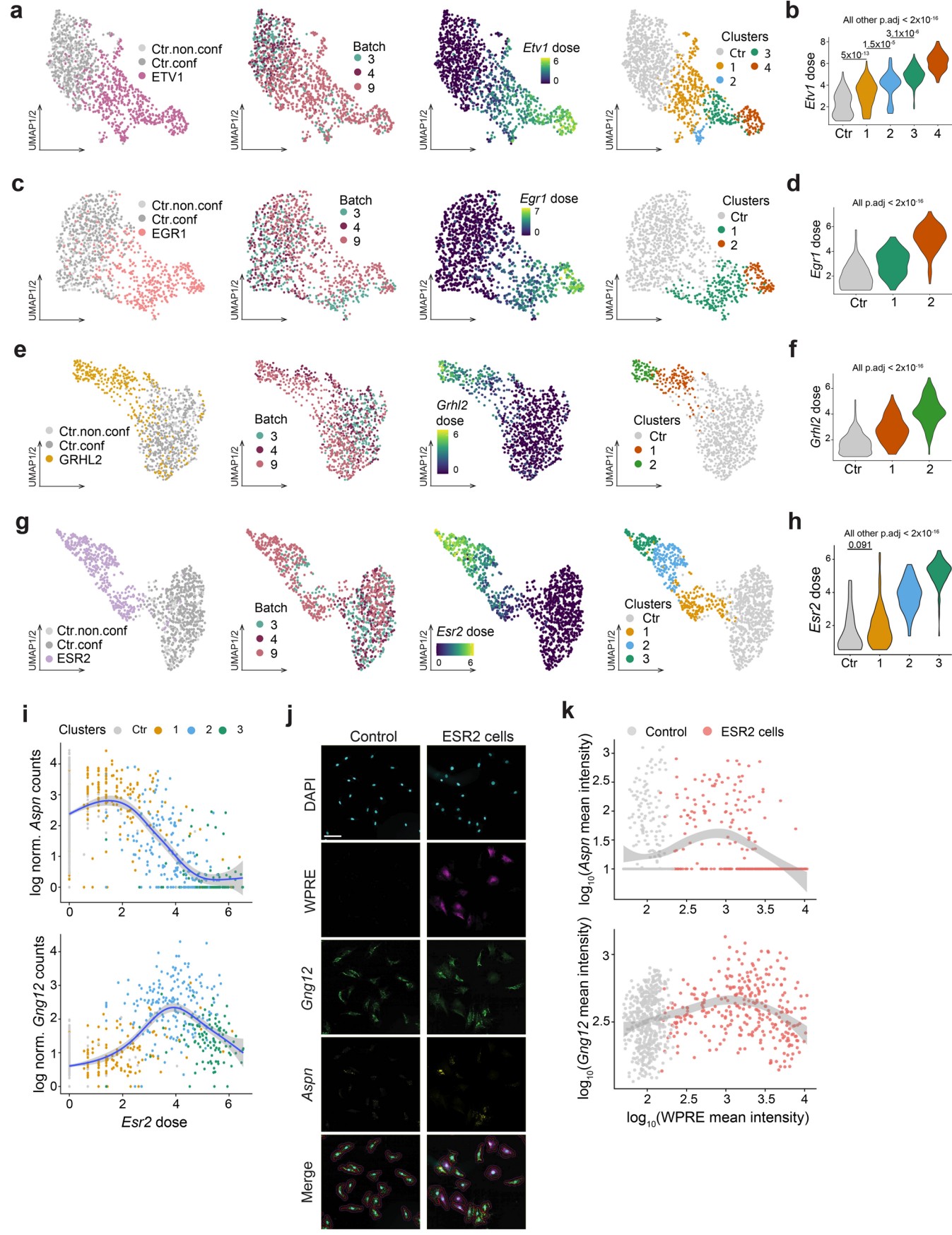

**Extended Data Fig. 6 | See next page for caption.**

**Extended Data Fig. 6 | Examples showing dose-dependent cell state transition, related to Fig. 5. a**,**c**,**e**,**g**, UMAP plots of ETV1 (**a**), EGR1 (**c**), GRHL2 (**e**), or ESR2 (**g**) cells and their batch-paired control cells colored by category, batch, TF dose, or cluster. Ctr.non.conf, non-confluent control cells; Ctr.conf, confluent control cells. **b**,**d**,**f**,**h**, Violin plots showing the dose distribution of ETV1 (**b**), EGR1 (**d**), GRHL2 (**f**), or ESR2 (**h**) cells in the different clusters shown in **a**, **c**, **e**, **g**, respectively. p.adj, adjusted P value. **i**, Scatter plot showing the expression of *Aspn* (top) or *Gng12* (bottom) in ESR2 cells (colored by the clusters shown in **g**, right) and batch-paired control cells in function of *Esr2* dose. Fitted model:

GAM. **j**, RNAscope images for DAPI, WPRE (proxy for TF dose), *Gng12*, and *Aspn* expression in ESR2 cells. Wildtype C3H10T1/2 cells were used as the control. All fluorescent channels were merged for cell segmentation, indicated by the red (cell boundary) and purple (expanded cell boundary) outlines. Representative images of two independent experiments. Scale bar = 100 μm. **k**, Single-cell quantification of RNAscope (as shown in **j**) showing the log-normalized mean intensity of WPRE (proxy for TF dose) versus *Aspn* (top) or *Gng12* (bottom) in control and ESR2 cells. Fitted model: LOESS. See Supplementary Table 5 and Methods for statistics and exact P values.

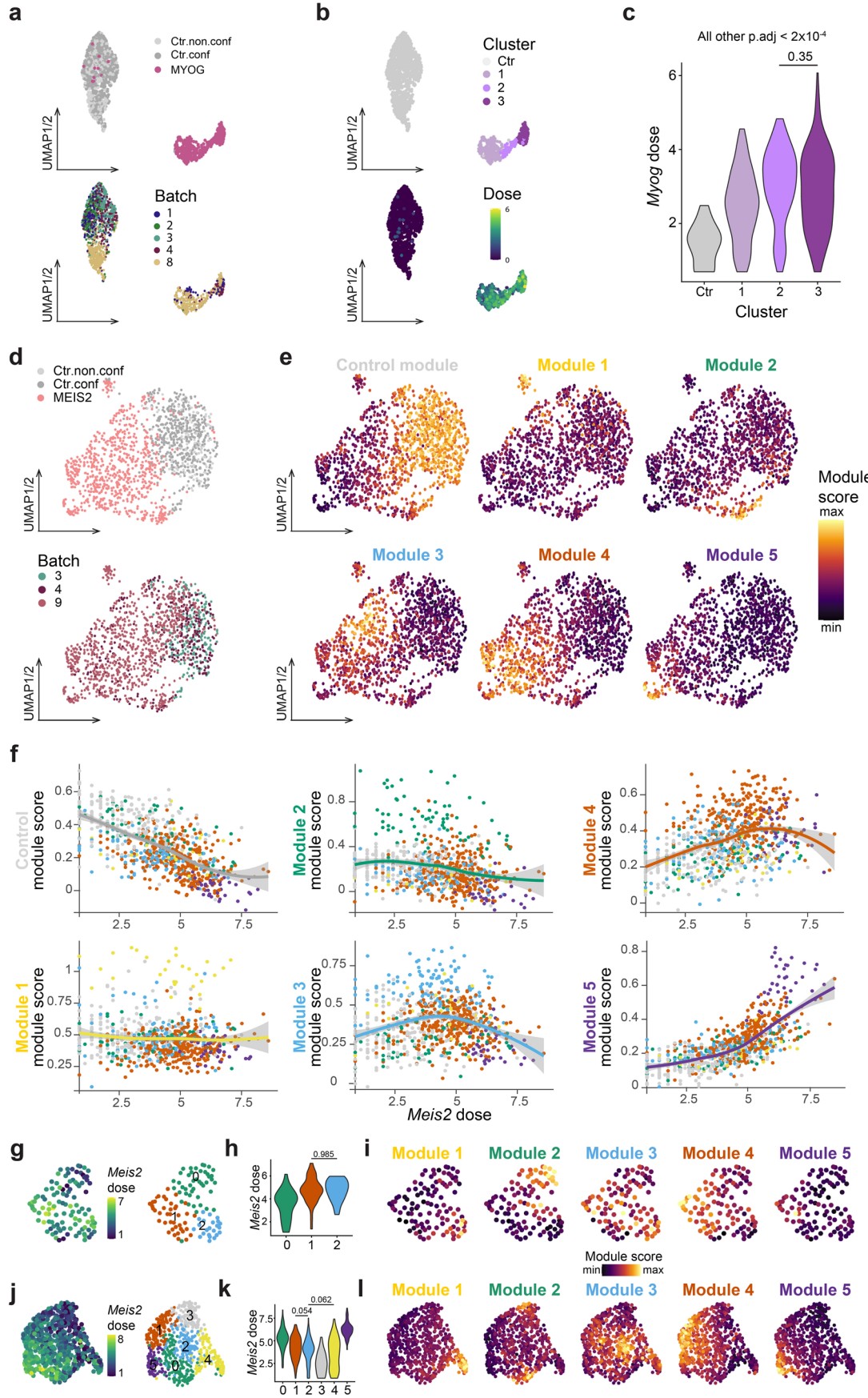

**Extended Data Fig. 7 | See next page for caption.**

**Extended Data Fig. 7 | MYOG and MEIS2 showing dose-dependent and stochastic cell state transition, related to Fig. 5. a,b,** UMAP plots of MYOG and batch-paired control cells colored by category or batch (**a**), cluster or dose (**b**). Ctr.non.conf, non-confluent control cells; Ctr.conf, confluent control cells. **c**, Violin plot showing the dose distribution of MYOG cells in the different clusters shown in **b**. p.adj, adjusted *P* value. **d**, UMAP plots of MEIS2 and batch-paired control cells colored by condition (top) or batch (bottom). Ctr.non.conf, non-confluent control cells; Ctr.conf, confluent control cells. **e**, UMAP plots of MEIS2 and batch-paired control cells colored by the module scores based on the top differentially expressed genes of each Control (Ctr) and MEIS2 cluster (Cluster 1 to 5) shown in Fig. 5k. **f**, Scatter plots showing the module scores in MEIS2 and batch-paired control cells along *Meis2* dose. The module scores were based on the top differentially expressed genes of each Control (Ctr) or MEIS2

cluster (cluster 1 to 5) shown in Fig. 5k. Fitted model: GAM. **g**, UMAP plots of MEIS2 cells from batch 4 (n = 108 cells) colored by *Meis2* dose or unsupervised clustering results that were independently acquired on batch 4. **h**, Violin plot showing the dose distribution of MEIS2 cells in the different clusters shown in **g**. **i**, UMAP plots of MEIS2 cells from batch 4 colored by the module scores based on the top differentially expressed genes of each Control (Ctr) and MEIS2 cluster (Cluster 1 to 5) shown in Fig. 5k. **j**, UMAP plots of MEIS2 cells from batch 9 (n = 645 cells) colored by *Meis2* dose or unsupervised clustering results that were independently acquired on batch 9. **k**, Violin plot showing the dose distribution of MEIS2 cells in the different clusters shown in **j**. **l**, UMAP plots of MEIS2 cells from batch 9 colored by the module scores based on the top differentially expressed genes of each Control (Ctr) and MEIS2 cluster (Cluster 1 to 5) shown in Fig. 5k. See Supplementary Table 5 and Methods for statistics and exact *P* values.

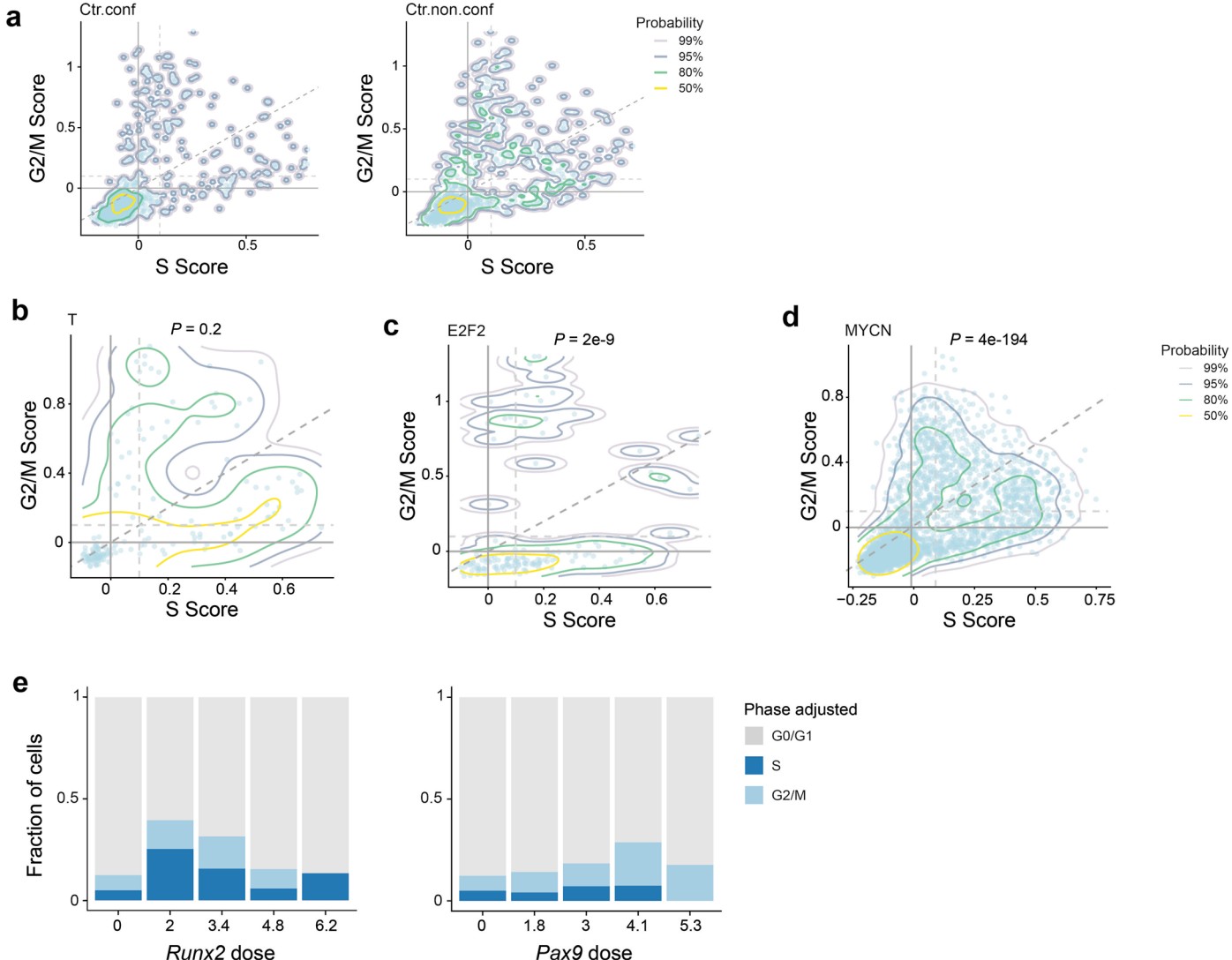

**Extended Data Fig. 8 | Cell cycle dynamics regulated by TFs and their doses, related to Fig. 6. a-d**, 2D density contour plots of S and G2/M scores showing cell cycle dynamics of confluent (**a**, left) and non-confluent control (**a**, right), T (**b**), E2F2 (**c**), and MYCN (**d**) cells. See Supplementary Note 7. The colors of contour lines represent the probability associated with the computed highest density region. For example, 50% represents the region capturing 50% of the data points. $P$, adjusted $P$ value. **e**, Bar plots showing the fraction of cells in each adjusted cell cycle phase across binned doses of *Runx2* or *Pax9*. See Supplementary Table 5 and Methods for statistics and exact $P$ values.

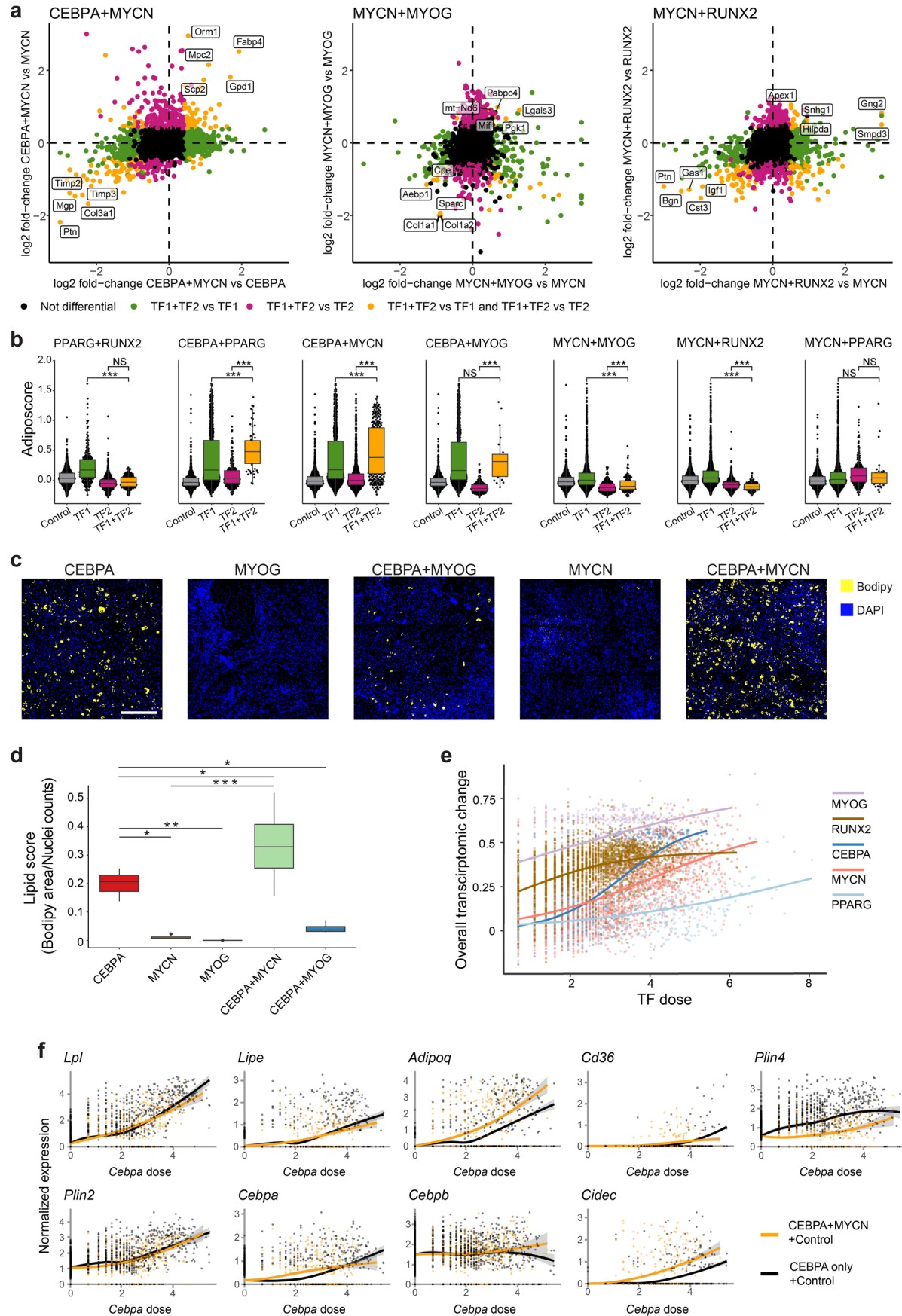

**Extended Data Fig. 9 | See next page for caption.**

**Extended Data Fig. 9 | Global and dose-dependent transcriptomic and phenotypic interactions between pairs of TFs, related to Fig. 7. a**, Differential expression between cells with a pair of overexpressed TFs compared to those with a single TF overexpression. Shown are log2-fold changes between TF1 + TF2 cells compared to a set of reference cells, defined as the union of all 5-nearest neighbors single TF cells for each TF1 + TF2 cell. Significant differential expression was defined using an FDR-adjusted p-value < 0.05 and absolute log2 fold-change >1.5. Highlighted in orange are those genes that were up- or down-regulated in both comparisons, and that constitute genes that are uniquely regulated in the combination cells. **b**, Transcriptomic adiposcore between control cells, single TF cells and TF1 + TF2 cells. **c**, Representative fluorescence images of lipid droplets (stained with Bodipy, yellow) and nuclei (stained with DAPI, blue) in single TF and TF1 + TF2 cells. Scale bar, 500 μm. **d**, Quantification of the lipid score as in (**c**). Data were collected from two independent experiments, with 1-2 independent wells for each. **e**, Scatter plot showing the overall transcriptomic change across TF dose colored for the individual TFs involved in combinatorial experiments. The lines represent the fitted logistic regression. **f**, Dose-response curves for the expressions of all transcriptomic adiposcore genes in control and CEBPA cells (black), compared to those for control and MYCN + CEBPA cells (orange). Fitted model: LOESS. See Supplementary Table 5 and Methods for statistics and exact *P* values.

# Reporting Summary

## Statistics

For all statistical analyses, confirm that the following items are present in the figure legend, table legend, main text, or Methods section.

| n/a | Confirmed | |
|---|---|---|
| ☐ | ☒ | The exact sample size (*n*) for each experimental group/condition, given as a discrete number and unit of measurement |
| ☐ | ☒ | A statement on whether measurements were taken from distinct samples or whether the same sample was measured repeatedly |
| ☐ | ☒ | The statistical test(s) used AND whether they are one- or two-sided<br>*Only common tests should be described solely by name; describe more complex techniques in the Methods section.* |
| ☐ | ☒ | A description of all covariates tested |
| ☐ | ☒ | A description of any assumptions or corrections, such as tests of normality and adjustment for multiple comparisons |
| ☐ | ☒ | A full description of the statistical parameters including central tendency (e.g. means) or other basic estimates (e.g. regression coefficient) AND variation (e.g. standard deviation) or associated estimates of uncertainty (e.g. confidence intervals) |
| ☐ | ☒ | For null hypothesis testing, the test statistic (e.g. *F*, *t*, *r*) with confidence intervals, effect sizes, degrees of freedom and *P* value noted<br>*Give P values as exact values whenever suitable.* |
| ☒ | ☐ | For Bayesian analysis, information on the choice of priors and Markov chain Monte Carlo settings |
| ☒ | ☐ | For hierarchical and complex designs, identification of the appropriate level for tests and full reporting of outcomes |
| ☐ | ☒ | Estimates of effect sizes (e.g. Cohen's *d*, Pearson's *r*), indicating how they were calculated |

*Our web collection on statistics for biologists contains articles on many of the points above.*

## Software and code

Policy information about availability of computer code

Data collection | Basecalls performed on NextSeq 500/HiSeq 4000/NovaSeq 6000 result using bcl2fastq v.2.19/v2.20.

Data analysis | All source code used to analyze the data in this study can be found at https://github.com/DeplanckeLab/TF-seq and is publicly available upon publication.

R (v4.1.0, v4.3.1)
Cell Ranger (v3.0.2, v6.0.0, and v7.1.0 were used to generate count matrices of experiments 1-7, 8, and 9, respectively)
SamSPECTRAL (v1.46.0)
Seurat (v4.4.0)
scran (v1.22.1)
scanpy (v1.10.2)
clustree (v0.5.0)
cellpose (v3)
edgeR (v3.36.0)
fgsea (v1.18.0)
clusterProfiler (v4.0.5)
Hmisc (v5.1-1)
stats (v4.1.0)
Monocle3
ks (v1.14.1)

| Flowjo v10.10 |
|---|

For manuscripts utilizing custom algorithms or software that are central to the research but not yet described in published literature, software must be made available to editors and reviewers. We strongly encourage code deposition in a community repository (e.g. GitHub). See the Nature Portfolio guidelines for submitting code & software for further information.

## Data

Policy information about availability of data

All manuscripts must include a data availability statement. This statement should provide the following information, where applicable:
- Accession codes, unique identifiers, or web links for publicly available datasets
- A description of any restrictions on data availability
- For clinical datasets or third party data, please ensure that the statement adheres to our policy

All raw sequencing data generated in this paper has been deposited at ArrayExpress with the accession number E-MTAB-13010. Vector sequence for TF barcode counting has been deposited at github (vector_sequence, https://github.com/DeplanckeLab/TF-seq). The following genome assembly was used: GRCh38, release 96 from Ensembl. Marker genes of cell types of interest and hallmark gene sets were downloaded from MSigDB v2023.1.Mm and PanglaoDB v2019.
150 annotated single-cell datasets were downloaded from CELLXGENE census database. Mutational constraints quantified from variation in 141,456 humans were downloaded from gnomAD (v4.0). TF class annotations were collected from AnimalTFDB (v4.0). Microscopy data reported in this paper will be shared upon request.

## Research involving human participants, their data, or biological material

Policy information about studies with human participants or human data. See also policy information about sex, gender (identity/presentation), and sexual orientation and race, ethnicity and racism.

| Reporting on sex and gender | This information was not collected as only cell lines and no human or animal subjects were used. |
|---|---|
| Reporting on race, ethnicity, or other socially relevant groupings | This information was not collected as only cell lines and no human or animal subjects were used. |
| Population characteristics | This information was not collected as only cell lines and no human or animal subjects were used. |
| Recruitment | This information was not collected as only cell lines and no human or animal subjects were used. |
| Ethics oversight | This information was not collected as only cell lines and no human or animal subjects were used. |

Note that full information on the approval of the study protocol must also be provided in the manuscript.

# Field-specific reporting

Please select the one below that is the best fit for your research. If you are not sure, read the appropriate sections before making your selection.

☒ Life sciences        ☐ Behavioural & social sciences        ☐ Ecological, evolutionary & environmental sciences

For a reference copy of the document with all sections, see nature.com/documents/nr-reporting-summary-flat.pdf

# Life sciences study design

All studies must disclose on these points even when the disclosure is negative.

| Sample size | Nine independent scTF-seq experiments were performed with some TFs involved in more than one experiment as replicates (as described in the manuscript and metadata). 45978 cells covering 384 individual TFs and 7 TF pairs were used for the construction of scTF-seq atlas. Sample sizes for this study were determined based on established empirical thresholds commonly used in single-cell omics to ensure robust detection of biological signals and statistical reliability. For differential expression (DE) analysis, we required a minimum of 5 functional cells per TF to retain sufficient power to detect expression changes while mitigating false positives from underpowered comparisons. For gene set enrichment analysis (GSEA), which requires greater statistical resolution to assess pathway-level effects, we imposed a stricter threshold of ≥25 functional cells per TF. To evaluate TF dose sensitivity, reprogramming capacity, and cell fate transitioning—analyses that demand finer granularity to resolve dynamic or bifurcating behaviors—we restricted analysis to TFs with ≥30 cells. Cell cycle dynamics, which involve partitioning cells into discrete phases (G1/S/G2M), required ≥50 cells per TF to ensure adequate representation across phases for statistical testing. Statistic tests were performed for all the analyses as indicated in the Figure Legends and Methods accordingly. |
|---|---|
| Data exclusions | Single-cell RNA-seq samples showing low quality were excluded. The criteria are described in the Methods and Supplementary table 1. Cell clumps, debris, and 1% of outliers at the extreme lower or upper tails of the mean intensity distribution for individual RNAscope image channels were filtered out. The criteria are described in the Methods. |
| Replication | At least six transcription factors (as described in the metadata) were involved in more than one scTF-seq experiment. The imaging of mCherry expression was performed with three replicates. The profiling of mCherry fluorescence intensity was performed on two mCherry- |

overexpressing cell line, each with two replicates. RNAscope experiment was perfomed with at least two replicates. Validations of adipogenic capacity of single TFs or TF pairs were performed with at least three replicates. Cell death staining was performed with three replicates. These replicates successfully show consistent results.

Randomization

In this study, randomization was not performed due to the controlled experimental design and systematic nature of the scTF-seq approach. Specifically:
1. Only one isogenic cell line was used for the TF overexpression screen, minimizing genetic and environmental variability.
2. Each TF or TF pair was tested in a targeted manner to directly assess its effect, with cells assigned to experimental groups based on the TF(s) expressed (not random assignment). This ensures unambiguous attribution of observed outcomes to specific TFs.
3. Variability was addressed by performing nine independent scTF-seq experiments, with TFs tested across replicates. Batch effects were mitigated computationally (integration/batch correction) rather than via randomization.
4. Statistical robustness was ensured by excluding TFs with low cell counts (e.g., <5 or < 25 functional cells for DE or GSEA analysis), which serves a similar purpose to randomization in reducing noise from undersampled groups.
5. In single-cell perturbation screens, systematic testing of all factors under matched conditions (rather than randomized subsets) is standard practice to enable direct comparisons and atlas-scale profiling.

Blinding

The imaging of mCherry expression, lipid accumulation or cell viability, and RNAscope was performed blindingly. Blinding is not necessary for other analyses as they are quantitative and no subjective interpretation is required.

# Reporting for specific materials, systems and methods

We require information from authors about some types of materials, experimental systems and methods used in many studies. Here, indicate whether each material, system or method listed is relevant to your study. If you are not sure if a list item applies to your research, read the appropriate section before selecting a response.

## Materials & experimental systems

| n/a | Involved in the study |
|-----|----------------------|
| ☒ ☐ | Antibodies |
| ☐ ☒ | Eukaryotic cell lines |
| ☒ ☐ | Palaeontology and archaeology |
| ☒ ☐ | Animals and other organisms |
| ☒ ☐ | Clinical data |
| ☒ ☐ | Dual use research of concern |
| ☒ ☐ | Plants |

## Methods

| n/a | Involved in the study |
|-----|----------------------|
| ☒ ☐ | ChIP-seq |
| ☐ ☒ | Flow cytometry |
| ☒ ☐ | MRI-based neuroimaging |

## Eukaryotic cell lines

Policy information about cell lines and Sex and Gender in Research

Cell line source(s)

HEK293T and C3H10T1/2 cells used in this study were obtained from ATCC.

Authentication

None of the cell lines used were authenticated

Mycoplasma contamination

All cell lines used in this study were tested negative for mycoplasma contamination.

Commonly misidentified lines
(See ICLAC register)

No cell lines is misidentified to our knowledge

## Plants

Seed stocks

This information was not collected as only cell lines and no human or animal subjects were used.

Novel plant genotypes

This information was not collected as only cell lines and no human or animal subjects were used.

Authentication

This information was not collected as only cell lines and no human or animal subjects were used.

# Flow Cytometry

## Plots

Confirm that:

☒ The axis labels state the marker and fluorochrome used (e.g. CD4-FITC).

☒ The axis scales are clearly visible. Include numbers along axes only for bottom left plot of group (a 'group' is an analysis of identical markers).

☒ All plots are contour plots with outliers or pseudocolor plots.

☒ A numerical value for number of cells or percentage (with statistics) is provided.

## Methodology

| | |
|---|---|
| Sample preparation | C3H10T1/2 cells (Wildtype and mCherry overexpressing) were harvested by trypsinization, followed by quenching with growth medium. Cells were washed with PBS and resuspended in ice-cold PBS with DAPI (1 μg/mL) on ice. |
| Instrument | BD LSR Fortessa 5-laser cell analyzer |
| Software | Collection: BD FACSDiva 8.0.1<br>Analysis: FlowJo v10.10 and R v4.1.0 |
| Cell population abundance | Between 75-90 % events passed the FSC/SSC gating used for analyzing single cells, of which at least 95 % were alive based on negative DAPI signal. Cells were not sorted further, just analyzed for mCherry fluorescence intensity. |
| Gating strategy | Cells were gated based on FSC and SSC to select single cells. Live cells were gated based on negative DAPI signal (355nm - 450/50nm). Live cells were analyzed for mCherry signal (561nm - 610/20nm). |

☒ Tick this box to confirm that a figure exemplifying the gating strategy is provided in the Supplementary Information.

