## [Peer Review File · Nature Genetics]

Dissecting the impact of transcription factor dose on cell reprogramming heterogeneity using scTF-seq

Corresponding Author: Professor Bart Deplancke

Version 0:

Decision Letter:

3rd Apr 2024

Dear Bart,

Thank you for you and your co-authors' patience during this prolonged review period.

Your Article, "Dissecting the impact of transcription factor dose on cell reprogramming heterogeneity using scTF-seq" has now been seen by 2 referees. You will see from their comments copied below that while they find your work of considerable potential interest, they have raised quite substantial concerns that must be addressed. In light of these comments, we cannot accept the manuscript for publication, but would be very interested in considering a revised version that addresses these serious concerns.

Briefly, the two referees are split in their overall opinions of your manuscript, and highlight some overlapping concerns of great import, but also seem to suggest there is a clear path to an eventual publication.

Referee #1, while noting features of scTF-seq as "exciting", is unconvinced that the strength of evidence presented for your claims is sufficient, and notes that there may be substantial batch effects in your dataset, while also noting it's unclear how reproducible the results are, amongst other comments.

Reviewer #2 is more supportive, but their comments overlap with some of Q1's technical critiques (e.g. batch effects). They also request that experimental validation be performed on at least some of your candidate TFs.

In our reading of these reports, we think that the basic technical/methodological/statistical questions are all fair, and we think you will need to fully address all of these concerns to persuade the reviewers - especially the more critical Referee #1. We would particularly encourage you to focus on solidly demonstrating that scTF-seq is reproducible/reliable (which Reviewer #1 is clearly skeptical about, currently), and also to add the experimental validation requested by Reviewer #2. Finally, it's not clear whether Referee #1 is asking for additionally scTF-seq experiments to be performed - perhaps further data analysis would be sufficient to answer their questions - but we would certainly recommend doing so, if practical.

We hope you will find the referees' comments useful as you decide how to proceed. If you wish to submit a substantially revised manuscript, please bear in mind that we will be reluctant to approach the referees again in the absence of major revisions.

To guide the scope of the revisions, the editors discuss the referee reports in detail within the team, including with the chief editor, with a view to identifying key priorities that should be addressed in revision and sometimes overruling referee requests that are deemed beyond the scope of the current study. We hope that you will find the prioritised set of referee points to be useful when revising your study. Please do not hesitate to get in touch if you would like to discuss these issues further.

If you choose to revise your manuscript taking into account all reviewer and editor comments, please highlight all changes in the manuscript text file. At this stage we will need you to upload a copy of the manuscript in MS Word .docx or similar editable format.

*2) If you have not done so already please begin to revise your manuscript so that it conforms to our Article format instructions, available here. Refer also to any guidelines provided in this letter.

Please be aware of our guidelines on digital image standards.

Link Redacted

If you wish to submit a suitably revised manuscript we would hope to receive it within 6 months. If you cannot send it within this time, please let us know. We will be happy to consider your revision so long as nothing similar has been accepted for publication at Nature Genetics or published elsewhere. Should your manuscript be substantially delayed without notifying us in advance and your article is eventually published, the received date would be that of the revised, not the original, version.

Nature Genetics is committed to improving transparency in authorship. As part of our efforts in this direction, we are now requesting that all authors identified as 'corresponding author' on published papers create and link their Open Researcher and Contributor Identifier (ORCID) with their account on the Manuscript Tracking System (MTS), prior to acceptance. ORCID helps the scientific community achieve unambiguous attribution of all scholarly contributions. You can create and link your ORCID from the home page of the MTS by clicking on 'Modify my Springer Nature account'. For more information please visit please visit www.springernature.com/orcid.

Thank you for the opportunity to review your work.

Sincerely,

Michael Fletcher, PhD
Senior Editor, Nature Genetics
ORCID: 0000-0003-1589-7087

Referee expertise: gene regulation; synthetic/systems biology.

Reviewers' Comments:

Reviewer #1:

Remarks to the Author:

This manuscript introduces scTF-seq, a method for quantifying single-cell transcriptional changes induced by the overexpression of transcription factors (TFs). Using scTF-seq, the authors produce an overexpression atlas for 384 TFs in mouse embryonic multipotent stromal cells (MSCs), identifying key TFs for MSC lineage differentiation. The paper explores dose-dependent effects of TFs and classifies TFs into four categories based on their reprogramming capacity and their sensitivity to doses. Finally, the paper examines transcriptional changes caused by pairs of TFs, revealing that the combined effect of two TFs is influenced by the relative dose of the two TFs.

Overall, the data in this study do not support the claims being made. The primary limitation of the study is that the scTF-seq experiment was only performed once, making it difficult to assess the reproducibility of the method or estimate error bars for the measurements. This is an important point because many of the claims are based on observations that are subtle (e.g. the clustering in Figs 4C and 4F) and could be within the margin of error.

Another issue is that the TFs are assayed in 8 separate batches. The authors appropriately perform batch correction, but some evidence is needed to show that batch correction is indeed working. Could the authors show that the clustering of TF functional modules and the classification of reprogramming capacity/dose-sensitivity are not influenced by batch effects? In Figure 1e, it appears that certain batches (e.g. the pink batch) exhibit distinct transcriptional patterns on the UMAP. The authors should clarify whether these batch-specific patterns are due to enrichment of certain TF groups in one batch, or due to batch artifacts. A related point is that the technique uses UMI counts of TF-IDs to quantify TF doses. UMI counts can vary across different batches because of variations in 10x capture efficiency and PCR efficiency.

An exciting feature of scTF-seq is the potential to discern dose-dependent responses to TFs, because the Dox-inducible promoter drives different levels of expression at different Dox concentrations. However, the authors only expressed TFs at one dox concentration and relied on the natural fluctuations of the Dox promoter to produce different TF doses. The rationale for using 2ug/ml Dox concentration is not given and it is not clear whether the TF levels induced at this Dox concentration are sufficient for effective reprogramming. Also, the range of concentrations produced by natural cell-to-cell fluctuations does not appear to be wide enough to accurately fit the dosage curves in Figure 3c,d and Supplementary Figure 3c. In most cases the dose-response curves did not reach saturation, which makes fitting unreliable.

Minor comments:

It should be made clear in the main text that each TF is transduced separately before pooling the cells, as this is not immediately obvious.

It would be helpful if the authors can explain the criteria for selecting the 384 TFs. Are these TFs expressed/not expressed in MSCs? Are they known to be involved in MSC reprogramming ?

Reviewer #2:

Remarks to the Author:

Overall this was a very well written paper and the experiment is well designed (e.g. arrayed format to avoid recombination etc.) and analyzed. The analysis of TF dosage in particular is interesting and novel.

Major points:

- Apart from the other points mentioned below, it is important that the authors could experimentally validate some of their conclusions. Mainly because their cell numbers per perturbation are quite low, so replicating two or three of these results in an orthogonal way would increase confidence in them. This could be quite simple, e.g. show that low Pparg dose leads to higher adipogenic capacity than higher doses via qPCRs.

- There appear to be some batch effects between the experimental batches (Fig. 2C, 1e), and likely there are also some batch effects across wells within the experiment, given the arrayed format. Can the authors assess well-to-well variability by measuring the same effects as they measure for TFs for mCherry (using a different mCherry well in the same batch as control)? Can they put the observed variances associated with TFs from different batches in context with the observed variances observed for mCherry within the same and across batches? Were some TFs in all batches and if so how does their variance compare to intra-batch inter-TF variance? Did they measure multiple wells per TF and batch, and if so how well do these replicate? Also the batch composition information should ideally be provided somewhere in a table.

- It would be good to put the observed TF levels into context with endogenous TF levels (e.g. using non TF-enriched single cell data). For example generate a version of Figure 1f for endogenously expressed TFs and ORF TFs without TF enrichment. This is key to assess whether the generated levels are far beyond or similar to what is normally observed.

- 'Euclidean distance of cells reprogrammed by TFs to the centroids of control cells in the space consisting of significant principal component coordinates' is used for assessing functional TF cells. It is notoriously challenging to accurately measure distance in high-dimensional space. Can the authors show that similar results are obtained with a different distance measure, e.g. cosine distance?

- The cell numbers are very low in some cases (e.g. 4f the clusters only contain 20 or so cells - are these from same or different batch?). Likewise in that figure it is unclear which cells are actually control cells. Cells would need to be colored by batch and control.

Minor questions:

- why these 419 TFs, which criteria were applied? Are they already expressed endogenously in the starting cell line?

- what is the transduction rate, do cells receive 1 or potentially multiple TF-IDs based on a Poisson distribution? This wouldn't largely affect results given the arrayed format, but would help to put the 86% assignment rate into context

- Why are there only 13 TFs in Figure 2c?

- is there a correlation between mCherry fluorescence and mCherry TF-ID? This would be interesting to see whether all TF-ID constructs are functional (although potential truncations will be length dependent)

Minor point:

- Add fold-change to enrichment tests (e.g. line 243)

Version 1:

Decision Letter:

20th Feb 2025

Dear Bart,

My apologies for the prolonged duration of review; but we do have some useful and, pleasingly, positive feedback for you and your co-authors.

Your Article, "Dissecting the impact of transcription factor dose on cell reprogramming heterogeneity using scTF-seq" has now been seen by 2 referees. You will see from their comments below that while they find your work of interest, some important points are raised. We are interested in the possibility of publishing your study in Nature Genetics, but would like to consider your response to these concerns in the form of a revised manuscript before we make a final decision on publication.

Briefly, the two reviewers are both supportive of publication; one of them has a few new comments which they say would improve the work, and (in our reading) they do seem relatively simply addressable (i.e. with more computational analysis), such that we hope you'll be able to respond to them in full.

To guide the scope of the revisions, the editors discuss the referee reports in detail within the team, including with the chief editor, with a view to identifying key priorities that should be addressed in revision and sometimes overruling referee requests that are deemed beyond the scope of the current study. We hope that you will find the prioritized set of referee points to be useful when revising your study. Please do not hesitate to get in touch if you would like to discuss these issues further.

We therefore invite you to revise your manuscript taking into account all reviewer and editor comments. Please highlight all changes in the manuscript text file. At this stage we will need you to upload a copy of the manuscript in MS Word .docx or similar editable format.

*2) If you have not done so already please begin to revise your manuscript so that it conforms to our Article format instructions, available

[here](http://www.nature.com/ng/authors/article_types/index.html).

*3) Include a revised version of any required Reporting Summary: <https://www.nature.com/documents/nr-reporting-summary.pdf>

Please be aware of our [guidelines](https://www.nature.com/nature-research/editorial-policies/image-integrity) on digital image standards.

EXTENDED DATA FIGURES

Link Redacted

We hope to receive your revised manuscript within four to eight weeks. If you cannot send it within this time, please let us know.

Sincerely,

Michael Fletcher, PhD
Senior Editor, Nature Genetics
ORCID: 0000-0003-1589-7087

Reviewers' Comments:

Reviewer #1 (Remarks to the Author):

The revised manuscript from Liu et al does a thorough job of addressing the two main points raised in review, reproducibility and batch effects. With respect to reproducibility the inclusion of a new experiment (batch 9) and the analysis of TFs shared between batches provides enough data for readers to assess the reproducibility of the technique. With respect to batch effects, the revised Figure 2C shows clearly that most of the major effects are observable in multiple batches. I also appreciated the attempt to include batch-specific scaling factors to deal with more subtle effects. I thank the authors for providing such a clear and detailed response to the critiques.

Reviewer #2 (Remarks to the Author):

Deplanke and colleagues have done an excellent job of revising the manuscript. This paper represents an enormous amount of work. The authors are to be applauded on a careful and thought-provoking study. I also appreciate the detailed response and yellow marking of changes. Note that I previously co-reviewed this with a trainee who has since moved on to bigger and better things, so I apologize if this feels like a relatively "fresh take" (I had read it previously and edited/appended their review, but I'm taking more of the lead on the re-review). My new comments (some of which I imagine could have been raised originally, but others of which were prompted by certain revisions) are offered in a constructive spirit in hopes of helping you further improve the manuscript, and none of the suggestions below are "deal breakers" from my perspective. But particularly the ones listed under "Major", I think are relatively straightforward and would add considerable value for many reviewers if addressed, even if briefly.

Major

- Legends need careful review for completeness. Various axis labels are also missing. As one example looking at Fig 2c and that dotted red box and no discussion in legend re: what it is. I strongly suggest a full review of your figures/legends for completeness w/r/t labeling of axes and explaining everything in each figure in the corresponding legend, not just in the text.

- I'm sorry if I missed this somewhere but I feel like the lack of a clear "key" for evaluating the extent to which the distribution of expression of exogenous TFs overlaps (or doesn't) w/ endogenous TF expression levels in the context of these differentiations feels really missing. As you know, one of the main concerns re: TF overexpression schemes is that the doses used are non-physiologic. What part of the range of each TF corresponds to physiologic here? I don't want be too prescriptive about how you do this, but it really does feel missing now. Can you either look at the levels of key TFs (e.g. RUNX, etc.) that are known to play roles in certain cell types and ask how their levels look in the context of cocktail driven differentiation, or in vivo single cell data from mouse? (e.g. calibrating to some housekeeping genes or something like that). Some calibration for the reader of what "regime" we are in here would be really helpful.

- On a related point, the analysis in Supp Fig 1.1H is very helpful, but I am confused about what the y-axis units are, and the claim that this is less variation than seen w/ the exogenous TFs doesn't really feel well supported by this histogram on its own. Is there some way of representing this information so we can compare both the variation and the mean levels of the TFs' exogenous vs. endogenous expression? (again, more interested in what the TFs' expression levels are like during natural differentiation along a process like adipogenesis, than in their baseline levels in MSCs/ESCs).

- Fig 2B lacks explicit labelling of which cells correspond to "cocktail controls" (i.e. differentiated w/ chemical cocktail) so I can't really judge the claim that certain clusters also have those cells. What I would really like is for you to make separate UMAPs (fine if they are supplementary) for each of the four major differentiated cell types (e.g. adipogenic, osteogenic, etc.), taking only cells from those clusters and labeling them by which TFs they are expressing OR if they are cocktail controls. This would give the reader clarity on how much substructure there is within a given cell type and how much of that is a consequence of the specific TF (or chemical cocktail). This relates to the Results around lines 230-239 and could be cited there. For example, I'd like to see a UMAP that lets me see if adipocytes expressing MYCN cluster separately from adipocytes expressing CEBPA, PPARG, RHOX12 or adipocytes generated via the chemical cocktail. This is hard to discern in a composite UMAP of the entire dataset.

- The material around lines 270 and 300 are related (i.e. what are the features of TFs in the three classes) and should be consolidated to the same part of the paper. Also, readers will be highly interested in knowing if particular kinds of TFs (e.g. ZFs, homeodomain TFs) are enriched/depleted in the three TF groupings. Can you add a quick enrichment analysis along these lines? For example, I imagine homeodomain TFs might be enriched amongst high-capacity, high-dose sensitive TFs.

Minor

- You are inconsistent and incorrect in your style for naming genes/proteins, to my knowledge. Mouse gene names should be capital first letter only, and italicized, and protein names should be the same but not italicized. Only human genes/proteins are "all caps". Paper and figures are all over the place w/r/t this, and should be made consistent/correct. I think the only place you should be using caps is when you are referencing literature that concerns human orthologs of mouse genes? (e.g. Mycn (gene) and Mycn (protein), not MYCN; sorry to be the nomenclature police!)

- Colors in many figures, particularly yellow tones/text, are really hard to see for me. Suggest careful review w/ an eye to optimal color schemes / consideration for the color blind.

- Text clearer but Fig 1A still makes it look like this is pooled packaging and transduction. Suggest making it more consistent w/ the arrayed format in which you did it.

- Early in Results or in Discussion, can you comment on the *source* of the variation in TF expression? I presume that it is just copy number variation on the high MOI transductions, coupled with expression variegation on whatever gets in, but it would be nice to see that stated (assuming I have that right)

- I suggest being more careful, and potentially just avoiding altogether, the term "branching". I felt a bit confused in those parts of the paper re: the difference between branching, vs. non-monotonic behaviors that might be better characterized as 'nesting'. To me branching implies two mutually exclusive fates, which I recognize you think you are seeing for certain findings, but the UMAPs imply a continuity of states, one of which may simply precede the other (nesting?), with higher doses simply driving the cells further/faster. Not asking for any new work or analyses here, just a careful review of your language, perhaps more careful articulation of what branching means to you, and which examples are branching vs. simply non-monotonic.

- Very optional: I would have appreciated a brief analysis re: whether the TFs that are implicated w/ a particular cell type are upregulated in that cell type when other TFs are used to induce it and/or with the chemical cocktail. In other words, is their a network among the TFs that is self-reinforcing, and any one of the TFs can be used to drive it (in which case not on the other TFs but also the endogenous TF might be upregulated?).

Version 2:

Decision Letter:

Our ref: NG-A64657R1

15th Apr 2025

Dear Dr. Deplancke,

hope you're doing well.

Thank you for submitting your revised manuscript "Dissecting the impact of transcription factor dose on cell reprogramming heterogeneity using scTF-seq" (NG-A64657R1). It has now been seen by the original referees and their comments are below. The reviewers find that the paper has improved in revision, and therefore we'll be happy in principle to publish it in Nature Genetics, pending minor revisions to comply with our editorial and formatting guidelines.

Thank you again for your interest in Nature Genetics. Please do not hesitate to contact me if you have any questions.

Congratulations!

Sincerely,
Chiara

Chiara Anania, PhD
Associate Editor
Nature Genetics
<https://orcid.org/0000-0003-1549-4157>

Reviewer #2 (Remarks to the Author):

The authors have addressed my remaining comments to my full satisfaction. I appreciate their care and attention in doing so. Congratulations on the terrific work and I look forward to seeing it published!

Rebuttal Manuscript #NG-A64657:

“Dissecting the impact of transcription factor dose on cell reprogramming heterogeneity using scTF-seq”

Liu, Saelens*, Rainer* et al.*

Summary of the Reviews and our Responses

We sincerely thank the reviewers for their valuable comments, which greatly contributed to further enhancing our manuscript and rigorously evaluating our scTF-seq methodology, particularly regarding best practices, data quality & robustness, and the reproducibility of our key findings.

To address the primary concerns raised by both reviewers, i.e. those on batch effects in scTF-seq datasets and the reproducibility of the scTF-seq assay, we conducted a new scTF-seq experiment (batch 9). This experiment generated another 7,059 high-quality cells replicating 14 TFs and controls, increasing our total cell count to 45,978. Although our original data already demonstrated considerable overlap between TFs that were shared across different batches, adding this new dataset enabled us to thoroughly assess the reproducibility of the scTF-seq assay, which we have integrated throughout the manuscript, as detailed below.

To further support our findings, we also performed orthogonal multiplex RNA in situ hybridization (RNAscope) experiments, which validated dose dependency and (non-monotonic) dose-response curves. These RNAscope results are included in revised **Figure 4h, j** and **Supplementary Figure 4.2j, k**.

Additionally, Reviewer #2 pointed out that the dose range for some TFs may not be wide enough to accurately fit dose-response curves. This concern is particularly relevant for low-capacity TFs which could in fact be high-capacity if cells that express these TFs at a sufficiently high dose would be available for analysis. We addressed this by including a power analysis (added in revised **Supplementary Figure 3f-h**) showing that 77% of the TFs are predicted to be correctly classified as ‘low-capacity TFs’. We thereby note that our study is to our knowledge the first to achieve such a wide dose range, enabling the detailed mapping of phenomena such as non-monotonic responses. This contrasts with the lower TF doses and the more binary responses reported in a recent study by Joung et al. 2023¹, which also did not focus on TF dose as a read-out). The comparison has been incorporated into the revised **Supplementary Figure 1.2a-d**.

After incorporating the new data and analyses, we revised the figures, supplementary figures, and manuscript text to improve clarity and rigor. Due to space limitations, we could not include all rebuttal figures in the manuscript. However, we selected key

examples from those figures and compiled them into three new supplementary figures (revised **Supplementary Figures 1.2, 1.3, and 4.3**). We also detailed in the rebuttal letter which figures and panels have been added to the manuscript, providing clear guidance on the nature of our manuscript revision. Finally, we are in favor of making the complete rebuttal letter and figures publicly available to provide additional clarity.

Reviewer #1:

Remarks to the Author:

This manuscript introduces scTF-seq, a method for quantifying single-cell transcriptional changes induced by the overexpression of transcription factors (TFs). Using scTF-seq, the authors produce an overexpression atlas for 384 TFs in mouse embryonic multipotent stromal cells (MSCs), identifying key TFs for MSC lineage differentiation. The paper explores dose-dependent effects of TFs and classifies TFs into four categories based on their reprogramming capacity and their sensitivity to doses. Finally, the paper examines transcriptional changes caused by pairs of TFs, revealing that the combined effect of two TFs is influenced by the relative dose of the two TFs.

Response 1:

We thank Reviewer 1 for the succinct and accurate summary of our study.

Overall, the data in this study do not support the claims being made. The primary limitation of the study is that the scTF-seq experiment was only performed once, making it difficult to assess the reproducibility of the method or estimate error bars for the measurements. This is an important point because many of the claims are based on observations that are subtle (e.g. the clustering in Figs 4C and 4F) and could be within the margin of error.

Another issue is that the TFs are assayed in 8 separate batches. The authors appropriately perform batch correction, but some evidence is needed to show that batch correction is indeed working. Could the authors show that the clustering of TF functional modules and the classification of reprogramming capacity/dose-sensitivity are not influenced by batch effects? In Figure 1e, it appears that certain batches (e.g. the pink batch) exhibit distinct transcriptional patterns on the UMAP plot. The authors should clarify whether these batch-specific patterns are due to enrichment of certain TF groups in one batch, or due to batch artifacts. A related point is that the technique uses UMI counts of TF-IDs to quantify TF doses. UMI counts can vary across different batches because of variations in 10x capture efficiency and PCR efficiency.

Response 2:

We thank Reviewer 1 for raising these insightful questions on reproducibility and batch effects, and we apologize for the lack of clarity in our initial submission. Since batch effects impact data reproducibility, we decided to address these concerns together.

First, as the reviewer pointed out, our initial submission included data from *eight* batches representing *eight separate* scTF-seq experiments (now increased to nine). We ensured reproducibility between these experiments by including negative controls (mCherry-overexpressing cells) and positive controls (cells induced for differentiation using for example an adipogenic cocktail). Furthermore, in every experiment, at least six TFs, typically more, were also included in other experiments (**Rebuttal Figure 1**), meaning that these constitute true **biological replicates**. Although we have shown the batch information in **Supplementary Table 1**, we acknowledge that we may not

have highlighted this sufficiently in the previous manuscript, and we now detail this in **lines 120-121** and **136-139**.

Second, since these biological replicates were assayed in separate batches, there are indeed batch effects that need to be corrected and reproducibility needs to be assessed across batches.

Rebuttal Figure 1. Overlap in high-capacity (lower triangle) and all (upper triangle) TFs between batches.

In our revision, we now provide clear evidence that we perform adequate batch correction, and further confirm the reproducibility of our findings by (1) performing a large, new scTF-seq experiment to validate dose-dependent and stochastic cell-fate branching, as well as the capacity and dose sensitivity of several TFs, (2) including several analyses that assess the reproducibility across biological replicates, particularly focusing on the observations that are more subtle (as highlighted by the reviewer), and (3) validating the observed dose dependency using an orthogonal RNAscope assay. We also assess the “error bars” of our main findings, namely the dose sensitivity and TF classification. In particular, we find that the dose-response is very consistent, even in the case where only a few dozen cells per TF are analyzed. As a result, the TF classification is also reproducible. That said, our new analyses did reveal that the efficiency of dose profiling can be different between batches. Therefore, we now propose a rather straightforward technique to correct this (see below).

To demonstrate that our findings are independently reproducible, we generated a new scTF-seq experiment (“batch 9”) for a subset of TFs that are key to our original/main findings:

Rebuttal Table 1: The list of TFs and the purpose of including them in batch 9.

TF	Aim

ZFP692	Validate inflammatory TFs
ZFP24	
KLF4	Validate cell fate branching TFs
ESR2	
RUNX2	
ETV1	
EGR1	
GRHL2	
MEIS2	
IRF3	Validate dose response (high capacity & dose insensitivity)
HOXA6	Validate dose response (high capacity & high dose sensitivity)
FOS	Validate dose response (high capacity & low dose sensitivity)
POU5F1	
VDR	Validate dose response (low capacity)

An average of 441 cells per TF were captured in this experiment, ensuring robust statistical power. We integrated the data from this experiment and the original batches 1-8, and thoroughly analyzed the reproducibility of our methodology and conclusions, as described in detail below.

1) The batch effect on cell transcriptomes is well corrected.

To clearly show the batch correction, we have now added the UMAP plot of the batch-uncorrected TF atlas in the revised **Supplementary Figure 1.1f (Rebuttal Figure 2 left)**. Comparison of this UMAP plot to revised **Figure 1e (Rebuttal Figure 2 right)** shows a successful batch correction on the cell transcriptomes given that control or TF cells from different batches are well mixed together.

Rebuttal Figure 2, added in revised Supplementary Figure 1f and Figure 1e. UMAP plots of the batch-uncorrected (left) or batch-corrected (right) scTF-seq atlas colored by experimental batch.

The reviewer correctly deduced that in the original Figure 1e (also in **Rebuttal Figure 2 right**, with batch 9 added), the transcriptomic pattern of the pink batch (batch 8) reflects the enrichment of certain TF groups in that batch. In particular, in that final experiment of our initial submission, we specifically analyzed a large number of cells for specific TFs, including RUNX2, CEBPA, MYOG, and MYCN, which exhibited strong reprogramming effects in batch 1-7 (**Rebuttal Figure 3**). This allowed us to establish a robust baseline of the main lineage effects in the data. However, despite this imbalance in cell numbers, the cells from these TFs clearly cluster together with their respective cell counterparts from different batches (**Rebuttal Figure 3**), indicating that the batch effect is well corrected. As explained later, the more subtle differences are mostly attributable to dose differences.

Rebuttal Figure 3, related to Supplementary Figure 1.3a. UMAP plots of the scTF-seq atlas colored by RUNX2 (top) or MYOG (bottom)-overexpressing cells in different experimental

batches. The number in parentheses indicates the number of cells overexpressing the specific TF per batch.

In addition to these lineage TFs, we found that the reprogramming effects of TFs are in general very well conserved at the single-cell level across biological replicates. On the UMAP plot, cells linked to the same TFs tend to cluster together and away from other TFs, even when these TFs were processed in the same batch (e.g. RUNX2 in batches 5, 6, 8, and 9; KLF4 and HOXA6 in batches 7 and 9; ESR2 and ETV1 in batches 3 and 9, **Rebuttal Figure 4**). The main heterogeneity arises from differences in dose levels, which, we observed, can shift in distribution between biological replicates. For example, RUNX2 had only a low dose in batch 5, while KLF4 reached a higher maximal dose in batch 9 (**Rebuttal Figure 4**). Still, the cells with similar dose levels from different batches clearly cluster together. As such, the differences in dose reflect true biological variation and not a batch effect.

We now include a new panel in revised **Supplementary Figure 1.3a** that summarizes the information from **Rebuttal Figures 3** and **4**.

Rebuttal Figure 4, related to revised **Supplementary Figure 1.3a**. UMAP plots of the scTF-seq atlas colored by *Runx2* (top), *Klf4* (mid left), *Hoxa6* (mid right), *Esv2* (bottom left), or *Etv1* (bottom right) dose across different experimental batches. The red arrows highlight the main

cluster of cells where the TF had a discernible effect (often because of high dose). The number in parentheses indicates the number of cells overexpressing the specific TF per batch.

We can further demonstrate that the clustering of functional TF modules is primarily driven by gene expression similarities among TF biological replicates or homologs, rather than batch artifacts. This clustering is already apparent prior to batch correction (**Rebuttal Figure 5 left**), but becomes even stronger after correction, as shown in **Rebuttal Figure 5 right / Figure 2c**. We revised **Figure 2c** (the right panel in **Rebuttal Figure 5**) to incorporate data from batch 9. Due to space constraints, we did not include the left panel of **Rebuttal Figure 5** in the figures or supplementary figures.

Rebuttal Figure 5, related to revised Figure 2c. Heatmap showing a pairwise Pearson correlation of functional TF cells from the unintegrated (**left**) or integrated (**right**) atlas. Cells are annotated by TF (in column) and batch (in row) and ordered by hierarchical clustering.

To assess the reproducibility of all TFs, we created a pseudobulk expression measurement for each TF in each batch and visualized these in a reduced space (**Rebuttal Figure 6**). We then verified if TFs that were sequenced in multiple batches overlap in that space. For the vast majority of these TFs, the biological replicates clearly cluster together (**Rebuttal Figure 6**), indicating that the reprogramming effects are reproducible. In the revised manuscript, we have now included the pseudobulk results from TFs overlapping in at least 4 batches (revised **Supplementary Figure 1.3e**, top row of **Rebuttal Figure 6**).

*Rebuttal Figure 6, related to revised Supplementary Figure 1.3e. UMAP plots of pseudobulk expression for all TF-batch combinations. Highlighted are the TFs that were processed in at least two batches. Part of this figure is now included in the revised manuscript (revised **Supplementary Fig. 1.3**).*

For a small subset of TFs (8/29), one or more replicates do not cluster together. This is primarily caused by differences in dose across batches and/or low TF capacity (**Rebuttal Figure 6**). For example, one replicate of RUNX2 has a lower dose than the other replicates and given the high dose dependency of RUNX2 (including many non-monotonic effects), it is expected that a low-dose batch will cluster separately. Similar observations can be made for RHOX12, GATA4, and PPARG. Conversely, the position of low-capacity TFs, such as BHLHE40, ZFP24, and ZFAND5, is expected to vary between batches because they have minimal reprogramming effects (i.e. low-capacity). For this reason, we revised our conclusions in the manuscript for ZFP24, since the inflammatory response induced by this TF is, at the end, only observed at a low dose in one batch.

This analysis highlights that our findings are very reproducible when a sufficiently high dose (which varies from TF to TF) is reached. The challenge lies in functionally interpreting the observations for TFs that may not have reached a sufficient dose to

fully capture their effect. These TFs may for example have a higher probability of being falsely assigned to the low-capacity TF group. To highlight this, we have now added a paragraph (lines **274-283**) discussing potential false negatives caused by insufficient dose in the revised manuscript (revised **Supplementary Figure 1.2a-d**), and further elaborated below in our response to another comment by the Reviewer (**pages 31-32**). In short, we find that, the predicted probability of correctly classifying the TFs (at max-dose > 3.5) as low-capacity is 77%. That means that about 8 out of 10 low-capacity TFs were correctly classified as such.

2) Differential gene expression results are reproducible across replicates.

While the replicates cluster together, it may be that the behavior of individual (target) genes is still batch-dependent. To test this, we compared the differentially expressed genes (comparing functional TF cells with mCherry control cells) across replicates for the top-replicated TFs above. We found that the differential expression results are remarkably consistent across replicates for all genes (**Rebuttal Figure 7a-d**). Only in cases where there was a considerable difference in dose between batches (e.g. RUNX2 batch 5 vs batch 9, HEY1 batch 3 vs the rest, ZFP692, **Rebuttal Figure 7d-g**), the R^2 between fold changes across batches decreased below 0.8 (**Rebuttal Figure 7f**), which was also the case for low-capacity TFs, as perhaps could be expected. We selected FOS and ZFP692 as representative examples and incorporated the panels **Rebuttal Figure 7a, f, g** into the revised **Supplementary Figure 1.3b-d**.

Rebuttal Figure 7. Correspondence of differential gene expression across batches, related to revised Supplementary Figure 1.3b-d. (a-d) Fold changes for all genes between control cells and TF-overexpressing cells. Shown are the corresponding R^2 values for the differentially expressed genes (in black). (e) UMAP plots of the scTF-seq atlas colored by *Hey1* dose across different experimental batches. The red arrows highlight the main cluster of cells where the TF had a discernible effect (often because of high dose). The number in parentheses indicates the number of cells overexpressing the specific TF per batch. (f) Distribution of R^2 's of all high-capacity TFs shared between batches, with the TFs with the lowest R^2 highlighted. (g) Same as for (e) but now for the TF ZFP692.

3) Dose-response curves are consistent across replicates.

We next examined whether dose-response curves of either individual genes or overall transcriptomic changes are consistent across replicates. As exemplified by FOS in **Rebuttal Figure 8**, the dose curves align very well for most differentially expressed genes downstream of FOS, including the genes with non-monotonic responses (such as *Cav1*, *Cav2*, and *Gchfr*).

Rebuttal Figure 8. (Top) Dose-response curves for the top differentially expressed genes downstream of FOS. (Bottom) Heatmap plots showing the dose-response dynamics for the top 500 differentially expressed genes downstream of FOS in each batch.

Similarly, the overall transcriptomic dose responses align well for most TFs. This is illustrated by examples such as HOXA6, PAX9, and RUNX2 (high-capacity & high dose-sensitive TFs), POU5F1, FOS, and EGR1 (high-capacity & low dose-sensitive TFs), and VDR and HMGB3 (low-capacity TFs), together covering all batches (Rebuttal Figure 9).

Rebuttal Figure 9. Dose response curves colored by batch, reflecting the overall transcriptomic changes in function of TF dose for the indicated TFs.

Although **Rebuttal Figures 8 and 9** were not included in the revised main or supplementary figures, we have added dose-response curves after dose alignment for representative FOS genes into revised **Supplementary Figure 1.3f**, as detailed below.

4) Batch effect on TF dose is corrected by dose alignment.

While the previous analyses showed that the dose responses are well conserved across batches, there may still be subtle differences in dose capturing efficiency that can only be detected using a quantitative analysis across multiple genes. As suggested by the reviewer, such a subtle quantitative shift may be caused by for example differences in capture efficiencies between different 10x runs. To assess whether such a shift exists and whether it is shared among TFs, we focused on batches 3, 4, and (newly generated) 9, since these batches collectively include 9 high-capacity TFs that are represented in at least two of the batches. High-capacity TFs are selected here because they have dozens, if not hundreds of differentially expressed genes, which allows us to accurately align the dose ranges across batches. For each gene across different batches, we calculated how well the dose-response curves would align if we applied a batch-specific scaling factor to the dose (**Rebuttal Figure 10**). Please note that because of space constraints, **Rebuttal Figure 10** was not included in the main figures or supplementary materials.

Rebuttal Figure 10. Dose-dependent expression kinetic of *Cav1* in two batches (batch 3 green, batch 9 red) depending on *Fos* dose. By calculating the Euclidean distances between these dose-response curves for various scaling factors and averaging across genes, we can determine the optimal scaling factor for a given TF-batch pair.

By averaging this across all genes, we can robustly determine the appropriate scaling factor for a TF. We found that the optimal factor is very consistent across high-capacity TFs, ranging between 0.72-0.83 depending on the TF (**Rebuttal Figure 11a-c**, included in the revised manuscript as **Supplementary Figure 1.3g-i**), indicating that this is likely caused by a common technical effect per batch, e.g. a difference in TF-ID capture rate.

Rebuttal Figure 11. Aligning dose response curves between batches 3-4 and 9 using overlapping TFs, added in revised **Supplementary Figure 1.3g-i**. (a-b) Average Euclidean distance between dose-response curves for all variable genes at different scaling parameters.

The scaling factor scales the dose of batch 9 to match the dose-response curves of batches 3-4. (c) Histogram of the most optimal scaling factor for all 9 TFs.

The fact that the scaling factor slightly differs from 1 is an important consideration when comparing doses across batches, and we thank the reviewer for bringing this to our attention. We have now included a paragraph in the Results section of our revised manuscript (**lines 159-166**) to highlight this, advising the readers who would use scTF-seq that ideally a subset of dose-sensitive TFs should be included in each batch as “positive controls” to accurately map the optimal scaling factor. We suggest that about 5-10 TFs should be shared between batches, to ensure that enough TFs with overlapping high doses end up in the final datasets. These positive controls can also be used for other purposes, i.e. to assess batch correction across datasets as we have done in aforementioned analyses.

We implemented dose alignment using the calculated scaling factor and applied it to all downstream analyses. Unless specified, all TF doses in the subsequent rebuttal letter, rebuttal figures, and the revised manuscript are doses after alignment. Nevertheless, since the optimal scaling factor is either equal to 1 or close to 1 (e.g., 0.79), its impact on the dose-response curve is still relatively subtle, as demonstrated by the top differentially expressed genes for FOS and the overall transcriptomic changes of POU5F1 and EGR1 (**Rebuttal Figures 12 and 13**).

Rebuttal Figure 12, related to revised Supplementary Figure 1.3f. Dose-response curves for top differentially expressed genes downstream of FOS before and after dose alignment between batches.

Rebuttal Figure 13. Dose-response curves colored by batch, reflecting the overall transcriptomic changes in function of TF dose or aligned TF dose for the indicated TFs.

Consequently, the downstream categorization of TFs is reproducible, especially for high-capacity TFs (**Rebuttal Figure 13**). In contrast, low-capacity TF classification is more sensitive to variation in dose levels between batches (i.e. a TF might not reach a sufficiently high dose in one batch to achieve its max capacity/effect). For this reason, we integrated TF cells from different replicates to enhance overall power and robustness. Furthermore, to ensure a sufficient dose range for reliable dose-response fitting, we excluded 21 TFs with fewer than three cells in the lowest dose bin (dose < 1.68) and 33 TFs lacking a sufficient maximum dose (3.5) before applying the logistic model (as described in the revised **Methods** section).

Leveraging the new data and insights detailed above, we decided to revisit the TF classification from our initial submission. This revealed that, in the end, there is hardly any dose independence, consistent with the notion that even a small TF dose alteration can be sufficient to modulate gene expression. Therefore, we decided to merge the previously designated 'high-capacity & dose-insensitive' class into the 'high-capacity & high dose-sensitive' one. **Figure 3, Supplementary Figure 3,** and the manuscript (**lines 284-294** and **500-508**) have been revised accordingly.

We included dose-response curves after dose alignment for representative FOS genes (shown on the first two rows of **Rebuttal Figure 12**) in the revised **Supplementary Figure 1.3f**. Due to space constraints, we did not include **Rebuttal Figure 13** in the main or supplementary figures, opting to show the alignment of individual genes instead.

5) Dose-dependent and stochastic cell fate branching shows consistency across replicates.

In our original manuscript, we reported cell fate branching for several TFs initially based on the large, integrated atlas comprising all TFs in which cells are clustered together based on their transcriptomic similarities (**Rebuttal Figures 14, 15 / Supplementary Figures 4.1b-d**). Although we focused on functional TF cells for better clustering, potentially overlooking some branches, this clustering, which incorporates cells from many TFs, strengthens the reliability of the observed cell branching populations.

Rebuttal Figure 14, added in revised Supplementary Figure 4.1b-c. (Left) UMAP plot of the integrated scTF-atlas, including control and functional TF cells in adjusted-G1 phase (referred to as the 'phase-adjusted functional TF atlas'), colored by cluster. (Right) Heatmap showing the proportion of cells in each cluster per TF relative to the total number of cells for that TF, indicating the occurrence of two or more cell states per TF. Ctr.conf, confluent control cells; Ctr.non.conf, non-confluent control cells.

Rebuttal Figure 15, added in revised Supplementary Figure 4.1d. UMAP plots of the phase-adjusted functional scTF-seq atlas with KLF4, RUNX2, ETV1, EGR1, GRHL2, ESR2, MEIS2, or MYOG cells highlighted (red for dose-dependent TFs, green for stochastic-branching TFs).

Nevertheless, we agree with the reviewer’s point that, in our initial submission, several ‘cell fate branching’ events (and linked TFs) were observed in only one batch with relatively small sampling sizes. Consequently, this might limit the representativeness of actual cell populations and TF doses, potentially constraining our ability to discern dose-dependent from stochastic branching. To address this, we designed experimental batch 9 to specifically target a large number of cells for ‘cell fate branching’ TFs (**Rebuttal Table 1**). We then performed a detailed analysis of the branch-inducing capability and dose dependency of each TF, thus revisiting our original classification.

As demonstrated in **Rebuttal Figures 16-24** / revised **Figure 4c-g**, **Supplementary Figures 4.1e, f** and **4.2a-h**, TF cells from batch 9 are well integrated with cells from prior batches across all clusters. Alongside the previously observed dose-sensitive cell states induced by KLF4, ESR2, and ETV1, the higher dose achieved in batch 9 has even enabled the emergence of additional cell states (**Rebuttal Figures 16-18** and **20** / revised **Figure 4c-g**, **Supplementary Figure 4.2a, b, g, h**). As for RUNX2, both monotonic and non-monotonic gene expression patterns regulated by low and high *Runx2* doses remained consistent across batches (**Rebuttal Figure 19** / revised **Supplementary Figure 4.1e, f**). As for EGR1, while the clustering analysis lacks the resolution needed to capture this TF as a branching candidate, a closer examination of EGR1 cells shows that two distinct cell states induced by varying doses are robust across batches (**Rebuttal Figure 21** / revised **Supplementary Figure 4.2c, d**). With these findings, we can confirm that all observed dose-dependent branching events for TFs including KLF4, ESR2, ETV1, RUNX2, and EGR1 are robust and reproducible.

Rebuttal Figure 16, added in revised

Figure 4c-e. (Left) UMAP plots of KLF4 cells and their batch-paired control cells colored by categories, batch, Klf4 dose, or cluster. Clusters composed of more than 40% of control cells were merged together as controls (Ctl.s). (Right) Violin plot showing the dose distribution of KLF4 cells across the clusters. Cell counts: $n = 583$ for KLF4 cells, $n = 226$ for non-confluent control cells (Ctr.non.conf), $n = 404$ for confluent control cells (Ctr.conf).

Rebuttal Figure 17, added in revised Figure 4g. Top 10 uniquely enriched GO terms based on the top differentially expressed genes of the three clusters enriched among KLF4 cells (see Rebuttal Figure 16).

Rebuttal Figure 18, added in revised Supplementary Figure 4.2g, h. (Left) UMAP plots of ESR2 cells and their batch-paired control cells colored by categories, batch, ESR2

dose, or cluster. Clusters composed of control cells were merged together and colored in gray. (Right) Violin plot showing the dose distribution of ESR2 cells across the clusters. Cell counts: $n = 439$ for ESR2 cells, $n = 176$ for non-confluent control cells (Ctr.non.conf), $n = 498$ for confluent control cells (Ctr.conf).

Rebuttal Figure 19, added in revised Supplementary Figure 4.1e, f. (Top) UMAP plots of RUNX2 cells and their batch-paired control cells colored by dose, clustering, and experimental batch. Clusters composed of control cells were merged together.

(Bottom) Heatmap displaying log-normalized expression (Z-score scaled by gene) of differentially expressed genes of high and low RUNX2 and control cells. Cells are ordered by Runx2 dose as indicated by the color bar on the left. Cell counts: $n = 2332$ for Runx2 cells, $n = 650$ for control cells including both non-confluent and confluent mCherry overexpressing cells.

Rebuttal Figure 20, added in revised Supplementary Figure 4.2a, b. (Left) UMAP plots of ETV1 cells and their batch-paired control cells colored by categories, batch, ETV1 dose, or cluster.

Clusters composed of control cells were merged together and colored in gray. (Right) Violin plot showing the dose distribution of ETV1 cells across the clusters. Cell counts: $n = 662$ for ESR2 cells, $n = 176$ for non-confluent control cells (Ctr.non.conf), $n = 498$ for confluent control cells (Ctr.conf).

Rebuttal Figure 21, added in Supplementary Figure 4.2c, d. (Left) UMAP plots of EGR1 cells and their batch-paired control cells colored by categories, batch, EGR1 dose, or cluster. Clusters composed of control cells were merged together and colored in gray.

(Right) Violin plot showing the dose distribution of EGR1 cells across the clusters. Cell counts: $n = 418$ for EGR1 cells, $n = 176$ for non-confluent control cells (Ctr.non.conf), $n = 498$ for confluent control cells (Ctr.conf).

Next to dose-dependent branching, in our original manuscript, we reported that for some TFs such as for example **MYOG**, **MEIS2**, and **GRHL2**, TF dose alone is not sufficient to rationalize the cell fate branching effects.

MYOG: Our previous data have already shown consistent branching in MYOG cells across batches (as demonstrated by well-integrated MYOG cells from different batches in individual clusters in **Rebuttal Figure 22** / revised **Supplementary Figure 4.3a-c**).

Rebuttal Figure 22, added in revised Supplementary Figure 4.3a-c. (a, b) UMAP plots of MYOG cells and their batch-paired control cells colored by categories, batch, cluster, or Myog dose. Ctr.non.conf, non-confluent mCherry overexpressing cells. Ctr.conf, confluent mCherry overexpressing cells. Clusters (Ctr) mostly composed of control cells were merged and colored in gray. Cell counts: $n = 450$ for MYOG cells, $n = 442$ for non-confluent control cells (Ctr.non.conf), $n = 629$ for confluent control cells (Ctr.conf) (c) Violin plot showing the dose distribution of MYOG cells in individual clusters in b.

MEIS2: Leveraging the new scTF-seq experiment (batch 9), we could further validate stochastic branching for MEIS2 (**Rebuttal Figure 23** / revised **Figure 4k-m** and **Supplementary Figure 4.3d-I**). Moreover, we were able to obtain higher doses for MEIS2 in batch 9 compared to prior batches, which even revealed new cell states (such as cluster 5), indicating that MEIS2 also induces dose-dependent cell fate branching. Interestingly, however, clusters 2 and 3 are enriched for distinct gene expression modules, even though the dose difference between them is minimal. In fact, by calculating gene expression scores of the same modules from batch-integrated data (module scores) on MEIS2 cells of individual batches, we found that the clusters that have high scores for respectively module 2 or 3 (i.e. clusters 0 and 2 in individual batch 4 and also in batch 9) exhibit opposite dose variations across the two batches (i.e. the *Meis2* dose is higher for cluster 0 compared to cluster 2 in batch 9, while the opposite is true in batch 4), effectively canceling each other out. Thus, our data are consistent with our original observations, namely that at a low/mild *Meis2* dose (dose < 5), factors other than TF dose likely contribute to the emergence of distinct MEIS2-driven cell states, reflecting stochastic branching.

Rebuttal Figure 23, added in revised Figure 4k-m and Supplementary Figure 4.3d-l. (a) UMAP plots of MEIS2 cells and their batch-paired control cells colored by categories, batch, Meis2 dose, or cluster. Ctr.non.conf, non-confluent mCherry overexpressing cells. Ctr.conf, confluent mCherry overexpressing cells. Clusters (Ctl) composed of more than 40% control cells were merged and colored in gray. Cell counts: $n = 783$ for MEIS2 cells, $n = 176$ for non-confluent control cells (Ctr.non.conf), $n = 498$ for confluent control cells (Ctr.conf). (b) Violin plot showing the Meis2 dose distribution of MEIS2 cells across the clusters. NS, not significant; p_{adj} , FDR-adjusted p -value; pairwise two-sided t -test followed by FDR correction. (c, d) Dose-response curves of all module scores (c) or individual module scores (d) of all MEIS2 cells. Module scores were calculated based on the expression of differentially expressed genes of individual clusters on a. (e, f) (Top left pair): UMAP plots generated by independent analysis on MEIS2 cells from each individual batch (e for batch 4 and f for batch 9) and colored by dose or cluster. (Top right for each e and f panel): violin plot (showing the dose distribution of corresponding clusters). NS, not significant; pairwise two-sided t -test followed by FDR correction. All other adjusted p values that were not indicated are < 0.0002 . Middle/bottom panels: UMAP plots of MEIS2 cells in batch 4 (e) or batch 9 (f) colored by scores of the modules identified using the batch-integrated data (a-d).

GRHL2: For GRHL2, the three non-control clusters shown in our original submission (without batch 9) remained distinguishable after integrating cells from batch 9, as demonstrated by the clustering in **Rebuttal Figure 24a** versus **d** and the conserved expression patterns of the previously identified differentially expressed genes of each cluster (**Rebuttal Figure 24g**). However, likely due to the increased sampling rate (77 cells originally, 356 cells now), the dose difference between clusters 1 and 2, although subtle and still showing a large overlap in the distribution, is now statistically significant. In light of these new observations, we decided to assign GRHL2 to the 'dose-dependent branching' TF category.

Because our data on GRHL2 now better align with dose-dependent branching, while MEIS2 still shows clear reproducible stochastic branching, we opted to highlight MEIS2 in the revised main manuscript, and GRHL2 in revised **Supplementary Figure 4.2e, f**.

Rebuttal Figure 24, related to revised Supplementary Figure 4.2e, f. (a) UMAP plot showing individual GRHL2 clusters which were identified in our original manuscript. Ctr, controls that were merged from clusters mostly composed of mCherry-overexpressing control cells. (b-e) UMAP plots of GRHL2 cells and their batch-paired control cells colored by categories, batch, cluster, or *Grhl2* dose. Clusters mostly composed of control cells were labeled as Ctr. Cell counts: $n = 356$ for GRHL2 cells, $n = 176$ for non-confluent control cells (Ctr.non.conf), $n = 498$ for confluent control cells (Ctr.conf). (f) Violin plot showing the *Grhl2* dose distribution of GRHL2 cells across the clusters in d. (g) UMAP plots of GRHL2 cells colored by the normalized expression level of differentially expressed genes of GRHL2 clusters.

Based on **Rebuttal Figures 14-24**, we have revised **Figure 4** and **Supplementary Figures 4.1** and **4.2** and organized a new **Supplementary Figure 4.3**.

6) Dose-dependency can be validated using an orthogonal assay.

Next to scTF-seq data replication, we validated our findings on dose-dependent cell fate branching by using an orthogonal assay, namely multiplex RNA in situ hybridization (RNAscope). While this approach offers exceptional sensitivity and specificity, it is limited by low throughput, as it can detect only up to four RNA targets simultaneously. We thus focused on two representative dose-dependent branching TFs: KLF4 and ESR2. For each of them, two marker genes specific to the observed branching subpopulations were selected (**Rebuttal Figure 25a** / revised **Figure 4i** and **Supplementary Figure 4.2i**). We probed RNA transcripts of the selected marker genes, along with WPRE, a viral element located on the vector adjacent the TF-ID, which serves as a proxy for TF dose. We then quantified fluorescence intensities, reflecting expression levels for each marker or TF dose on a per-cell basis across individual channels. As shown in **Rebuttal Figure 25 b** / revised **Figure 4h, j**, distinct cell populations exhibited mutually exclusive expression of *Glul* and *Postn* in response to low *versus* mild/high *Klf4* overexpression. In addition, the dose-dependent, non-monotonic expression patterns of *Gng12* and *Aspn* in ESR2 cells, initially identified by scTF-seq, were also corroborated by the RNAscope experiments which nicely captured a full range of *Esr2* dose variation (**Rebuttal Figure 25a, c** / revised **Supplementary Figure 4.2i-k**).

Rebuttal Figure 25, added in revised Figure 4h-j and Supplementary Figure 4.2i-k. (a) Log-normalized maker gene expression in control and KLF4 (**left pair**) or control and ESR2 (**right pair**) cells detected by scTF-seq. Cells are colored by clusters as shown in Rebuttal Figures 16 and 18. Ctl refers to control clusters. **(b, c) (Left images)** Representative images of multiplex RNA in situ hybridization (RNAscope) detecting RNA transcripts of WPRE (representing the TF-ID), *Postnr*, and *Glul* in KLF4 cells, or RNA transcripts of WPRE, *Gng12*, and *Aspnr* in ESR2 cells. Wildtype C3H10T1/2 cells were used as the control. Nuclei were stained by DAPI. Scale bar, 100 μ m. All fluorescent channels were merged for cell segmentation, indicated by the red (cell boundary) and purple (expanded cell boundary) outlines. The mean fluorescence intensity was quantified for each segmented cell. **(Right panels)** Scatter plots showing the correlation between the normalized mean fluorescence intensity of each gene and WPRE in TF (KLF4 or ESR2) and wildtype control cells. Cell counts: for KLF4 and control in panel **b**, $n = 713$ and 747 , respectively; for ESR2 and control in panel **c**, $n = 300$ and 481 , respectively.

We believe that the new data generated from both scTF-seq and the orthogonal assay, along with the comprehensive analyses and resulting findings, demonstrate the reproducibility of scTF-seq and strongly support our claims.

An exciting feature of scTF-seq is the potential to discern dose-dependent responses to TFs, because the Dox-inducible promoter drives different levels of expression at different Dox concentrations. However, the authors only expressed TFs at one dox concentration and relied on the natural fluctuations of the Dox promoter to produce different TF doses. The rationale for using 2ug/ml Dox concentration is not given and it is not clear whether the TF levels induced at this Dox concentration are sufficient for effective reprogramming. Also, the range of concentrations produced by natural cell-to-cell fluctuations does not appear to be wide enough to accurately fit the dosage curves in Figure 3c,d and Supplementary Figure 3c. In most cases the dose-response curves did not reach saturation, which makes fitting unreliable.

Response 3:

We appreciate the reviewer's positive feedback on the unique capability of scTF-seq. Different from most of the previously published pooled library strategies, where a low (around 0.1-0.3) multiplicity of infection (MOI) was used to avoid the transduction of multiple genes in each cell^{1,2}, the array-based strategy used here allows implementation of a much higher MOI (at least 2.3). The higher virus titer does not only increase the overall TF dose, but also the dose variation because of viral copy number variation between cells. Additionally, the transgene expression fluctuations due to the random insertion of viral vectors in genomic loci with different transcriptional activity, and the intrinsic fluctuations in promoter activity, as mentioned by the reviewer, further increase the heterogeneity of TF expression, which scTF-seq fully leverages. We would thereby like to point out that 0.5-2 ug/ml Doxycycline is the standard in the field for inducing exogenous gene expression^{1,2}, and we demonstrate here using new data that 2ug/ml Doxycycline results in saturated mCherry expression in the same vector as we use to overexpress TFs (**Rebuttal Figure 26** / revised **Supplementary Figure 1.1a**), aligning with previous findings³. All these characteristics have, to our knowledge, enabled us to achieve so far not only the highest TF doses but also a broad dose range for most TFs at the same Doxycycline concentration (**Figure 1f**).

Rebuttal Figure 26, added in revised Supplementary Figure 1.1a. Ridgeline plot showing the mCherry fluorescence intensity distributions measured by flow cytometry across different Doxycycline concentrations (0-4 µg/mL). Cells were collected from two mCherry-transduced cell lines, with two replicates each. WT refers to wildtype C3H10T1/2 cells.

In fact, and while we cannot rule out differences in how specific cell types accommodate high TF overexpression, our data offers a much wider dynamic range of TF doses compared to previous studies, making it based on our analyses the most robust study to date for assessing dose responses. Specifically, we compared FOS overexpression in our data with that from a recent study (Joung et al. 2023)¹ which uses a constitutive EF1a promoter and low MOI < 0.3. Even though our study has fewer cells (396 FOS and 213 batch-paired control cells for scTF-seq *versus* 990 FOS and 565 control cells for Joung et al. 2023¹), scTF-seq identifies over 5 times the amount of differentially expressed genes (1733 *versus* 259; fold-change > 2, FDR < 0.05) (**Rebuttal Figure 27a / revised Supplementary Figure 1.2a**). Furthermore, the fold-changes are much larger, consistent with the notion that the TF was expressed at higher levels (**Rebuttal Figure 27a / revised Supplementary Figure 1.2a**). Indeed, a visual inspection of some common differentially expressed genes reveals that the dynamic range of the dose appears to be much wider in scTF-seq, with smooth and non-linear dose-dependent gene expression patterns (**Rebuttal Figure 27b / revised Supplementary Figure 1.2b**), while the approach presented by Joung et al. 2023¹ showed a more binary response (**Rebuttal Figure 27c / revised Supplementary Figure 1.2c**). We simulated what would happen to scTF-seq's fold-changes if it were constrained by a smaller TF dose, and compared the distribution of these values with that from Joung et al. 2023¹, showing that the corresponding dose is around 5, thus several orders of magnitude smaller than our dose levels (capping at 7.8) (**Rebuttal Figure 27 / revised Supplementary Figure 1.2d**), aligning with the different viral titers applied. Such a dose is sufficient to observe some effects (**Rebuttal Figure 27a, b / revised Supplementary Figure 1.2a, b**), as demonstrated by Joung et al. 2023¹. However, it is not enough to capture the full dose-response curve, thereby missing many differentially expressed genes, as also detailed above. Overall, this shows that our scTF-seq design is so far unique in enabling the mapping of wide dose-responses for a large number of TFs.

Rebuttal Figure 27, added in revised Supplementary Figure 1.2a-d. (a) Volcano plots for differential expression between control cells and FOS cells in our study (batch 9) and Joung et al. 2023¹. (b-c) Exemplary UMAP plots for the expression of six shared up- and down-regulated genes in response to FOS overexpression. (d) Distribution of fold-changes when filtering the scTF-seq cells according to several maximal dose cutoffs (red) compared to the distribution of differential expression from Joung et al. 2023¹. Box indicates first and third quartiles, line the median, and whiskers first and third quartiles expanded with 1.5 times the interquartile range.

To ensure that a broad dose range is available to reliably fit the dose-response curves, as shown in the revised **Supplementary Figure 3d** and **Methods**, we applied new quality controls. Specifically, we excluded 21 TFs with fewer than three cells in the lowest dose bin (dose < 1.68) and 33 TFs lacking a sufficient maximum dose (3.5) before applying the logistic model.

Achieving a high enough dose is a key determinant of the power to classify a TF as high/low capacity. In this regard, we agree with the reviewer that there may be false-negatives among the low-capacity TFs in our analyses due to a low maximal dose. To address this, we have now performed a power analysis to evaluate whether our scTF-seq data is still adequate to detect high-capacity TFs at high power. We used the high-capacity, high-dose TFs in our study to simulate the max dose level at which these true high-capacity TFs start to be classified as false-negative, low-capacity TFs. We found that the false-negative rate indeed increases with lower maximal dose, passing 50% at a dose of 3.5 (**Rebuttal Figure 28** / revised **Supplementary Figure 3f-h**). Among the TFs with a maximal dose higher than 3.5, we estimate that about 23% or 36 TFs may be falsely classified as low-capacity, i.e., our total power is 77%. This is very close to the “standard” power level of 80% used in many statistical settings (e.g., clinical trials, omics⁴). As such, we believe that the dose levels in our study are high enough to correctly resolve the regulatory capacity of the vast majority (198 out of 234) of TFs.

Rebuttal Figure 28. Power analysis for detecting high-capacity TFs with respect to maximal dose, added in revised Supplementary Figure 3f-h. (Top) sensitivity to identify a TF as being high-capacity in function of the maximal dose reached by a TF. (Middle) Overall transcriptomic change compared to maximal dose, with the probability for a TF as being identified as a false-negative, low-capacity TF highlighted in color. (Bottom) Predicted number of false-negative, low-capacity TFs at various maximal dose bins.

To ensure that the reader understands the importance of the dose for TF classification, and the caveat that some TFs classified as ‘low-capacity’ (approximately 23%) may still be high-capacity in the context of a higher respective dose, we included this analysis in the revised manuscript (lines 274-283, Supplementary Figure 3f-h), and included the “false negative probability” for each TF in Supplementary Table 4.

Minor comments:

It should be made clear in the main text that each TF is transduced separately before pooling the cells, as this is not immediately obvious. It would be helpful if the authors can explain the criteria for selecting the 384 TFs. Are these TFs expressed/not expressed in MSCs? Are they known to be involved in MSC reprogramming?

Response 4:

We have clarified in the revised manuscript that each TF was transduced separately (**lines 106-107**). Additionally, we described in the revised **Methods** section that we selected those TFs largely because of available ORF resources in the lab and EPFL Genomics core facility. Those TFs have varied expression levels in wildtype C3H10T1/2 cells, from highly expressed, such as FOS, to lowly expressed, such as MEIS2, but also others like ESR2, which are not endogenously expressed. Our selection also included TF families (such as TF homologs shown in **Figure 2c** and **Supplementary Figure 2i**), as well as TFs known to play key roles in cell reprogramming and MSC lineage differentiation, such as PPARG, RUNX2, and MYOG for adipose, osteo, and myo lineage development, respectively.

Reviewer #2:

Remarks to the Author:

Overall this was a very well written paper and the experiment is well designed (e.g. arrayed format to avoid recombination etc.) and analyzed. The analysis of TF dosage in particular is interesting and novel.

Response 1:

We thank Reviewer 2 for their positive comments and appreciation for our design and analysis.

Major points:

- Apart from the other points mentioned below, it is important that the authors could experimentally validate some of their conclusions. Mainly because their cell numbers per perturbation are quite low, so replicating two or three of these results in an orthogonal way would increase confidence in them. This could be quite simple, e.g. show that low Pparg dose leads to higher adipogenic capacity than higher doses via qPCRs.

Response 2:

We thank Reviewer 2 for the thoughtful suggestion. To increase the cell number per perturbation and validate cell-fate branching, dose-dependent and/or low-capacity TFs, we first conducted a new scTF-seq experiment. This experiment produced an average of 441 cells per TF, ensuring robust statistical power. We thoroughly assessed the reproducibility of our methodology and the conclusions presented in the manuscript, confirming that our findings are consistent and well-supported. We have integrated the additional scTF-seq data and analyses in the revised figures and manuscript. We kindly refer the reviewer to the **Response 2 to Reviewer 1** for detailed analyses (pages 3-25).

Here, as suggested by the reviewer, we focused on illustrating the validation of the dose-dependent cell fate branching using multiplex RNA in situ hybridization (RNAscope) which is orthogonal to scTF-seq. While this approach offers exceptional sensitivity and specificity, it is limited by low throughput, as it can detect only up to four RNA targets simultaneously. We thus focused on two representative dose-dependent branching TFs: KLF4 and ESR2. For each of them, two marker genes that are specific to the observed branching subpopulations were selected (**Rebuttal Figure 29a** / revised **Figure 4i** and **Supplementary Figure 4.2i**). We probed RNA transcripts of the selected marker genes, along with WPRE, a viral element located on the vector

adjacent to the TF-ID, which serves as a proxy for TF dose. We quantified fluorescence intensities, representing expression levels for each marker or TF dose, on a per-cell basis across individual channels. As illustrated in **Rebuttal Figure 29b** / revised **Figure 4h, j**, distinct cell populations displayed mutually exclusive expression of *Glul* and *Postn* in response to low *versus* mild/high *Klf4* overexpression. Moreover, the dose-dependent, non-monotonic expression patterns of *Gng12* and *Aspn* in ESR2 cells, initially observed via scTF-seq, were confirmed by the RNAscope experiments which nicely captured a full range of *Esr2* dose variations (**Rebuttal Figure 29a, c** / revised **Supplementary Figure 4.2i-k**).

Rebuttal Figure 29, added in revised Figure 4h-j and Supplementary Figure 4.2i-k. (a) Log-normalized maker gene expression in control and KLF4 (left pair) or control and ESR2 (right pair) cells detected by scTF-seq. Cells are colored by clusters as shown in Rebuttal Figures 16 and 18. Ctl refers to control clusters. **(b, c) (Left images)** Representative images

of multiplex RNA in situ hybridization (RNAscope) detecting RNA transcripts of *WPRE* (representing the TF-ID), *Postn*, and *Glul* in *KLF4* cells, or RNA transcripts of *WPRE*, *Gng12*, and *Aspn* in *ESR2* cells. Wildtype *C3H10T1/2* cells were used as the control. Nuclei were stained by DAPI. Scale bar, 100 μ m. All fluorescent channels were merged for cell segmentation, indicated by the red (cell boundary) and purple (expanded cell boundary) outlines. The mean fluorescence intensity was quantified for each segmented cell. **(Right panels)** Scatter plots showing the correlation between the normalized mean fluorescence intensity of each gene and *WPRE* in TF (*KLF4* or *ESR2*) and wildtype control cells. Cell counts: for *KLF4* and control in panel b, $n = 713$ and 747 , respectively; for *ESR2* and control in panel c, $n = 300$ and 481 , respectively.

- There appear to be some batch effects between the experimental batches (Fig. 2C, 1e), and likely there are also some batch effects across wells within the experiment, given the arrayed format. Can the authors assess well-to-well variability by measuring the same effects as they measure for TFs for mCherry (using a different mCherry well in the same batch as control)? Can they put the observed variances associated with TFs from different batches in context with the observed variances observed for mCherry within the same and across batches? Were some TFs in all batches and if so how does their variance compare to intra-batch inter-TF variance? Did they measure multiple wells per TF and batch, and if so how well do these replicate? Also the batch composition information should ideally be provided somewhere in a table.

Response 3:

We thank the reviewer for raising this point. To clearly show the batch correction, we have now added the UMAP plot of the batch-uncorrected TF atlas in the revised **Supplementary Figure 1.1f (Rebuttal Figure 30 left)**. Comparison of this UMAP plot to revised **Figure 1e (Rebuttal Figure 30 right)** shows a successful batch correction on the cell transcriptomes given that control or TF cells from different batches are well mixed together.

Rebuttal Figure 30, added in revised Figure 1e and Supplementary Figure 1.1f. UMAP plots of the batch-uncorrected (left) or batch-corrected (right) scTF-seq atlas colored by experimental batch.

The reviewer correctly deduced that in the original Figure 1e (also in **Rebuttal Figure 30 right** with batch 9 added), the transcriptomic pattern of the pink batch (batch 8) reflects the enrichment of certain TF groups in that batch. In particular, in that final batch of our initial submission, we specifically analyzed a large number of cells for specific TFs, including RUNX2, CEBPA, MYOG, and MYCN, which exhibited strong reprogramming effects in batches 1-7 (**Rebuttal Figure 31**). This allowed us to establish a robust baseline of the main lineage effects in the data. However, despite this imbalance in cell numbers, the cells from these TFs clearly cluster together with their respective cell counterparts from different batches (**Rebuttal Figure 31**), indicating that the batch effect is well corrected. The remaining subtle differences (also in original Figure 2c) are mostly attributable to dose differences, as shown in **Rebuttal Figure 32 right**. We kindly refer the reviewer to the **Response 2 to Reviewer 1** for detailed analyses of the dose (**page 7**).

*Rebuttal Figure 31. UMAP plots of the scTF-seq atlas colored by RUNX2 (top) or MYOG (bottom) cells in different experimental batches. The number in parentheses indicates the number of cells overexpressing the specific TF per batch. A variant of this panel is included in the revised manuscript as **Supplementary Figure 1.3a**.*

Rebuttal Figure 32, related to Figure 2c. Heatmap showing a pairwise Pearson correlation of functional TF cells from the unintegrated (**left**) or integrated (**right**) scTF-seq atlas. Cells are annotated by TF (in column) and batch (in row) and ordered by hierarchical clustering. We revised **Figure 2c** which is similar to the right panel to incorporate data from batch 9.

Besides, as also correctly noticed by the reviewer, in our array-based scTF-seq experiments, we did include well replicates for certain controls and TFs, each uniquely barcoded with distinct TF-IDs for demultiplexing. We found that there is minimal well-to-well variability, as evidenced by mixed cells overexpressing mCherry or RUNX2 across separate wells within the same batch (**Rebuttal Figure 33**).

The batch composition and well information (indicated) has already been provided in the original **Supplementary Table 1**. We have now integrated batch 9 in the revised **Supplementary Table 1**.

Rebuttal Figure 33. Cell clustering is reproducible across different wells. UMAP plots of the scTF-seq atlas on which non-confluent (Ctr.non.conf) or confluent (Ctr.conf) mCherry-expressing control cells, or RUNX2 cells from multiple wells of batch 6 are colored by well (W).

- It would be good to put the observed TF levels into context with endogenous TF levels (e.g. using non TF-enriched single cell data). For example generate a version of Figure 1f for endogenously expressed TFs and ORF TFs without TF enrichment. This is key to assess whether the generated levels are far beyond or similar to what is normally observed.

Response 4:

We thank the reviewer for this important point. In addition to the TF doses derived from enriched libraries (with an additional amplification step and a higher TF-ID recovery rate, **Methods**), we now analyze the normalized counts of both exogenous and endogenous TFs from the non-enriched, conventional 10x libraries as suggested (**Rebuttal Figure 34 / Supplementary Figure 1.1h**). As expected, the endogenous expression levels varied across TFs, with some TFs not or very lowly expressed in control cells at the basal state. In addition, the exogenous TFs have a much larger range of expression levels than the endogenous ones, as seen in both enriched and non-enriched libraries. Consistent with common practices in many cell reprogramming applications^{5,6}, which use multiple copies of viral transgenes to enhance reprogramming efficiency, our scTF-seq approach employed a high virus titer (MOI > 2.3) to achieve elevated ectopic TF expression. This high titer also contributes to dose variations because of the viral copy number variation between cells. Additionally, random integration of viral vectors into genomic loci with differing transcriptional activity and intrinsic fluctuations in promoter activity further enhances the heterogeneity of TF expression. These factors collectively generated a broad range of TF doses in scTF-seq, enabling the quantitative mapping of TF dose effects on cell reprogramming.

Rebuttal Figure 34. Natural log-normalized counts of the reads mapping to the vector in non-enriched 10x libraries (a) and the endogenous expression of individual TFs in mcherry-overexpressing control cells (b) or TF cells (c). The TFs are in the same order in a, b, and c. Colors indicate cell density (number of neighbors) after randomly sampling up to 900 cells for each TF. Figure 1f panel is added for comparison. Panels b and c were added in revised Supplementary Figure 1.1h.

- 'Euclidean distance of cells reprogrammed by TFs to the centroids of control cells in the space consisting of significant principal component coordinates' is used for assessing functional TF cells. It is notoriously challenging to accurately measure distance in high-dimensional space. Can the authors show that similar results are obtained with a different distance measure, e.g. cosine distance?

Response 5:

Accurately measuring distance in high-dimensional space is indeed challenging. To address this, we first applied principal component analysis (PCA) to the normalized gene expression data, reducing the dimensionality to 10 components. This transformation captures the primary variance within the data, projecting it onto a new coordinate system defined by individual principal components (PCs). We performed this dimensionality reduction separately for each combination of TF and its batch-paired control cells.

To identify functional TF cells, PCA was performed for cells of each individual TF and its batch-paired control in each phase. The PCA space is mostly dominated by the monodirectional difference between TF and control cells. Therefore, we prioritized capturing the magnitude of variation rather than merely the direction of changes between TF cells and the centroid of control cells. We opted to use Euclidean distance, which considers the actual values of PC coordinates and captures absolute differences more effectively than Cosine distance, which measures the orientation differences. This approach allowed us to measure distinct deviations in TF cells from controls, as demonstrated in **Rebuttal Figure 35** / revised **Supplementary Figure 2a-c**.

For comparison, we also calculated the Cosine distance from TF cells to the mean of control cells in the same PCA space. As illustrated in **Rebuttal Figure 36**, while Cosine and Euclidean distances show some correlation ($r = 0.32$, P value $< 2.2e-16$), Cosine distance overlooks meaningful differences when PC coordinates differ significantly in magnitude. This reinforces our original choice of Euclidean distance as a more appropriate metric for capturing the variations essential to functional cell identification. We now clearly describe the reason for using Euclidean distance in the revised **Methods** section “Selection of functional cells”.

Rebuttal Figure 35, added in revised Supplementary Figure 2a-c. An example of identifying functional TF cells. PCA analysis (a) was performed on TF cells of one batch and their batch-paired control (ctr) cells involving both non-confluent (Ctr.non.conf) and confluent (Ctr.conf) mCherry-expressing C3H10T1/2 cells, allowing to calculate the Euclidean distance between each cell to the mean of control cells of interest in the respective PCA space. TF cells having a distance larger than the distance that 80% of control cells have to their mean (b) were selected as ‘functional’ (funct) cells. The Euclidean distance of functional and non-functional cells was plotted over TF dose (c).

Rebuttal Figure 36. Scatter plot showing the correlation ($r = 0.32$, p -value $< 2.2e-16$) between Cosine distance and natural log-transformed Euclidean distance calculated during functional cell identification.

- The cell numbers are very low in some cases (e.g. 4f the clusters only contain 20 or so cells - are these from same or different batch?). Likewise in that figure it is unclear which cells are actually control cells. Cells would need to be colored by batch and control.

Response 6:

We performed a new scTF-seq experiment (batch 9) to specifically target a large number of cells for cell fate branching TFs, with an average of 1124 cells per branching TF after quality control. There are now at least two batches (replicates) for cell fate branching TFs. We have clearly colored cells by batch and TF/control in our revised figures (**Rebuttal Figure 16-24** / revised **Figure 4** and **Supplementary Figures 4.1-4.3**). We kindly refer the reviewer to **point 5 of the Response 2 to Reviewer 1** for detailed analyses.

Minor questions:

- why these 419 TFs, which criteria were applied? Are they already expressed endogenously in the starting cell line?

Response 7:

We have clarified in the revised manuscript that each TF was transduced separately (**lines 106-107**). Additionally, we describe in the revised **Methods** section that we selected the 419 TFs largely because of available ORF resources in our lab and in the EPFL genomics Facility. Those TFs have varied expression levels in wildtype C3H10T1/2 cells, including some that are highly expressed, such as FOS, some that are lowly expressed, such as MEIS2, and others like ESR2, which lack endogenous

expression. Our selection also included TF families (such as TF homologs shown in revised **Figure 2c** and **Supplementary Figure 2i**), as well as TFs known to play key roles in cell reprogramming and MSC lineage differentiation, such as PPARG, RUNX2, and MYOG for adipose, osteo, and myo lineage development.

- what is the transduction rate, do cells receive 1 or potentially multiple TF-IDs based on a Poisson distribution? This wouldn't largely affect results given the arrayed format, but would help to put the 86% assignment rate into context

Response 8:

Because of the high throughput and arrayed format, we did not measure the multiplicity of infection (MOI) for each individual well. Instead, we excluded wells with over 10% cell loss following puromycin selection. However, based on Poisson's distribution model, the estimated MOI was at least 2.3. We have provided this information in the revised **Methods** section.

- Why are there only 13 TFs in Figure 2c?

Response 9:

In Figure 2c, the 13 TFs were chosen as examples showcasing that functional TF modules governing the same gene expression programs may or may not be from the same TF families. Other TFs are shown in **Supplementary Figure 2i** as TF families.

- is there a correlation between mCherry fluorescence and mCherry TF-ID? This would be interesting to see whether all TF-ID constructs are functional (although potential truncations will be length dependent)

Response 10:

We measured the RNA transcripts of WPRE (representing the TF-ID) by in situ hybridization (RNAscope) and the mCherry fluorescence intensity in parallel in *mCherry*-overexpressing cells. As shown in **Rebuttal Figure 37** / revised **Supplementary Figure 1.1i**, the increased TF-ID expression correlates monotonically with the enhanced mCherry fluorescence intensity.

Rebuttal Figure 37, added in revised Supplementary Figure 1.1i. Correlation between mCherry fluorescence and mCherry TF-ID. (Left images) Representative images simultaneously showing the RNA transcripts of WPRE (representing the TF-ID) detected by in situ hybridization (RNAscope) and mCherry fluorescence in mCherry overexpressing cells. Wildtype C3H10T1/2 cells were used as the control. Nuclei were stained by DAPI. Scale bar, 100 μm . All fluorescent channels were merged for cell segmentation, indicated by the red (cell boundary) and purple (expanded cell boundary) outlines. The mean fluorescence intensity was quantified for each segmented cell. (Right) Scatter plot showing the correlation between the normalized mean fluorescence intensity of mCherry and WPRE in control and mCherry cells. The data are collected from three well replicates and pooled together after normalization. Cell counts: for control and mCherry cells, $n = 349$ and 677 , respectively.

Minor point:

- Add fold-change to enrichment tests (e.g. line 243)

Response 11:

We have added the odds ratio (~ 1.97) for the Fisher's exact test performed on pLI and LOEUF scores of high-capacity and low-capacity TFs (as shown in revised Supplementary Figure 3e).

References

1. Joung, J. *et al.* A transcription factor atlas of directed differentiation. *Cell* **186**, 209–229.e26 (2023).
2. Ng, A. H. M. *et al.* A comprehensive library of human transcription factors for cell fate engineering. *Nat. Biotechnol.* **39**, 510–519 (2021).
3. Lamartina, S. *et al.* Construction of an rtTA2s-m2/ttskid-Based transcription regulatory switch that displays no basal activity, good inducibility, and high responsiveness to doxycycline in mice and Non-Human primates. *Mol. Ther.* **7**, 271–280 (2003).
4. Baker, E. A. G., Schapiro, D., Dumitrascu, B., Vickovic, S. & Regev, A. In silico tissue generation and power analysis for spatial omics. *Nat. Methods* **20**, 424–431 (2023).
5. Wernig, M. *et al.* A drug-inducible transgenic system for direct reprogramming of multiple somatic cell types. *Nat. Biotechnol.* **26**, 916–924 (2008).
6. Polo, J. M. *et al.* A Molecular Roadmap of Reprogramming Somatic Cells into iPS Cells. *Cell* **151**, 1617–1632 (2012).

Rebuttal Manuscript #NG-A64657R:

“Dissecting the impact of transcription factor dose on cell reprogramming heterogeneity using scTF-seq”

Liu, Saelens*, Rainer* et al.*

Reviewer #1 (Remarks to the Author):

The revised manuscript from Liu et al does a thorough job of addressing the two main points raised in review, reproducibility and batch effects. With respect to reproducibility the inclusion of a new experiment (batch 9) and the analysis of TFs shared between batches provides enough data for readers to assess the reproducibility of the technique. With respect to batch effects, the revised Figure 2C shows clearly that most of the major effects are observable in multiple batches. I also appreciated the attempt to include batch-specific scaling factors to deal with more subtle effects. I thank the authors for providing such a clear and detailed response to the critiques.

Response 1.1:

We thank the reviewer for appreciating our work and efforts.

Reviewer #2 (Remarks to the Author):

Deplanke and colleagues have done an excellent job of revising the manuscript. This paper represents an enormous amount of work. The authors are to be applauded on a careful and thought-provoking study. I also appreciate the detailed response and yellow marking of changes. Note that I previously co-reviewed this with a trainee who has since moved on to bigger and better things, so I apologize if this feels like a relatively “fresh take” (I had read it previously and edited/appended their review, but I’m taking more of the lead on the re-review). My new comments (some of which I imagine could have been raised originally, but others of which were prompted by certain revisions) are offered in a constructive spirit in hopes of helping you further improve the manuscript, and none of the suggestions below are “deal breakers” from my perspective. But particularly the ones listed under “Major”, I think are relatively straightforward and would add considerable value for many reviewers if addressed, even if briefly.

Response 2.1:

We thank the reviewer for recognition of our revision efforts. We truly appreciate the new constructive feedback provided, as well as the time taken to re-review the manuscript. We have carefully considered each comment and addressed the

remaining concerns in detail below. We believe that this revision has further improved the clarity of our manuscript.

Thank you again for your thoughtful consideration and valuable input.

Major

- Legends need careful review for completeness. Various axis labels are also missing. As one example looking at Fig 2c and that dotted red box and no discussion in legend re: what it is. I strongly suggest a full review of your figures/legends for completeness w/r/t labeling of axes and explaining everything in each figure in the corresponding legend, not just in the text.

Response 2.2:

We acknowledge that the legend did not explain the meaning of the dotted red box in Figure 2c. We have updated the legend for Figure 2c with a clear explanation. Furthermore, we have reviewed all figures and legends. Where necessary, we have added additional details (highlighted in Okabe-Ito orange with underlines) to ensure that all elements of the figures are thoroughly explained in the respective legends. We believe this revision improves the overall clarity of the manuscript.

- I'm sorry if I missed this somewhere but I feel like the lack of a clear "key" for evaluating the extent to which the distribution of expression of exogenous TFs overlaps (or doesn't) w/ endogenous TF expression levels in the context of these differentiations feels really missing. As you know, one of the main concerns re: TF overexpression schemes is that the doses used are non-physiologic. What part of the range of each TF corresponds to physiologic here? I don't want be too prescriptive about how you do this, but it really does feel missing now. Can you either look at the levels of key TFs (e.g. RUNX, etc.) that are know to play roles in certain cell types and ask how their levels look in the context of cocktail driven differentiation, or in vivo single cell data from mouse? (e.g. calibrating to some housekeeping genes or something like that). Some calibration for the reader of what "regime" we are in here would be really helpful.

- On a related point, the analysis in Supp Fig 1.1H is very helpful, but I am confused about what the y-axis units are, and the claim that this is less variation than seen w/ the exogenous TFs doesn't really feel well supported by this histogram on its own. Is there some way of representing this information so we can compare both the variation and the mean levels of the TFs' exogenous vs. endogenous expression? (again, more interested in what the TFs' expression levels are like during natural differentiation along a process like adipogenesis, than in their baseline levels in MSCs/ESCs).

Response 2.3:

Since these two comments are related, we decided to address them together.

We agree that putting exogenous TF expression in the context of endogenous or physiological TF expression is an important consideration that has to be made. We also acknowledge that the previous Supplementary Figure 1.1h may not have represented this comparison as effectively as we had intended. To improve this, we have now included a new analysis comparing “normal” endogenous dose (derived from *in vivo* data of specific cell types) with exogenous TF expression (dose). To obtain dose values that are relatively comparable across various studies and single-cell technologies, we normalized TF expression against 13 housekeeping genes from a variety of central cellular processes (cytoskeleton, translation, ubiquitination, ...). To minimize bias and contamination issues, we looked across all cell types within the CELLxGENE census database, covering over 150 annotated single-cell datasets. We then compared our data to cell types with the highest expression of each TF and found that the latter often align well with known TF-cell type associations. We have also included “endogenous” TF expression as an additional factor within the analysis. As shown in a later comment (**Response 2.11**), exogenous TFs often induce or repress their endogenous counterparts, adding another dimension to the dose-response relationship.

To illustrate this analysis, we first show the comparison of *Fosl2*, *Klf12*, *Myog*, *Runx2*, *Nfkb1*, and *Pparg*, representing different types of dose effects (**Rebuttal Figure 1**).

- *Fosl2* and *Klf12* represent cases where a moderate amount of overexpression is necessary to induce a significant change in expression. For *Fosl2* (**Rebuttal Figure 1a**), this is above normal physiological levels in its functional contexts such as adipocytes and mesothelial cells. For *Klf12* (**Rebuttal Figure 1b**), this is exactly at its physiological levels in functional contexts of neuronal, retinal and liver development.
- *Myog* and *Runx2* represent cases where even a very small amount of exogenous expression is enough to trigger a strong response. For *Myog* (**Rebuttal Figure 1c**), this matches well with levels for its physiological context (muscle cells). For *Runx2* (**Rebuttal Figure 1d**), we highlight that its endogenous dose is very diverse across functional contexts (plasmacytoid dendritic cells, osteoblasts, neurons, chondrocytes, ...) although this full dose range is well-captured by the heterogeneity present in scTF-seq. This further confirms its strong dose-dependent (and often non-monotonic) effects as we have highlighted in our manuscript.
- *Nfkb1* (**Rebuttal Figure 1e**) and *Pparg* (**Rebuttal Figure 1f**) represent cases where a high amount of overexpression is necessary to reach a transcriptional response, far above the endogenous requirements. This is likely because these TFs need extra activation, respectively in the form of phosphorylation or ligand

binding, to be activated. Others have indeed already noted the “absurd” dose-dependency for *Pparg* before¹.

Rebuttal figure 1: Comparisons between exogenous TF expression in scTF-seq data with endogenous TF expression derived from in vivo single-cell atlases. Top scatter plots indicate the change in overall transcriptomic response (distance in PCA space to control cells) over various doses. The dot highlights the dose at which the overall transcriptomic change is above 0.23, the cutoff used in the manuscript to represent a “functional” TF. Bottom boxplots show the range of doses in the given cell type (standard boxplot representing 25 and 75 percentiles, and 1.5 IQR as whiskers), with the mean represented as the white dot. The dose here is

defined as the $\log_1 p$ (natural logarithm of 1+x) ratio in expression between TF and mean housekeeping genes (*Actb*, *Gapdh*, *Ppia*, *Ppib*, *Tubb5*, *Rpl13*, *Rpl13a*, *Rpl19*, *Ubc*, *Gusb*, *Ywhaz*, *Eef1a1*, *Pgk1*). In green, endogenous expression for induced adipogenesis or myogenesis. In blue, the endogenous gene expression in mCherry treated cells, and in purple this expression added to the exogenous expression.

We expanded this analysis towards all functional TFs, formulating the following definitions: we defined the minimal functional dose as the exogenous dose required to reach a strong transcriptional effect (>0.23 overall transcriptomic change, same cutoff as previously used in our manuscript, highlighted in **Rebuttal Figure 1**). We compared this minimal functional dose to the range (5%-95% quantile) of endogenous doses found *in vivo*, which we call the physiological dose (**Rebuttal Figure 2**). This physiological dose is sometimes quite narrow (e.g. for *Fos* in pancreatic polypeptide cells), while for others large (e.g. for *Pparg* in adipocytes).

For about half (47%) of TFs, the minimal functional dose falls within the physiological range. Notable exceptions that have a high physiological dose yet require an even higher exogenous dose to be functional include *Pparg*, *Nfkb1*, *Atf7*, *Vsx2*, and *Gcm2*. What is quite striking is that numerous homeodomain TFs (*Hoxa6*, *Cdx2*, *Nkx3-1*, *Rhox10*, *Six2*, *Nkx1-2*, *Hoxa13*, *Cdx4*, *Rhox12*, *Duxbl1*, *Nkx2-6*) have a high minimal functional dose, despite being lowly expressed endogenously. As with the previously highlighted *Nfkb1* and *Pparg*, this is likely because these TFs function synergistically during specific developmental phases. Nonetheless, we do want to highlight that many TFs for which key conclusions are made in the paper, such as *Runx2*, *Meis2*, *Cebpa*, *Klf4*, and *Fos*, have minimal functional doses that lie within the physiological range.

All-in-all, this analysis paints a nuanced picture of the dose levels obtained during overexpression. Yes, it is true that overexpressed doses are often higher than those in physiological conditions (the blue versus orange ticks in **Rebuttal Figure 2**). Yet, the minimal dose required to reach an effect does fall within the physiological range about half of the time (the green/red circles in **Rebuttal Figure 2**). Given the very variable dose ranges in physiological conditions, we cannot precisely denote at which regime we are operating here, because this truly depends on the TF in question. We have now replaced previous Supplementary Fig 1.1h with representative examples from these analyses, which we have added to revised **Supplementary Figure 3c, d** and **lines 250-260** and which are also detailed in the **Methods** section (**lines 910-922**).

Rebuttal figure 2: Comparison of the minimal functional dose required to reach a significant transcriptomic effect (circles) to the high endogenous dose observed in *in vivo* single-cell datasets (grey bands). The minimal functional dose is colored in respectively teal and red if it is above or below the physiological dose. The small vertical lines indicate the maximal dose observed in our study and are colored blue, except when the maximal dose falls below the highest dose observed endogenously, in which case they are colored orange.

- Fig 2B lacks explicit labelling of which cells correspond to “cocktail controls” (i.e. differentiated w/ chemical cocktail) so I can’t really judge the claim that certain clusters also have those cells. What I would really like is for you to make separate UMAPs (fine if they are supplementary) for each of the four major differentiated cell types (e.g. adipogenic, osteogenic, etc.), taking only cells from those clusters and labeling them by which TFs they are expressing OR if they are cocktail controls. This would give the reader clarity on how much substructure there is within a given cell type and how much of that is a consequence of the specific TF (or chemical cocktail). This relates to the Results around lines 230-239 and could be cited there. For example, I’d like to see a UMAP that lets me see if adipocytes expressing MYCN cluster separately from adipocytes expressing CEBPA, PPARG, RHOX12 or adipocytes generated via the chemical cocktail. This is hard to discern in a composite UMAP of the entire dataset.

Response 2.4:

To ensure greater clarity on the substructure of the four main multipotent stromal cell (MSC) clusters in the functional TF atlas, we have included a new UMAP plot in the revised **Supplementary Figure 2g / Rebuttal Figure 3**. Unlike MYOG, MYOD1, and myogenic reference (Myo ref) cells, which show substantial overlap, most MYCN cells are positioned in close proximity to, but distinct from, CEBPA, PPARG, and adipocyte reference cells (Adipo ref). This suggests a unique profile induced by MYCN, as previously discussed in the manuscript (**lines 234-241**).

Rebuttal Figure 3, added in revised Supplementary Figure 2g. UMAP plot showing a subset of the functional TF atlas, including cells in the four major MSC clusters (adipogenic, myogenic, osteogenic and inflammatory) in **Figure 2b**. Cells are colored by TF. Only TFs that have more than 25 cells were plotted. Adipo ref or Myo ref, adipocyte or myo reference cells. Ctr.conf, confluent control cells.

- The material around lines 270 and 300 are related (i.e. what are the features of TFs in the three classes) and should be consolidated to the same part of the paper. Also, readers will be highly interested in knowing if particular kinds of TFs (e.g. ZFs, homeodomain TFs) are enriched/depleted in the three TF groupings. Can you add a quick enrichment analysis along these lines? For example, I imagine homeodomain TFs might be enriched amongst high-capacity, high-dose sensitive TFs.

Response 2.5:

We agree with the reviewer and have now consolidated the features of TFs to the same part of the paper (**lines 278-302 and 522-525**). We also thank the reviewer for suggesting the enrichment analysis on TF classes, particularly for zinc-finger (ZF) and homeodomain TFs, given that they represent the two largest families among the 234 high-capacity and low-capacity TFs analyzed (55 ZFs and 38 homeodomain TFs, respectively). Interestingly, a Fisher's exact test showed that ZFs are less enriched in high-capacity TFs (0.32 odds ratio, p-value: 0.001842, 95% CI: 0.1389-0.6998), while homeodomain TFs are significantly overrepresented in high-capacity TFs (3.78 odds ratio, p-value 0.0003481, 95% CI 1.714-8.618), consistent with the reviewer's hypothesis. This latter overrepresentation is quite striking considering the strong dose-requirements of these TFs to induce a significant transcriptomic effect (**Rebuttal Figure 2**). We have added this enrichment analysis to the revised manuscript (**lines 285-291**).

Minor

- You are inconsistent and incorrect in your style for naming genes/proteins, to my knowledge. Mouse gene names should be capital first letter only, and italicized, and protein names should be the same but not italicized. Only human genes/proteins are "all caps". Paper and figures are all over the place w/r/t this, and should be made consistent/correct. I think the only place you should be using caps is when you are referencing literature that concerns human orthologs of mouse genes? (e.g *Mycn* (gene) and *Mycn* (protein), not MYCN; sorry to be the nomenclature police!)

Response 2.6:

We thank the reviewer for their attention to detail and for raising this important point regarding gene and protein nomenclature. However, according to the guidelines of MGI (<https://www.informatics.jax.org/mgihome/nomen/gene.shtml#ps>) and HGNC

(<https://doi.org/10.1038/s41588-020-0669-3>), protein names of both mouse and human should be “all caps”. In our manuscript, we consistently used gene names for TFs (italicized, with the first letter capitalized) when discussing TF (over)expression and TF dose given that we wanted to avoid referring to ‘proteins’ since we only read out mRNA levels. In contrast, we used protein names for TFs (capitalized) when emphasizing their functional roles and effects on reprogramming. For example, “RUNX2 cells” means cells reprogrammed by the RUNX2 protein and “Runx2 dose” means the level of the exogenous *Runx2* mRNA. We have carefully reviewed the manuscript to ensure strict adherence to the common nomenclature guidelines.

- Colors in many figures, particularly yellow tones/text, are really hard to see for me. Suggest careful review w/ an eye to optimal color schemes / consideration for the color blind.

Response 2.7:

We apologize for the visibility issues caused by the yellow tones and text. In our earlier submission, we had already implemented some color-blind-friendly palettes, such as Viridis, Inferno, and Okabe-Ito. However, we recognize the challenges of using multiple colors (e.g., for 9 batches in Figure 1e, 14 clusters in Figure 2b, or tens of TFs in Figure 2a, c) within the same plot. Following the reviewer’s feedback, we have carefully re-reviewed the color schemes throughout the manuscript by using color blindness simulators (e.g., Color Oracle) and made further adjustments to enhance accessibility, particularly for color-blind readers. Specifically, we modified the colors of some TF families in Figure 2c and all batch-related plots using Paul Tol’s Muted color scheme. Additionally, we have replaced yellow highlights with Okabe-Ito Orange (#E69F00) and underlines to indicate changes in the main text and figure legends in this revised version. We hope these revisions address the reviewer’s concern and improve the readability of our manuscript for a broader audience.

- Text clearer but Fig 1A still makes it look like this is pooled packaging and transduction. Suggest making it more consistent w/ the arrayed format in which you did it.

Response 2.8:

We have revised Figure 1a (**Rebuttal Figure 4**) accordingly to highlight the arrayed format of virus packaging and transduction.

Rebuttal Figure 4, added in revised Figure 1a. Schematic of the scTF-seq workflow.

- Early in Results or in Discussion, can you comment on the *source* of the variation in TF expression? I presume that it is just copy number variation on the high MOI transductions, coupled with expression variegation on whatever gets in, but it would be nice to see that stated (assuming I have that right)

Response 2.9:

We have now discussed the source of the TF dose variation in the revised manuscript (lines 145-150): “As intended in the design of scTF-seq, the array-based lentiviral transfection and transduction strategies allow the implementation of a high multiplicity of infection (MOI, **Methods**), leading to broad variations in viral copy number and transcriptional activity due to random transgene insertion. Combined with intrinsic fluctuations in promoter activity, these factors likely contribute to the substantial dose variation observed across cells for most TFs (**Fig. 1f**)”.

- I suggest being more careful, and potentially just avoiding altogether, the term “branching”. I felt a bit confused in those parts of the paper re: the difference between branching, vs. non-monotonic behaviors that might be better characterized as “nesting”. To me branching implies two mutually exclusive fates, which I recognize you think you are seeing for certain findings, but the UMAPs imply a continuity of states, one of which may simply precede the other (nesting?), with higher doses simply driving the cells further/faster. Not asking for any new work or analyses here, just a careful review of your language, perhaps more careful articulation of what branching means to you, and which examples are branching vs. simply non-monotonic.

Response 2.10:

We appreciate the reviewer’s insightful comments regarding the distinction between branching and nesting in our ‘trajectory’ analyses. We agree that branching, in its strictest definition, implies the emergence of mutually exclusive fates, whereas nesting suggests a hierarchical or continuous progression of states. Given this, we have carefully reviewed our use of “branching” and refined our terminology where appropriate to ensure greater clarity.

Importantly, we acknowledge a key limitation in our data: since our analysis is derived from a single time point, we cannot directly assess how TF dose-driven non-monotonic expression patterns relate to the dynamics of state transitions. This means that distinguishing between mutually exclusive fate decisions (branching) and continuous state progression (nesting) is inherently challenging. For example, in the KLF4-driven reprogramming trajectory, cells exposed to a low dose of *Klf4* may have constrained future possibilities that are distinct from the one induced by a high *Klf4* dose, suggesting a potential branching event. Alternatively, these low-dose KLF4 cells could represent an early state on a developmental trajectory, with high-dose KLF4 cells

representing a later stage. Given this uncertainty, we have taken a more cautious approach in our language. Specifically:

- We now explicitly acknowledge the limitations of single time-point data in distinguishing between these two scenarios (**lines 541-552**).
- Where relevant, we describe these transitions more generally as dose-dependent cellular state transitions, rather than committing to a strict branching model.
- For the MEIS2 example and other cases for which we reproducibly observed distinct cell states despite a similar TF dose, we kept the term ‘branching’, as this seems the most intuitive explanation, all while acknowledging the need for temporal resolution to more accurately infer trajectory models.

We are grateful for this suggestion, as it has helped us refine the precision of our interpretations and acknowledge the limitations of our dataset more explicitly. We hope these revisions address the reviewer’s concerns and improve the clarity of our discussion.

- Very optional: I would have appreciated a brief analysis re: whether the TFs that are implicated w/ a particular cell type are upregulated in that cell type when other TFs are used to induce it and/or with the chemical cocktail. In other words, is there a network among the TFs that is self-reinforcing, and any one of the TFs can be used to drive it (in which case not on the other TFs but also the endogenous TF might be upregulated?).

Response 2.11:

Certainly, key lineage TFs are induced upon adipogenesis or myogenesis (**Rebuttal Figure 5**), and key lineage TFs (*Cebpa*, *Pparg*, *Myod1*, *Myog*, *Runx2*) are sometimes induced by other TFs within the same lineage (**Rebuttal Figure 5**). However, these positive feedback loops are not absolute and many exceptions exist, so this is rather an intra-lineage enrichment rather than a general rule. What is very frequent though are positive self-loops, with nearly all adipo-, chondro-, or myo-lineage TFs positively regulating themselves (**Rebuttal Figure 5**).

We have now also included a new column titled “is.TFoe” in the revised **Supplementary Table 3**, which indicates whether the significantly up- or down-regulated genes in TF-overexpressing cells are the endogenous counterparts of the overexpressed TFs.

While we have observed some preliminary evidence suggesting certain TF self-reinforcement or TF-TF interactions, a more comprehensive understanding of these gene regulatory networks and their dynamics will require time-resolved investigation. We are currently working on adding temporal resolution to scTF-seq and plan to specifically map gene regulatory networks in future studies.

Rebuttal Figure 5: Positive/negative feedback loops between TFs. Indicated are groups of TFs uniquely associated with one of four key lineages, and what other TFs are activated/repressed (x-axis) upon overexpression (y-axis).

References

1. Zhao, J. *et al.* PPAR γ and C/EBP α enable adipocyte differentiation upon inhibition of histone methyltransferase PRC2 in malignant tumors. *J. Biol. Chem.* **300**, 107765 (2024).